# Metabolic syndrome promotes endometrial cancer by Oleic acid-mediated polyamine accumulation

Lirong Zhai[1,4], Yuan Cheng[1,4], Meixuan Wu[1], Tianzhuo Wang[2,3], Meichen Yin[1], Xiao Yang[1], Bowen Sun[1], Chengcheng Li[1], Miao He[1], Yi Sun[2], Yiqian Zhao[2], Yuqi Xing[2], Bo Liu[2], Ling Zhou[1], Yuanyuan Liu[1], Miao Yu[2], Yijiao He[1], Hongquan Zhang[2], Jun Zhan[2]✉ & Jianliu Wang[1]✉

Metabolic syndrome increases the risk of endometrial cancer development and progression, but the mechanism remains unclear. We find that polyamine metabolites are notably elevated in the sera and tumor tissues of endometrial cancer patients with metabolic syndrome. Oleic acid, one of the many components in hyperlipidemia, is the key factor for upregulating Ornithine Decarboxylase 1 (ODC1) (the rate-limiting enzyme in polyamine metabolism) and downstream polyamines. Mechanistically, Oleic acid binds to and stabilizes Homeobox B9 (HOXB9) by inhibiting the binding of HOXB9 to E3 ligase Praja2. Stable HOXB9 then competes with OAZ1 and combines with ODC1 to block ODC1 degradation. Targeting HOXB9 or ODC1 reduces polyamine levels and suppresses tumor growth/spread. Oleic acid-HOXB9-ODC1 stable cascading axis then is confirmed in patient tissues, and ODC1 inhibitors boost patient-derived tumor cells' chemosensitivity. This study links fatty acids to polyamine buildup, reveals a mechanism for metabolic syndrome-driven endometrial cancer, and points to HOXB9 and ODC1 as potential therapeutic targets.

Endometrial cancer (EC) is currently the fourth most prevalent cancer among women and the most common gynecological malignancy[1–3]. Unlike most other cancers, the incidence of EC has shown a steady increase over the past 20 years, and its disease-related mortality has continued to rise. The incidence of EC has increased by 3.4% over 20 years, and the mortality has increased by an average of 1.9% per year[4]. Although 67% of patients present with early-stage disease, which has an 81% 5-year overall survival (OS), late-stage metastatic EC has a very poor prognosis, with a 5-year OS of only 17%[5,6].

EC is more strongly associated with obesity than any other cancer[7–9], with an increase of 5 kg/m² in body mass index (BMI) associated with a 60% increase in EC risk[5]. As the prevalence of obesity and metabolic disorders rise globally, the incidence and mortality of EC

increase significantly[7], as do the numbers of young patients and patients with advanced stages[5]. Overweight and obesity account for ~41% of all ECs[7]. Metabolic syndrome (MS) is a series of metabolic disorders, including diabetes, insulin resistance, obesity, hypertension, and hyperlipidemia[10]. MS is not only related to the occurrence of EC; in recent years, research has shown that MS also plays an important role in the progression and poor prognosis of EC beyond traditional mechanisms such as progesterone insensitivity and tumor stemness[11,12]. A prospective study by the University of Calgary showed that MS was associated with worse OS and disease-free survival (DFS) in 540 cases of EC[11]. Jin et al. reported that MS was correlated with worse cancer-specific survival in the early stage of EC by evaluating a total of 10,090 EC patients[13]. Our team also reported that MS was

[1]Department of Obstetrics and Gynecology, Peking University People's Hospital, Beijing, China. [2]Program for Cancer and Cell Biology, Department of Human Anatomy, Histology and Embryology, School of Basic Medical Sciences; Peking University International Cancer Institute; Peking University Health Science Center, Beijing, China. [3]Department of Cell Biology, School of Basic Medical Sciences, Hangzhou Normal University, Hangzhou, China. [4]These authors contributed equally: Lirong Zhai, Yuan Cheng. ✉e-mail: zhanjun@bjmu.edu.cn; wangjianliu@pkuph.edu.cn

closely related to the stage, grade, myometrial invasion (MI), lymph node metastasis (LNM), lymphovascular invasion (LVSI), OS, and recurrence-free survival (RFS) in 506 EC patients[12]. However, the mechanism by which MS promotes the progression of EC remains unclear.

Tumor metabolic reprogramming is a hallmark for tumors. Rearranged metabolism of glucose, lipids, and amino acids serves a dual role: it supplies energy for rapid tumor proliferation and migration, while also providing metabolic intermediates that protect against redox stress and ferroptosis[14]. Metabolites also act as oncometabolites to promote cancer signaling[15] or participate communication networks between organs[16,17]. Even metabolic enzymes exert non-classical functions to participate in immune response, epigenetics and so on[18,19]. Also, metabolic pathways interact with oncogenic signaling to exhibit multiplicity of effects[20,21]. However, the exact mechanism of tumor metabolic reprogramming is uncertain.

Polyamines are small metabolites derived from amino acid catabolism, rich in positive charges and amino groups, including putrescine, spermidine, and spermine. They can bind to DNA and RNA, affect transcription, epigenetic modification[22,23], translation[20], and other processes[24]. Unlike glucose, lipids, and amino acids, which serve as the main "fuel" for tumors, polyamines are important regulatory metabolites. Functioning like the "oil of an engine", polyamines play crucial regulatory roles in multiple physiological and pathological processes within the body. They support cancer metabolism and signal transduction[25,26]. Ornithine decarboxylase 1 (ODC1), Spermidine synthase (SRM), and Spermine synthase (SMS) constitute the main enzymes of polyamine anabolism[24]. Studies have shown that increased ODC1 and polyamines promote various cancer malignancy, including EC[27], by participating in cancer proliferation[28,29], metastasis[30], immune evasion[31–34], and interacting with gut microbiota[24–26], by interacting with multiple oncogenic signaling pathways, including ER, MYC, PTEN-PI3K-mTORC1, WNT signaling, and RAS pathways[25]. The degradation of ODC1 is precisely regulated by polyamine content. Elevated polyamine content leads to transcoded transcription of ornithine decarboxylase antizyme 1 (OAZ1).

HOXB9, a transcription factor of the HOX family, plays important roles in the occurrence or development of many malignant tumors such as breast cancer[35,36], lung cancer[37], colorectal cancer[38], and EC[27]. Its downstream targets include Epithelial-Mesenchymal Transition (EMT) molecules[39], angiogenic molecules[35], DNA repair[40], JMJD6[41], EZH2[42], KRAS[37], E2F1[36,43], and E2F3[27]. Our previous work also revealed a non-transcriptional function of HOXB9: a glucose metabolism receptor that mediates tumor glucose reprogramming[37]. However, HOXB9 has not yet been reported as a sensor for hyperlipidemia.

In this study, we found an increase in polyamine metabolites in the sera and tumor tissues of EC patients with MS (ECWMS). This phenomenon is caused by the stabilization of the HOXB9-ODC1 protein cascade induced by Oleic acid (OA). Accumulation of polyamine metabolites, especially putrescine, further inhibits the degradation of HOXB9. Targeting the feedback loop decreases polyamines and inhibits tumor proliferation and metastasis in vitro and in vivo. The OA-HOXB9-ODC1-polyamine axis explains the key mechanism by which MS promotes the progression of EC.

## Results

### Increased polyamine metabolites are a signature of ECWMS
To clarify the key metabolites involved in the malignant progression of EC caused by MS, we screened and included a total of 62 postmenopausal endometrioid type EC (EEC) patients between 2012 and 2018 (mean age 60 ± 7 years). Endometrioid type is the most common histological type of EC (accounting for ~80%), and its incidence is closely associated with metabolic disorders (such as obesity and insulin resistance). There were 30 cases with MS (mean age 63 ± 6 years) and 32 cases without MS (mean age 56 ± 6 years). Their sera were sent for untargeted metabolomics testing (Fig. 1a). The clinical characteristics of

the identification cohort are detailed in Supplementary Table 1. Principal component analysis (PCA) revealed differences between the two groups (Fig. 1b). A Partial Least Squares Discrimination Analysis (PLS-DA) model identified 25 differential metabolites with Variable Importance for the Projection (VIP) > 1 and p < 0.05, of which 18 were upregulated and 7 were downregulated (Supplementary Table 2). The 15 with the most pronounced differences are shown in Fig. 1c, among which the polyamine pathway metabolites spermine and acetyl putrescine show significant accumulation (Fig. 1c right). Metabolic enzyme association analyses showed significant enrichment in arginine and proline metabolism, serine and methionine metabolism, ferroptosis metabolism, and SMS (Supplementary Fig. 1a). Both arginine metabolism and methionine metabolism are closely related to polyamine metabolism. These results indicated that increased polyamine metabolism is a signature of ECWMS.

Polyamine metabolites are important regulatory metabolites in organisms. Although they are not the direct fuel for cancer, as are the three major nutrients (glucose, lipids, and amino acids), they play a crucial regulatory role in physiological and pathological processes[26]. They are known as the engine oil of cancer and serve as lubricants for various reactions of cancer signals. Although elevated serum spermine or its derivatives have been reported to be associated with malignancy and prognosis in lung, breast, liver, colorectal, and urogenital cancers[44,45], no significant difference was found with pathologic factors in the small sample of this study (Supplementary Fig. 1b), however, spermine and N-acetyl putrescine was positively correlated with BMI (Supplementary Fig. 1c). Furthermore, we validated the results of serum untargeted metabolomics by including EC patients admitted between 2021 and 2022 (Supplementary Table 3). From 210 patients, 156 were screened for inclusion who were eligible to be postmenopausal and whose pathologic type was endometrioid EC (mean age 62 ± 8 years). These 156 cases of sera were sent for targeted polyamine metabolomics (Fig. 1a). The results showed that in patients with MS (Supplementary Table 4), the elevation of polyamine metabolites was statistically significant (Fig. 1d).

Understanding the molecular mechanisms underlying the upregulation of polyamine metabolites can help explain why MS promotes EC. We found that spermidine was elevated in patients with hyperlipidemia only, but showed no difference in patients with diabetes mellitus and hypertension only (Supplementary Fig. 1d), suggesting that hyperlipidemia is a key factor in MS. We analyzed differences in polyamine metabolite levels across clinicopathologic subgroups in these 156 patients and found that putrescine levels significantly differed with MI status in patients with dyslipidemia (p < 0.05), while no significant differences were observed between other metabolites and pathological factors (Fig. 1e; Supplementary Fig. 2a).

We further explore whether serum polyamines were the result of increased polyamine synthesis in tumor tissues. We applied targeted metabolomics to validate polyamine levels in the tumor tissues of EC patients with or without MS in the identification cohort and found increased putrescine in the tumor tissues of ECWMS patients (Fig. 1f). Notably, clinical association analysis of tumor tissue polyamines has revealed that tissue polyamine levels are elevated in association with multiple aggressive pathological factors, including MI, LVSI, higher histological grade, and larger tumor size (Supplementary Fig. 2b, c). Polyamine metabolites are catalyzed by three enzymes: ODC1, SRM, and SMS (Fig. 1g). The elevated levels of these products suggest abnormal expression of the enzymes in this pathway. We conducted western blot (WB) and immunohistochemistry (IHC) to analyze the expression of the three enzymes in 21 tumor tissues randomly selected from patients in the identification cohort. We found that ODC1, not SRM or SMS, was significantly higher in ECWMS (Fig. 1h–k). Therefore, the increase of ODC1, and consequently polyamine metabolites, caused by hyperlipidemia in MS, is an important mechanism in the progression of EC.

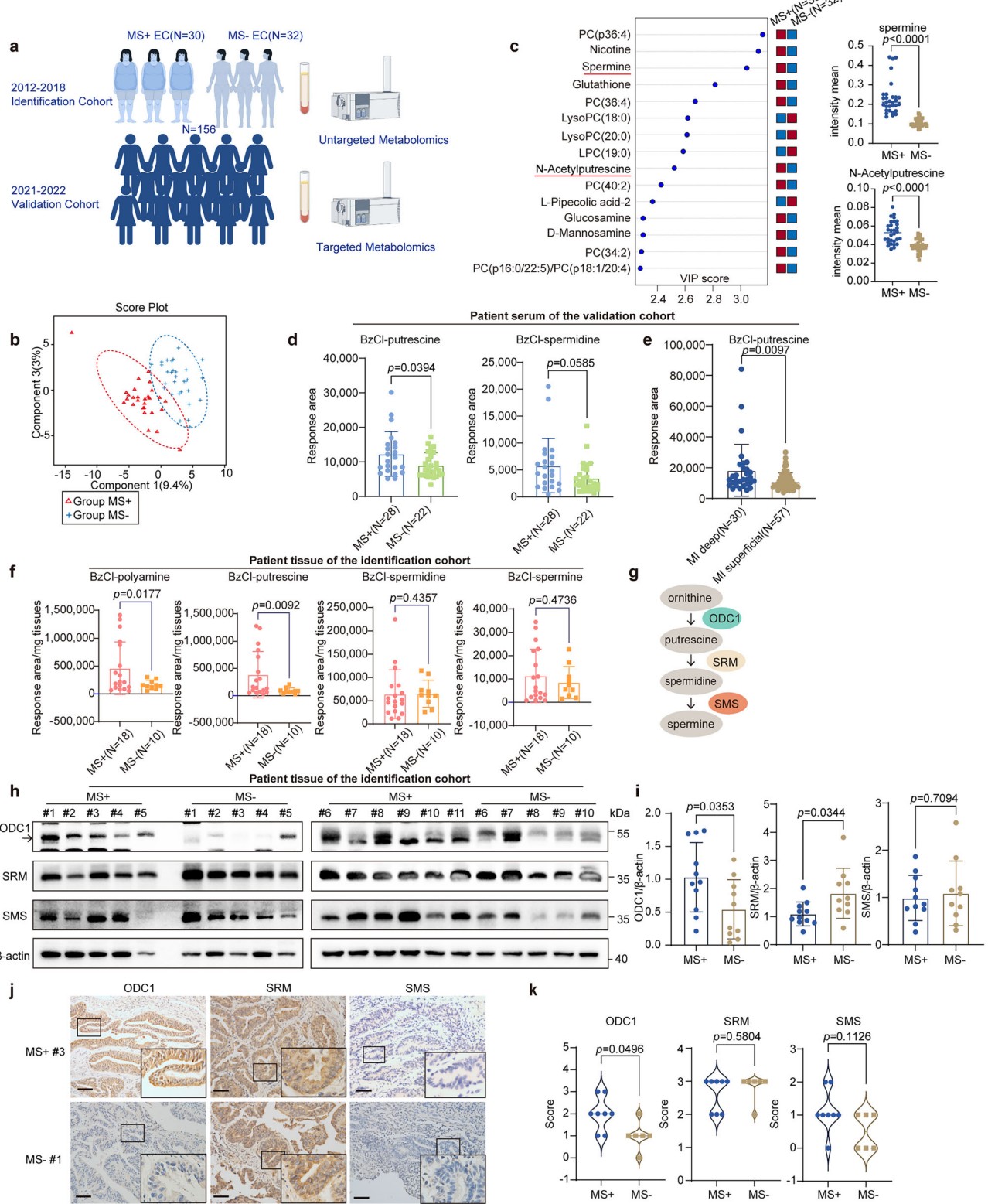

## OA in hyperlipidemia triggers an increase in ODC1 and polyamines

MS comprises a series of metabolic disorders, including obesity, hypertension, diabetes, and hyperlipidemia[10]. Serum targeted metabolomics results from our validation cohort showed that hyperlipidemia, but not diabetes mellitus or hypertension, elevated spermidine (Supplementary Fig. 1d). Additionally, our clinical analysis indicated

that only hyperlipidemia was associated with poorer OS and RFS in EC[12]. This suggests that hyperlipidemia contributes more to the invasiveness and metastasis of EC in MS than abnormal glucose metabolism, hypertension, and obesity. However, the role and mechanism of high lipids for EC are not clear.

We treated Ishikawa and AN3CA cells with common mono/poly-unsaturated fatty acids and short/medium-long chain saturated fatty

**Fig. 1 | Increased polyamine metabolites and ODC1 are signatures of postmenopausal ECWMS. a** Schematic workflow of the study cohort (created with Figdraw; https://www.figdraw.com). **b** PCA of serum untargeted metabolomics in Endometrial cancer (EC) patients with (N = 30)/without Metabolic syndrome (MS) (N = 32; identification cohort). **c** Left: Top 15 metabolites via PLS-DA (VIP > 1); right: colored boxes indicate relative metabolite concentrations (red: upregulated; blue: downregulated); quantification of spermine (two-tailed Mann-Whitney test) and N-acetylputrescine (two-tailed Student's t-test). Data: mean ± SD (MS+: N = 30; MS-: N = 32). **d** Serum Benzoyl chloride (BzCl)-derivatized putrescine/spermidine in EC patients (validation cohort: MS+: N = 28; MS-: N = 22). Data: mean ± SD. Two-tailed Student's t-test (putrescine); two-tailed Mann-Whitney test (spermidine). **e** Serum BzCl-derivatized putrescine in hyperlipidemia subgroup (validation cohort)

stratified by Myometrial invasion (MI) (MI deep: N = 30; MI superficial:57). Data: mean ± SD. Two-tailed Mann-Whitney test. **f** Tumor tissue BzCl-derivatized polyamines in EC patients (identification cohort: MS+: N = 18; MS-: N = 10). Data: mean ± SD. Two-tailed Student's t-test (polyamine, putrescine, spermine); two-tailed Mann-Whitney test (spermidine). **g** Polyamine synthesis pathway: ornithine→putrescine (ODC1), putrescine→spermidine (SRM), spermidine→spermine (SMS). **h** WB of ODC1, SRM, SMS in EC tumor tissues (MS+: N = 11; MS-: N = 10). Samples used different gels (ODC1/SRM; SMS/β-actin), processed in parallel. **i** Grayscale quantification of (**h**). Data: mean ± SD. Two-tailed Student's t-test. **j** IHC of ODC1, SRM, SMS in EC tissues (MS + : N = 8; MS-: N = 6). Scale bars: 50 μm. **k** IHC score quantification of (**j**). Data: mean ± SD. Two-tailed Mann-Whitney test. Source data are provided as a Source Data file.

acids: OA (C18:1, representing 8 carbons and 1 unsaturated bond), arachidonic acid (AA, C20:4), linoleic acid (LOA, C18:2), butyric acid (BA, C4:0), hexanoic acid (HA, C6:0), caprylic acid (CA, C8:0), decanoic acid (DEA, C10:0), lauric acid (LA, C12:0), myristic acid (MA, C14:0), palmitic acid (PA, C16:0), and stearic acid (SA, C18:0). Fatty acid treatment was preceded by a 48-h incubation with delipidated (DL) fetal bovine serum (FBS) to negate the effects of fatty acids from the serum. Additionally, these fatty acids were bound to bovine serum albumin (BSA) since free fatty acids tend to bind to apolipoproteins in the circulatory system. Among the tested fatty acids, only OA significantly stably upregulated ODC1, while the others did not (Fig. 2a). Moreover, only OA stably promoted the wound healing and transwell migration abilities of EC cells across both cell lines (Supplementary Figs. 3a and 3b). High glucose treatment failed to elevate ODC1 levels, in contrast to the effect observed with OA (Supplementary Fig. 3c). Given that ECWMS patients received anti-diabetic treatment, we also detected metformin accumulation in the metabolome (Supplementary Table 2). To rule out metformin's influence, we treated cells with physiological metformin concentrations[46]; results showed that metformin did not upregulate polyamines (Supplementary Fig. 3d). These findings suggest that high lipids, rather than high glucose, are a key upstream factor in upregulating polyamine metabolism.

We found that OA significantly promoted proliferation in EC cells and organoid (Fig. 2b, c) and promoted the motility and migration of EC cells (Fig. 2d, e). Further, we found that OA upregulated polyamine metabolites and ODC1 in a time-dependent manner (Fig. 2f, g; Supplementary Fig. 3e, f). However, due to differences in biological characteristics and malignancy degrees, Ishikawa cells (derived from G1 adenocarcinoma) and AN3CA cells (derived from lymph node metastasis) exhibit distinct metabolic response times to OA. Ishikawa cells demonstrate more significant responses at 12–24 h of OA treatment (Supplementary Fig. 3f), while AN3CA cells show more obvious responses at 36–72 h (Fig. 2f). Therefore, corresponding time points were selected for subsequent experiments based on these results. However, the mRNA level of ODC1 did not increase with OA treatment (Supplementary Fig. 3 g). We conducted a protein half-life experiment, which showed that adding OA for 48 h could extend the half-life of ODC1 from ~30 min to >2 h (Fig. 2h), suggesting that OA increases the level of ODC1 by stabilizing it. The ODC1-polyamines axis appears necessary for OA to promote the malignant phenotype of EC. Knocking down ODC1 blocked the increase of polyamine metabolites caused by OA, and over-expressing ODC1 increased polyamines in cells treated with delipidated serum (Fig. 2i and Supplementary Fig. 3h). Additionally, knocking down ODC1 blocked the migration induced by OA, while supplementing with polyamine metabolites (putrescine, spermidine, and spermine) rescued the migration as well as EMT-promoting transcription factors like Zeb1, and EMT related methyltransferase EZH2 (Fig. 2j, k, l; Supplementary Fig. 3i, j). These results indicate that the increase in ODC1 and subsequent polyamine metabolites is essential for OA to promote EC progression (Fig. 2m). Our study suggests that OA's up-regulation of ODC1 triggers polyamine

accumulation and cancer progression, explaining the mechanism by which OA promotes EC and establishes the chain of OA-ODC1-polyamine-EC progression.

## HOXB9 mediates OA-induced ODC1 stabilization

OA does not affect the binding between ODC1 and OAZ1, which mediates the ubiquitin-independent degradation of ODC1[47] (Supplementary Fig. 4a). The mechanism by which OA influences the stabilization of ODC1 involves HOXB9. We found HOXB9 to be a transcription factor for ODC1 by transcription factor prediction (Supplementary Fig. 4b). Moreover, previous studies on HOXB9 conducted by our group have demonstrated that various interacting proteins of HOXB9 identified through immunoprecipitation-mass spectrometry (IP-MS) participated in key lipid metabolism pathways[37] (Supplementary Fig. 4c), we speculated that HOXB9 might play a role in OA upregulation of ODC1. But HOXB9 surprisingly upregulates ODC1 not through transcriptional regulation but by enhancing protein stability (Fig. 3a, b; Supplementary Fig. 4d). Further investigations revealed that transfection with Flag-HOXB9 prolonged the half-life of ODC1 (Fig. 3c), whereas knocking down HOXB9 with siHOXB9 shortened it (Fig. 3d). The decrease in ODC1 expression due to HOXB9 knockdown could be inhibited by MG132, suggesting that HOXB9 knockdown promotes ODC1 degradation through the proteasome pathway (Fig. 3e). These results demonstrate that HOXB9 stabilizes ODC1.

Exploring further, we observed an enhanced endogenous interaction between ODC1 and HOXB9 in EC cells under OA treatment (Fig. 3f). Furthermore, we found that GFP-HOXB9 stabilized the luciferase signal of Flag-ODC1-luc in a dose-dependent manner, with this effect further potentiated by OA treatment (Supplementary Fig. 4e). Both ODC1 and HOXB9 were successfully pulled down in cell lysates using anti-Flag beads following transfection with Flag-HOXB9 or Flag-ODC1 (Fig. 3g). The interaction was mapped to the 1-184 segments of GST-HOXB9 binding to ODC1, and various segments of ODC1 interact with HOXB9, as demonstrated in GST pull-down assays (Fig. 3h). Confocal analysis showed that HOXB9 and ODC1 are mainly co-localized in the cytoplasm and, to a lesser extent, in the nucleus (Fig. 3i). These results indicate that HOXB9 directly interacts with ODC1. Furthermore, Co-immunoprecipitation (Co-IP) experiments showed that transfection of GFP-HOXB9 inhibited the binding of Flag-ODC1 to OAZ1 and the binding of Flag-OAZ1 to ODC1 (Fig. 3j and Supplementary Fig. 4f). Literature reports indicate that the 1-423 segment of ODC1 is the binding region for OAZ1[47], which coincides with the binding site of HOXB9 (Fig. 3h). This indicates that HOXB9 competes with OAZ1 to bind to ODC1 1-423, thereby inhibiting the degradation of ODC1.

With the addition of OA, both HOXB9 and ODC1 show an increase in the cytoplasm over time (Fig. 3k and Supplementary Fig. 4g). Additionally, in the tumor tissues of EC patients with LVSI or LNM from 2014−2020 (Supplementary Table 5), a significant correlation between HOXB9 and ODC1 (r = 0.56, p = 0.02) is demonstrated by IHC (Fig. 3l), indicating the association between HOXB9 and ODC1 at the protein level. Pathological correlation analysis revealed positive correlation

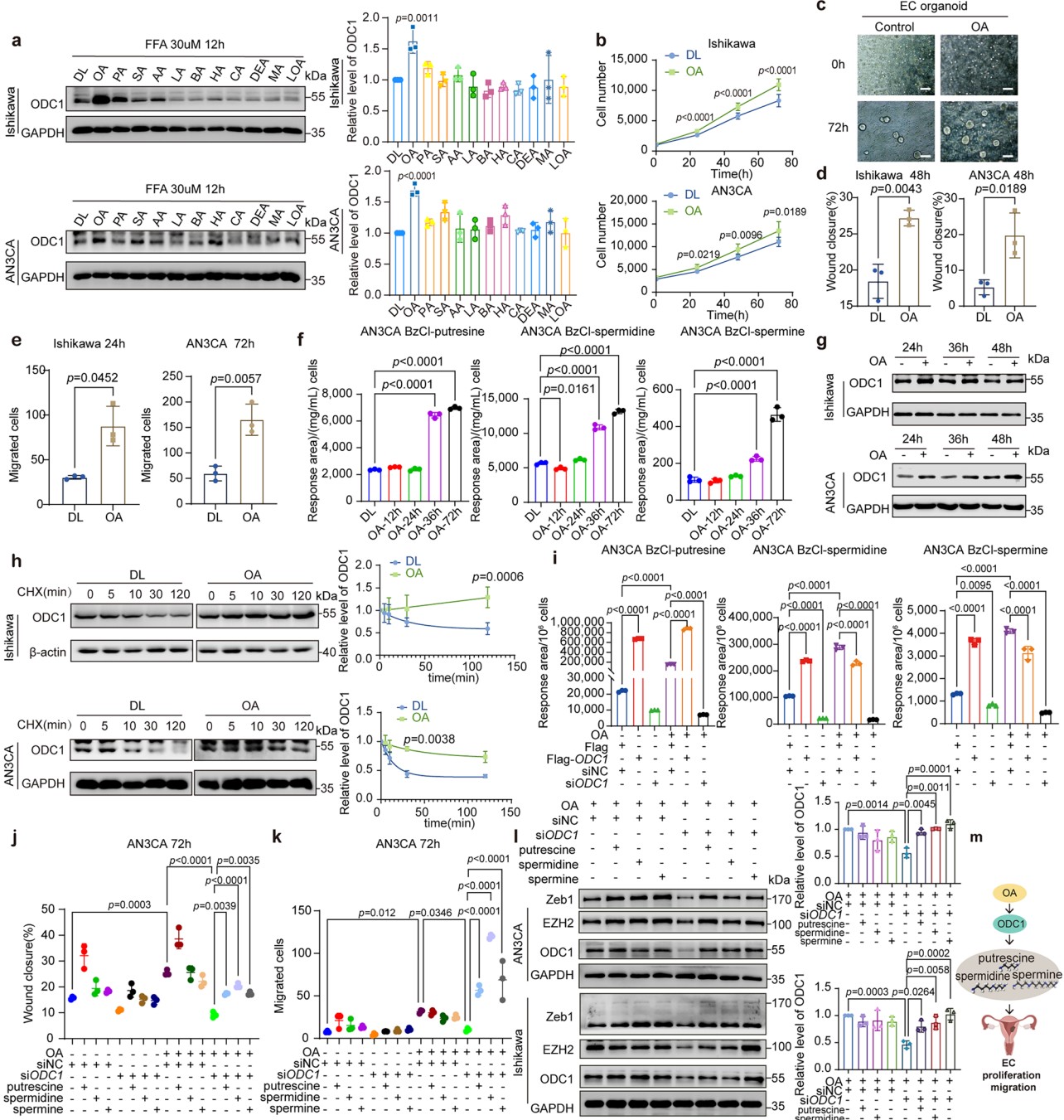

**Fig. 2 | OA in hyperlipidemia triggers an increase in ODC1 and polyamines.**
**a** ODC1 protein in Ishikawa/AN3CA cells treated with 30 μM Free fatty acids (FFAs): delipidated (DL), oleic acid (OA), palmitic acid (PA), stearic acid (SA), arachidonic acid (AA), lauric acid (LA), butyric acid (BA), hexanoic acid (HA), caprylic acid (CA), decanoic acid (DEA), myristic acid (MA), linoleic acid (LOA); right: ODC1 quantification. Data: mean ± SEM (n = 3 independent experiments). Two-tailed one-way ANOVA. **b** Cell growth curves of cells ±OA (30 μM). Data: mean ± SD (n = 10 wells; trend in 3 independent experiments). Two-tailed two-way ANOVA. **c** 30 μM OA effect on EC organoid (n = 3 independent experiments). Scale bars: 50 μm.
**d** Wound healing: Ishikawa/AN3CA cells treated with OA (30 μM) after 48 h. Data: mean ± SEM (n = 3 independent experiments). Two-tailed student's t-test.
**e** Transwell assay: Ishikawa cells treated with OA (30 μM) for 24 h; AN3CA cells for 72 h. Data: mean ± SEM (n = 3 independent experiments). Two-tailed student's t-test. **f** LC-MS of BzCl-polyamines in OA-treated AN3CA at indicated times; normalized to protein concentration. Data: mean ± SD (n = 3 technical replicates; trend detected in 3 independent experiments). Two-tailed one-way ANOVA. **g** ODC1 protein in Ishikawa/AN3CA cells treated with OA (30 μM) at indicated times (n = 3

independent experiments). **h** Left: ODC1 protein in Ishikawa/AN3CA cells ± OA (30 μM, 48 h), +cycloheximide (CHX, 100 μg/mL) at different times. Right: ODC1 degradation curve. Data: mean ± SEM (n = 3 independent experiments). Two-tailed two-way ANOVA. **i** LC-MS of BzCl-polyamines in AN3CA cells ± OA (30 μM, 48 h pre-harvest), transfected with siNC/si*ODC1*, Flag/Flag-*ODC1* for 72 h; normalized to cell number. Data: mean ± SD (n = 3 technical replicates; trend detected in 3 independent experiments). Two-tailed one-way ANOVA. **j, k** Wound healing/Transwell assays for AN3CA cells treated ±OA (30 μM, 48 h), transfected with siNC/si*ODC1*, ±putrescine (50 μM), spermidine (10 μM), spermine (10 μM) for 72 h. Data: mean ± SEM (n = 3 independent experiments). Two-tailed one-way ANOVA. **l** Zeb1/EZH2/ODC1 protein in AN3CA/Ishikawa cells (conditions in (**j**), OA 24 h for Ishikawa/48 h for AN3CA]. Right: ODC1 quantification. Data: mean ± SEM (n = 3 independent experiments). Two-tailed one-way ANOVA. Same-experiment samples: different gels (Zeb1, ODC1, GAPDH; EZH2, GAPDH), processed in parallel.
**m** Working model: OA stabilizes ODC1 to upregulate polyamines, promoting EC proliferation/migration. Created in BioRender. Zhai, L. (2025) https://BioRender. com/e31mxj9. Multiple comparisons were corrected. Source data are provided.

between ODC1 expression and grade (r = 0.59, p = 0.02) (Fig. 3l). The correlation between HOXB9 and ODC1 was further validated by multiplex immunofluorescence in EC tissue microarray (TMA) (r² = 0.34, p < 0.0001) (Fig. 3m). The patients' characteristics of TMA were summarized in Supplementary Table 6. These results suggest that the role of OA in up-regulating ODC1 stability may be mediated through its interaction with HOXB9.

Moreover, either knocking down or overexpressing HOXB9 can inhibit or promote, respectively, the upregulation of polyamine metabolites induced by OA (Fig. 3n and Supplementary Fig. 4h), further confirming that HOXB9 serves as a critical link in the upregulation of ODC1-polyamine metabolism by OA (Fig. 3o). Furthermore, overexpression of ODC1 increases HOXB9 levels, while knocking down ODC1 reduces them (Supplementary Fig. 4i). Our previous work has shown that Praja2 is an E3 ubiquitin ligase for HOXB9, which binds to Praja2 and is degraded through a ubiquitin-dependent proteasome pathway[37]. Transfection with Flag-ODC1 inhibits the interaction between HOXB9 and Praja2 (Supplementary Fig. 4j), suggesting a positive feedback regulation between HOXB9 and ODC1.

### HOXB9 is required for OA-induced accumulation of ODC1 and EC progression

HOXB9 is essential for OA-induced ODC1 accumulation and EC progression. Since HOXB9 inhibits the binding of ODC1 to OAZ1, thereby stabilizing ODC1, it is necessary for OA's upregulation of polyamine metabolites and the promotion of a malignant phenotype in EC cells. WB showed that knocking down HOXB9 blocked the OA-induced upregulation of ODC1, which could be rescued by overexpressing ODC1 (Fig. 4a and Supplementary Fig. 5a, indicated in blue box). Additionally, knockdown of HOXB9 inhibited the OA-induced prolongation of the half-life of ODC1 (Fig. 4b and Supplementary Fig. 5b); overexpression of HOXB9 prolonged the ODC1 half-life in either OA or control group (Supplementary Fig. 5b, c), suggesting that HOXB9 plays a crucial role in stabilizing ODC1 by OA.

Furthermore, transfecting cells with siHOXB9 inhibited OA-promoted migration, but this effect could be rescued by Flag-ODC1. Overexpression of Flag-HOXB9 and simultaneous knockdown of ODC1 also rescued the OA-induced phenotype blocked by ODC1 knockdown (Fig. 4c, d, indicated in blue box). Metabolite analysis showed that OA resulted in an upregulation of polyamines. Knocking down HOXB9 or ODC1 inhibited the increase in polyamines induced by OA, which could be restored by overexpressing ODC1 (Fig. 4e, Supplementary Fig. 5d, indicated in blue box). Together, these findings indicate that the OA-HOXB9-ODC1-polyamine metabolite axis regulates the malignant phenotype of EC cells.

### OA directly interacts with and stabilizes HOXB9

Given that OA's stabilization of ODC1 is mediated by HOXB9, it raises the question of how OA upregulates HOXB9. Notably, OA does not significantly upregulate the mRNA level of HOXB9 over time (Supplementary Fig. 6a), which suggests that OA's effect on HOXB9 may be due to stabilization. Through protein half-life experiments, we demonstrated that OA can prolong the half-life of HOXB9 (Fig. 5a). Co-IP experiments revealed that the addition of OA inhibited the interaction between HOXB9 and Praja2, its E3 ubiquitin ligase[37], and the simultaneous addition of putrescine and OA further inhibited this interaction, resulting in decreased ubiquitination of HOXB9 (Fig. 5b). This result indicates that OA and HOXB9 not only enhance polyamine metabolism, but polyamine metabolites also feedback to influence the upstream mechanism—in addition to their classic metabolic functions, these metabolites can also act as signaling molecules.

OA promotes intracellular lipid droplet generation, while knockdown of ODC1 inhibits OA-induced lipid droplet formation; however, supplementation with polyamines rescued the phenotype (Fig. 5c, d, Supplementary Fig. 6b, c). The isolation of proteins to the surface of

lipid droplets may control their involvement in processes at other cellular locations[48,49], and lipid droplets may promote the aggregation of protein molecules, potentially forming phase separation regions[50]. We speculate that the stabilizing effect of OA on HOXB9 and ODC1 could be through direct binding of lipid droplets with HOXB9 and ODC1, or even through encapsulation, aggregation, and enhancement of their interaction, thereby inhibiting their degradation. Using HIS-SIM super-resolution microscopy, we found that lipid droplets indeed co-localize with HOXB9 and ODC1 at multiple z-axis levels (Fig. 5e). Co-IP confirmed increased interaction between ODC1 and HOXB9 after the addition of OA (Figs. 3f and 5f). Surface plasmon resonance (SPR) assays indicated that, under exogenous conditions, OA can bind to GST-HOXB9 and GST-ODC1 at very low concentrations, with the dissociation constant (KD) lower than that of the GST control alone (Fig. 5g), suggesting a possible direct interaction between them. Themo Shift Assay confirmed that the thermal stability of HOXB9 and ODC1 increased after the addition of OA (Fig. 5h), indicating that OA may directly bind with HOXB9 and ODC1, promoting their stability. Molecular docking using the CB-DOCK2 platform suggested that OA and HOXB9 may have docking domains (Supplementary Fig. 6d). Therefore, the interaction between OA and HOXB9 and, further, between HOXB9 and ODC1, upregulates downstream polyamines and promotes EC progression (Fig. 5i).

### Knocking down HOXB9 and ODC1 inhibits tumor formation and LNM in high-fat diet (HFD)-induced EC mice models

We used EC subcutaneous xenograft and footpad injection LNM models in ovariectomized, HFD-induced Balb/c nude mice to simulate postmenopausal ECWMS (Fig. 6a). The weight of the HFD group was higher than that of the low-fat diet (LFD) group (p < 0.05) in both the subcutaneous xenograft and LNM models (Supplementary Fig. 7a, b). Blood lipid tests indicated an increase in total cholesterol (TC) in the HFD-fed mice in subcutaneous model and LNM model (Fig. 6b and Supplementary Fig. 7c). MRI demonstrated significantly more visceral fat in the renal capsular area of the HFD group and more whole-body adipose tissue (Fig. 6c, d).

In the subcutaneous xenograft tumor model, HFD promoted in situ tumor growth, whereas knockdown of ODC1 or HOXB9 attenuated this effect (Fig. 6e–g). WB and qPCR confirmed the knockdown efficiency of HOXB9 and ODC1 in Ishikawa cells (Supplementary Fig. 7d) and xenograft tissue lysates. However, the effect of high lipids in the HFD group may compromise this knockdown efficiency in the tissue lysates (Supplementary Fig. 7e). IHC showed increased expression of HOXB9, ODC1, and Ki67 in the HFD group (Supplementary Fig. 7f). Knocking down HOXB9 or ODC1 weakened the expression of Ki67 (Supplementary Fig. 7f). Lipid droplet staining indicated more lipid droplets in the tumor tissues of the HFD group, while knocking down HOXB9 or ODC1 reduced lipid droplet formation (Supplementary Fig. 7f). Targeted metabolomics analysis of polyamine metabolites in mouse sera showed a significant increase in the HFD group, confirming the successful simulation of the model of postmenopausal ECWMS patients (Fig. 6h, i and Supplementary Fig. 7g). Moreover, knockdown of HOXB9 and ODC1 could inhibit the increase of polyamine metabolites in sera induced by HFD (Fig. 6i, j and Supplementary Fig. 7g). We further measured the levels of polyamines and OA in subcutaneous xenograft tumor tissues and found that tumor size was closely correlated with putrescine content in the tissues (Fig. 6j, k and Supplementary Fig. 7h, i). Polyamine levels were increased in xenograft tumors from HFD-fed mice, whereas ODC1 knockdown led to a reduction in polyamines (Fig. 6j and Supplementary Fig. 7h).

Repeating the subcutaneous tumor model with sgRNA-mediated HOXB9 and ODC1 knockout cells yielded similar results. Based on CRISPR-Cas9 technology, we designed sgRNA sequences targeting different exons of HOXB9 and ODC1 (Supplementary Fig. 7j), and selected HOXB9-exon1 and ODC1-exon4 for subsequent experiments.

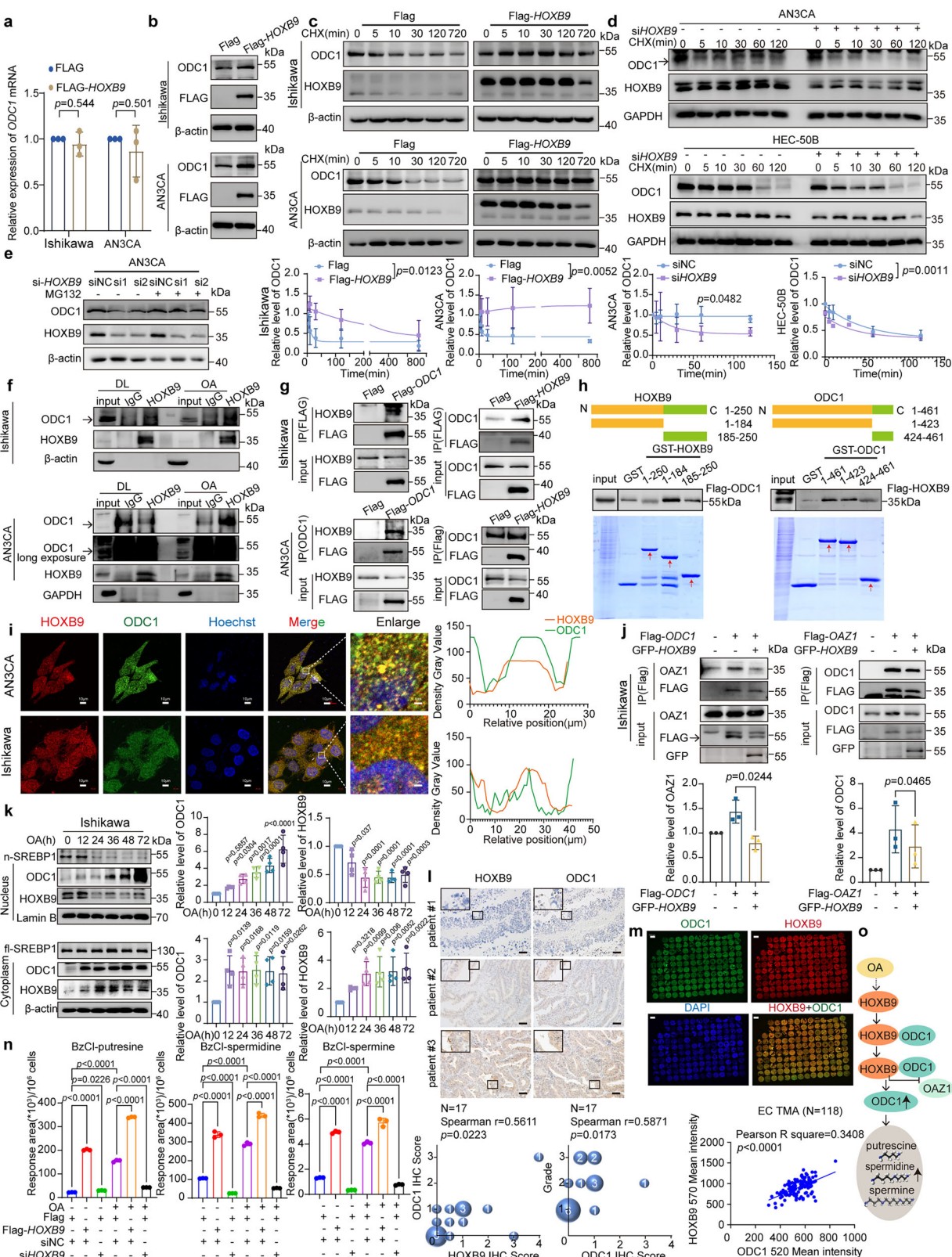

Ishikawa cells stably transfected with sgRNAs targeting HOXB9 exon 1 and ODC1 exon 4 were used to establish a subcutaneous xenograft tumor model (Supplementary Fig. 8a). After HFD feeding, the HFD group had increased adipose tissue (Supplementary Fig. 8b, c) and body weight (Supplementary Fig. 8d). Knockout of either HOXB9 or ODC1 suppressed HFD-induced subcutaneous tumor growth. Additionally, adding difluoromethylornithine (DFMO, an ODC1 inhibitor; 1%

w/v) to the drinking water also effectively inhibited HFD-induced xenograft growth (Fig. 6l, Supplementary Fig. 8e–f). WB and qPCR of xenograft lysates verified the knockout efficiency (Supplementary Fig. 8g, h).

In the footpad injection LNM metastasis model, ~3 months after injection, the metastatic signal was significantly stronger in the HFD group compared to the LFD group (Fig. 6m, n). This signal was reduced

**Fig. 3 | HOXB9 mediates OA induced ODC1 stabilization. a** *ODC1* mRNA in Ishikawa/AN3CA (Flag/Flag-*HOXB9*). Data: mean ± SEM (n = 3 independent experiments). Two-tailed t-test. **b** ODC1 protein in Ishikawa/AN3CA (Flag/Flag-*HOXB9*; n = 3 independent experiments). Same-experiment samples: different gels (ODC1/Flag; β-actin), parallel. **c** Upper: ODC1 in Ishikawa/AN3CA (Flag/Flag-*HOXB9*) + CHX (100 µg/mL, different times). Lower: ODC1 degradation curve. Data: mean ± SEM (n = 3 independent experiments). Two-tailed two-way ANOVA. Same-experiment samples: different gels (ODC1; HOXB9/β-actin), parallel. **d** Upper: ODC1 in AN3CA/HEC-50B (siNC/si*HOXB9*) + CHX (100 µg/mL, different times). Lower: ODC1 degradation curve. Data: mean ± SEM (n = 3 independent experiments). Two-tailed two-way ANOVA. Same-experiment samples: different gels (HOXB9/β-actin; ODC1/GAPDH), parallel. **e** AN3CA (siNC/si-1*HOXB9*/si-2*HOXB9* ± MG132; n = 3 independent experiments). Same-experiment samples: different gels (ODC1; HOXB9/β-actin), parallel. **f** Co-IP: Ishikawa (24 h OA) /AN3CA (48 h OA, 30 µM) + anti-HOXB9, WB (anti-ODC1) (n = 3 independent experiments). Same-experiment samples: different gels (ODC1/HOXB9; HOXB9/β-actin; ODC1/GAPDH), parallel. **g** Left: Co-IP: Ishikawa/AN3CA (Flag/Flag-*ODC1*) + anti-Flag/ anti-ODC1, WB (anti-HOXB9) (n = 3 independent experiments). Right: Co-IP: Cells (Flag/Flag-*HOXB9*) + anti-Flag, WB (anti-ODC1) (n = 3 independent experiments). **h** Top: Schematic of GST-HOXB9/ODC1 domains. Lower: Bacterial purified GST/GST-HOXB9/GST-ODC1 + 293 T (Flag-*ODC1*/Flag-*HOXB9*), WB (anti-Flag) (n = 3 independent experiments). Gels: Coomassie-Blue staining. **i** Triple immunofluorescence of Ishikawa/AN3CA (anti-HOXB9 [red], anti-ODC1 [green], Hoechst [blue]; n = 3 independent experiments). White boxes: co-localization. Scale bars: 10 µm; 1 µm (enlarged). Right: Co-localization quantitation. **j** Co-IP: Ishikawa (GFP-*HOXB9*/Flag-*ODC1*) + anti-Flag, WB (anti-OAZ1); or (GFP-*HOXB9*/Flag-*OAZ1*) + anti-Flag, WB (anti-ODC1). Lower: OAZ1/ODC1 quantification. Data: mean ± SEM (n = 3 independent experiments). Paired two-tailed t-test. Same-experiment samples: different gels (OAZ1/Flag; GFP), parallel. **k** HOXB9/ODC1/SREBP1 in Ishikawa cytoplasmic/nuclear lysates (OA, 30 µM, indicated times) + ODC1/HOXB9 quantification. Data: mean ± SEM (n = 3 independent experiments). Two-tailed one-way ANOVA. Same-experiment samples: different gels (n-SREBP1/ODC1; HOXB9/Lamin B; fl-SREBP1/ODC1; HOXB9/β-actin), parallel. **l** Upper: IHC of ODC1/HOXB9 in EC tissues (N = 17 with LVSI/LNM, 3 shown). Scale bars:50 µm. Lower: Two-tailed Spearman correlations. **m** Multiplex immunofluorescence of HOXB9/ODC1/DAPI in EC Tissue microarray (TMA) (118 cancer; 17 peritumoral excluded). Scale bars: 2000 µm. Two-tailed Pearson correlation. **n** BzCl-polyamines in AN3CA ( ± OA 30 µM, siNC/si*HOXB9*/Flag/Flag-*HOXB9*). Data: mean ± SD (n = 3 technical replicates; trend in 3 independent experiments). Two-tailed one-way ANOVA. **o** Model: HOXB9 competes with OAZ1 to bind ODC1, inhibiting ODC1 degradation, causing polyamine accumulation. Multiple comparisons corrected. Source data provided.

by knockdown of HOXB9 or ODC1, or by adding DFMO (1% w/v) to drinking water (Fig. 6m, n). WB and qPCR confirmed the knockdown efficiency of HOXB9 and ODC1 in the AN3CA cell line (derived from EC LNM lesions) used in this model (Supplementary Fig. 8i). Additionally, these interventions inhibited the HFD-induced elevation of polyamine metabolites (Supplementary Fig. 8j). Dissection showed minimal in-situ carcinoma growth at the footpad site in mice, yet the sub-iliac lymph nodes were significantly enlarged (Supplementary Fig. 8k). This differs from other tumor cell lines, which usually lead to popliteal lymph node enlargement after footpad injection[51]. This phenomenon might be associated with the origin of AN3CA. In the HFD group, lymph nodes were significantly larger than those in the LFD group (Supplementary Fig. 8l). Hematoxylin and eosin (H&E) staining, along with IHC using anti-keratin and anti-CK18 antibodies, indicated that HFD promoted LNM. Conversely, knockdown of ODC1 or HOXB9, or adding DFMO (1% w/v) to the drinking water, inhibited HFD-induced LNM (Supplementary Fig. 8m).

Therefore, knocking down HOXB9 or ODC1, or adding DFMO to drinking water effectively inhibits LNM as well as in-situ tumor growth, indicating a promising target for reversing of metabolic disorder promoted EC progress.

## OA-HOXB9-ODC1 axis in ECWMS

We performed multiplex immunofluorescence staining on paraffin-embedded tumor tissues from randomly selected patients in the identification cohort. This was to investigate the existence of the OA-HOXB9/Praja2-ODC1/OAZ1-polyamine axis in in-situ tissue cells. The results showed that in ECWMS patients, both the mean intensity of HOXB9 and ODC1 and their co-localization were increased (Fig. 7a, b). Additionally, we employed Raman spectroscopy[52], a non-destructive method for in-situ analysis of C-H bonds and tissue metabolic information, to quantitatively analyze the lipid profile in frozen sections from MS+ and MS- samples (Fig. 7c, Supplementary Fig. 9a, Supplementary Table 7). PCA and cluster map demonstrated that the samples from the two groups were distinct and well-differentiated (Fig. 7d, Supplementary Fig. 9b). We observed significant differences in lipid composition between the two groups (Supplementary Fig. 9c), with an increase in the 1057 peak representing total lipids and an increase in lipids containing C-C (peaks 1128) CH2 and CH (peaks 2879) in ECWMS patients (Fig. 7e, f), with statistically significant differences (p < 0.05). Further, we constructed a derivatized method to analyze OA by liquid chromatography-mass spectrometry (LC-MS) and found that OA was also significantly elevated in the tumor tissues of ECWMS (Fig. 7c, g).

We performed survival analysis using profiles of *ODC1*, *HOXB9*, and BMI, combined with survival data of EC patients retrieved from The Cancer Genome Atlas (TCGA) database. In patients with obesity (BMI ≥ 30 and <35), high *ODC1* expression was associated with poorer OS compared to low *ODC1* expression (HR = 3.32; p = 0.008) (Supplementary Fig. 9d), while high *HOXB9* expression was linked to worse OS than low HOXB9 expression (HR = 3.23; p = 0.007) (Supplementary Fig. 9e), indicating that high *HOXB9* and high *ODC1* expression in patients with obesity are associated with poor prognosis.

No statistically significant survival difference was observed between patients with obesity and normal-weight patients with high *ODC1* expression (Supplementary Fig. 9f). However, among patients with high *HOXB9* expression, patients with obesity had a worse prognosis than normal-weight patients (HR = 1.87; p = 0.04) (Supplementary Fig. 9g), suggesting that obesity is associated with poor prognosis.

Excitingly, we applied DFMO to patient-derived tumor cells (PTCs)[53] of EC. These PTCs are similar to EC tissue, as are organoids, and contain components such as epithelial cells and fibroblasts (Fig. 7h, i). A patient with stage IA grade 2 EC experienced recurrence 2 years after endocrine therapy, with comorbid hyperlipidemia and insulin resistance. MRI revealed diffuse lesions in the uterine cavity, and IHC showed elevated ODC1 and HOXB9 expression (Fig. 7i). Results demonstrated that DFMO could effectively enhance the sensitivity of PTCs to first-line chemotherapeutic agents such as paclitaxel and carboplatin (Fig. 7j). Similarly, DFMO were observed to enhance chemotherapy sensitivity to varying degrees in PTCs from other EC patients with potential primary chemotherapy resistance (Supplementary Fig. 10, Supplementary Table 8). These results strongly support that targeting ODC1 may serve as a therapeutic strategy for progressive, drug-resistant, or relapsed EC patients with concurrent metabolic disorders.

## Discussion

This study demonstrated that increased polyamine metabolism is associated with MS-promoted EC progression. ODC1, the key enzyme in polyamine metabolism, was identified as the most critical enzyme for the aberrant enrichment of polyamine metabolites (Fig. 1). Consequently, the link between lipid metabolism and ODC1-polyamine metabolism emerged as a significant scientific question. We identified OA (Fig. 2), a relatively common and widely studied monounsaturated fatty acid, as a stabilizer of ODC1 through the pro-cancer transcription factor HOXB9. OA protects HOXB9 from ubiquitination degradation (Fig. 5). Subsequently, HOXB9 interacts with ODC1, preventing its degradation and leading to the accumulation of polyamines, thus

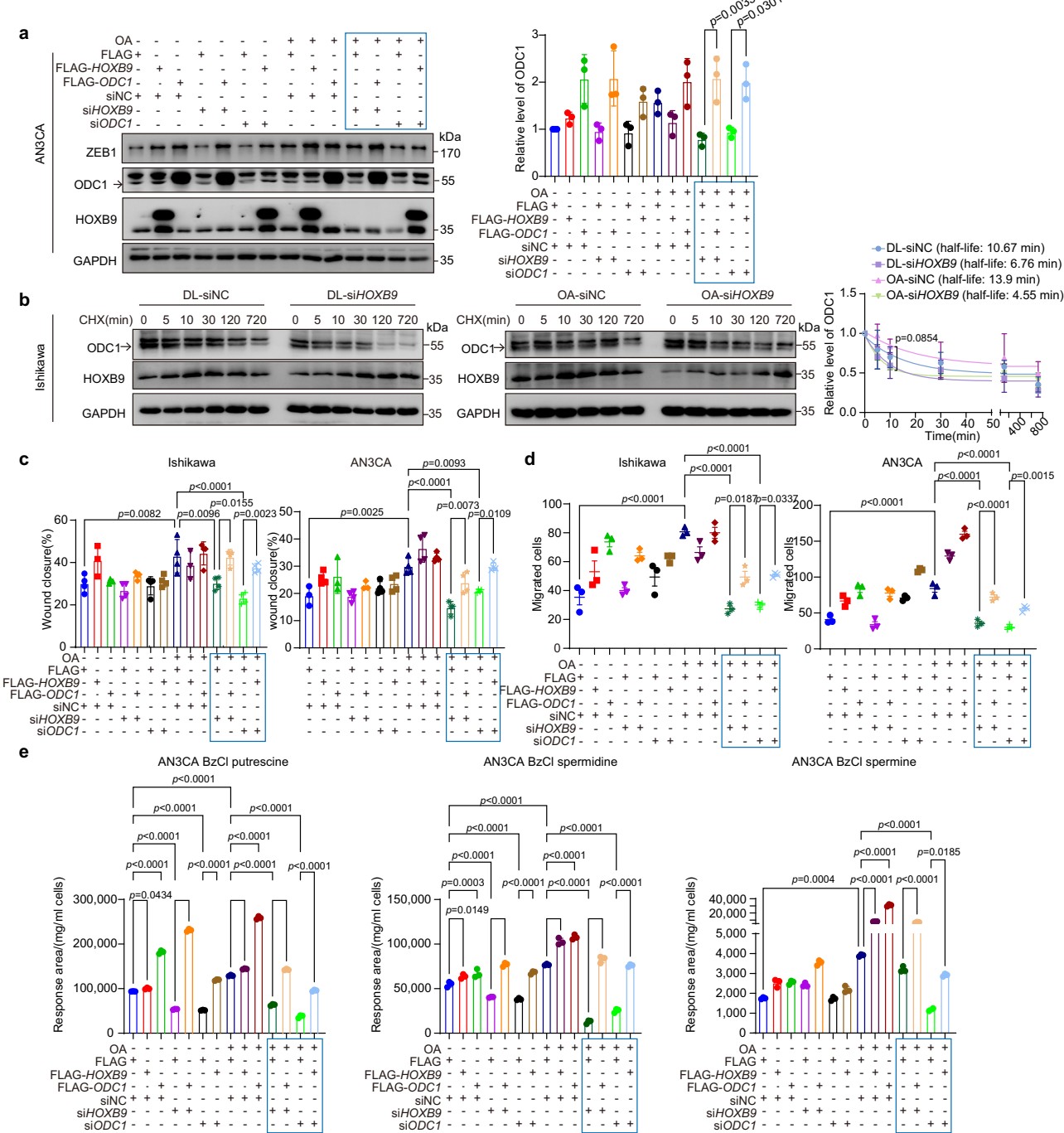

**Fig. 4 | HOXB9 is required for OA-induced accumulation of ODC1 and EC progression. a** WB of ODC1, HOXB9, ZEB1 in AN3CA cells with/without OA (30 μM), transfected with siNC, si*HOXB9*, si*ODC1*, Flag, Flag-*HOXB9*, Flag-*ODC1* (72 h; OA added 48 h). Right: ODC1 quantification. Data: mean ± SEM (n = 3 independent experiments). Two-tailed one-way ANOVA. Key rescue groups boxed: si*ODC1*+Flag-*HOXB9*, si*HOXB9*+Flag-*ODC1*. Same-experiment samples: different gels (Zeb1/ODC1/GAPDH; HOXB9), parallel processed. **b** WB of ODC1 in Ishikawa cells with/without OA (30 μM), transfected with siNC/si*HOXB9* (48 h) + CHX (100 μg/mL, different times). Right: ODC1 degradation curve. Data: mean ± SEM (n = 3 independent experiments). Two-tailed two-way ANOVA. Same-experiment samples: different gels (Zeb1/ODC1/GAPDH; HOXB9), parallel processed. **c** Wound healing of Ishikawa/AN3CA cells under conditions in (**a**) (Ishikawa: OA 24 h; AN3CA: OA 48 h). Data: mean ± SEM (n = 3 independent experiments). Two-tailed one-way ANOVA. Key rescue groups boxed. **d** Transwell of cells under conditions in (**a**) (Ishikawa: OA 24 h; AN3CA: OA 48 h). Data: mean ± SEM (n = 3 independent experiments). Two-tailed one-way ANOVA. Key rescue groups boxed. **e** LC-MS of BzCl-derivatized-polyamines in AN3CA cells under conditions in (**a**). Data: mean ± SD (n = 3 technical replicates; trend in 3 independent experiments). Two-tailed one-way ANOVA. Key rescue groups boxed. Multiple comparisons were corrected. Source data are provided as a Source Data file.

promoting cancer progression (Figs. 3 and 4). Targeting ODC1 and HOXB9 inhibits tumor formation and LNM in xenograft models (Fig. 6). Additionally, the ODC1 inhibitor can effectively enhance the chemotherapy sensitivity of PTCs (Fig. 7 and Supplementary Fig. 10), highlighting its potential clinical application in MS-related progressive or resistant EC. This reveals the mechanism by which the OA-HOXB9-ODC1-polyamine metabolism axis mediates EC progression due to MS (Fig. 8).

Previous studies have primarily addressed the role of MS in EC from an environmental perspective, focusing on factors such as

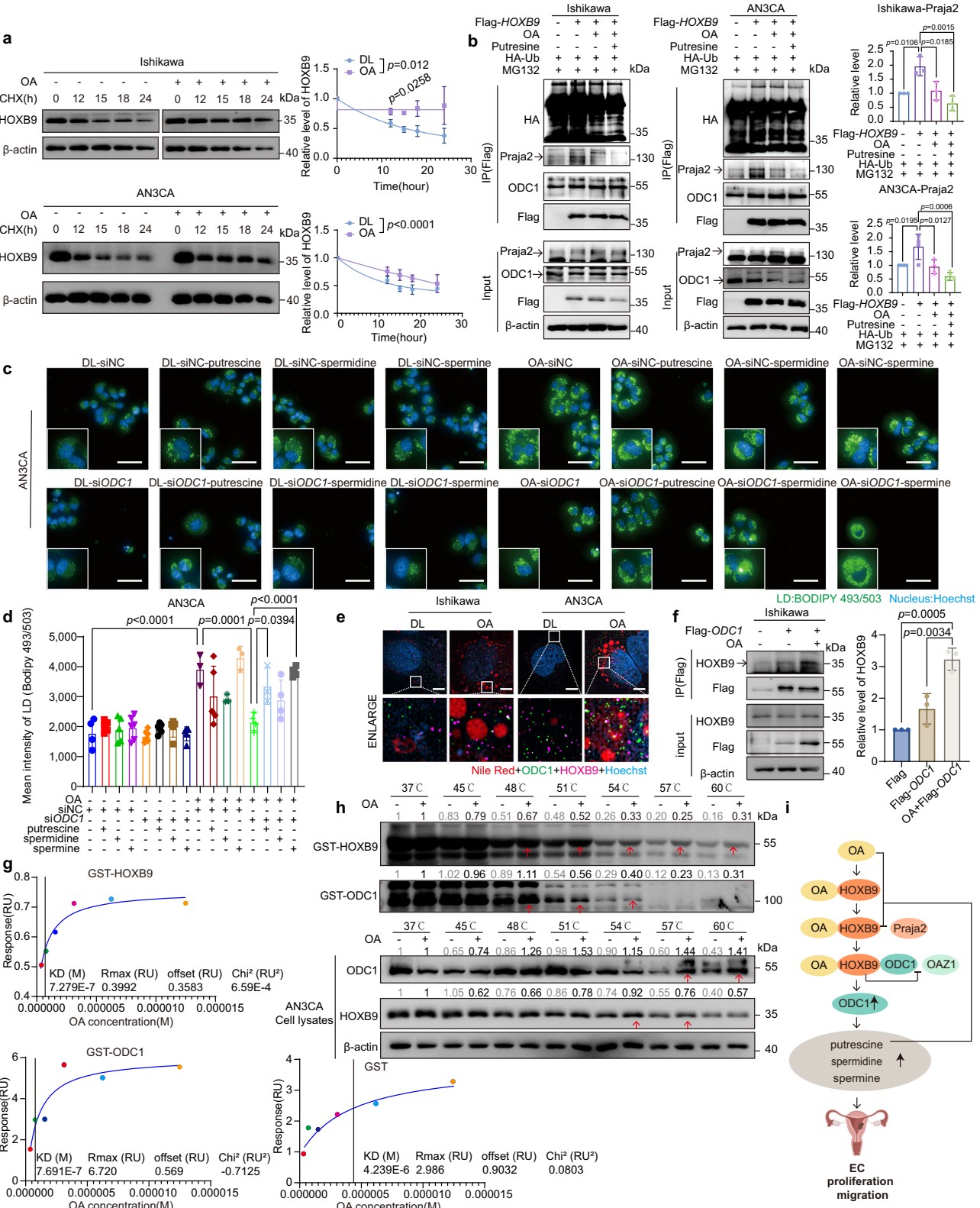

inflammation, adipokines, insulin, and estrogen[7,54]. Some research has suggested a link between EC and lipid reprogramming, though the underlying mechanisms remain unclear. Trousil et al. reported that choline phospholipid accumulation in EC tumor tissues was due to increased CHKα activity, as shown by HMR analysis, which led to the transcription of several pro-proliferative genes[55]. Dai et al. noted that the combination of estrogen, leptin, and insulin in EC patients with obesity could activate the AKT pathway, phosphorylate ACLY for nuclear translocation, and up-regulate pyrimidine synthesis, thus promoting EC proliferation[56]. However, these studies lacked depth in exploring the role and mechanisms of choline phospholipid accumulation in EC and did not individually screen for significant factors in obesity. In this study, we approached the problem from the angle of metabolic reprogramming in tumor cells, examining the interplay

**Fig. 5 | OA directly interacts with and stabilizes HOXB9. a** WB of HOXB9 in Ishikawa and AN3CA cells treated with/without OA (30 μM, 48 h), adding CHX (100 μg/mL) at different times; right: HOXB9 degradation curve. Data: mean ± SEM (n = 3 independent experiments). Two-tailed two-way ANOVA. **b** Co-IP: Ishikawa/AN3CA cells transfected with Flag+HA-*Ub*, Flag-*HOXB9* + HA-*Ub* (with/without OA 30 μM, with/without putrescine 50 μM) for 48 h, and treated with MG132 6 h before harvest; cell lysates were then incubated with anti-Flag M2 beads, blotted with anti-Praja2/ODC1. Praja2 blot grayscale: mean ± SEM (≥ 3 independent experiments). Two-tailed one-way ANOVA. Same-experiment samples: different gels (Praja2, ODC1, Flag; Flag, β-actin; HA), parallelled processed. **c** Representative Bodipy 493/503 staining of lipid droplets in AN3CA cells transfected with siNC/si*ODC1*, with/without OA 30 μM (48 h), with/without putrescine (50 μM), spermidine(10 μM), spermine (10 μM) for 72 h (n = 3 independent experiments). Scale bars: 100 μm. **d** Statistical analysis of mean lipid droplet intensity in (**c**): Data: mean ± SD (3–6 fields; trend observed in 3 independent experiments). Two-tailed one-way ANOVA. **e** Triple immunofluorescence of Ishikawa/AN3CA cells treated with/without 30 μM OA (24/48 h): anti-HOXB9 (violet), anti-ODC1 (green), lipid droplets (Nile red), Hoechst (blue); visualized by HIS-SIM (n = 3 independent experiments). Scale bars: 5 μm. **f** Co-IP: Ishikawa cells transfected with Flag, Flag-*ODC1* with/without OA (30 μM, 48 h); incubated with anti-Flag beads, blotted with anti-HOXB9. HOXB9 grayscale: mean ± SEM ((n = 3 independent experiments). Two-tailed one-way ANOVA. Same-experiment samples: different gels (Flag, HOXB9; Flag, β-actin), parallelled processed. **g** SPR assay of GST/GST-HOXB9/GST-ODC1 and OA; corrected response curves shown. **h** Thermal shift assay of HOXB9/ODC1 stability with/without 30 μM OA: GST-HOXB9/GST-ODC1 (upper) or AN3CA lysates (lower) (n = 3 independent experiments). Same-experiment samples: different gels (ODC1; HOXB9, β-actin), parallelled processed. **i** Working model: OA directly binds HOXB9, stabilizing it by preventing interaction with E3 ligase Praja2; HOXB9 interacts with ODC1, leading to polyamine accumulation and EC progression. Created in BioRender. Zhai, L. (2025) https://BioRender.com/e31mxj9. Multiple comparisons were corrected. Source data are provided.

between metabolites and oncogenic signals. We identified elevated polyamine metabolites and pinpointed ODC1 as the key enzyme. We also established OA as a crucial upstream regulator among various fatty acids. Thus, we have clarified the specific mechanism and unveiled the crosstalk between lipid metabolism and polyamine metabolism, which mediates the progression of EC caused by MS.

We found that polyamine metabolites were upregulated in the sera and tumor tissues of postmenopausal ECWMS (Fig. 1). Serum polyamine levels exhibit a positive correlation with BMI. In patients diagnosed with hyperlipidemia, serum polyamine levels also differ significantly with MI status (Fig. 1). Subsequent analysis of polyamine levels in patient tumor tissues revealed associations between tissue polyamine levels and MI, LVSI, recurrence, and tumor size (Supplementary Fig. 2). These findings suggest that the measurement of serum and tissue polyamine levels may serve as a valuable biomarker for disease assessment and prognosis prediction. Furthermore, the development of real-time polyamine tissue imaging techniques or simple reagent kits can be explored to improve the accuracy of disease assessment.

We validated key enzymes in the polyamine pathway and found an increase in ODC1 in tumor tissues of patients with MS (Fig. 1). However, the interaction between lipid metabolism and polyamine metabolism remains unclear. Glucose, lipid, and amino acid metabolisms are interrelated and continuously intersecting. Numerous studies have explored the correlation between lipid and polyamine metabolism. For instance, levels of putrescine, spermidine, and spermine have been reported to decrease in serum samples of patients with extremely obesity after bariatric surgery[57]. Activation of the polyamine-degrading rate-limiting enzyme SSAT has been shown to improve glucose and lipid metabolism[58,59]. Chemical activation of SSAT by TETA corrects diet-induced MS in mice[60]. Even ODC1 heterozygous knockout mice show significant reductions in body mass and fat content[61]. Newest research found that refeeding after starvation promotes intestinal stemness and tumorigenesis by boosting polyamine production[62]. While these studies support our conclusions, they do not provide a specific mechanism by which lipid metabolism regulates polyamine metabolism. We identified OA as playing a key role in upregulating ODC1 and polyamines, leading to EC proliferation and migration (Fig. 2). We propose the OA-HOXB9-ODC1-polyamine interaction as the main mechanism through which MS regulates EC, presenting a unique mechanism compared to the previously reported non-canonical role of ACLY[56].

EC is a metabolism-related tumor strongly associated with obesity, making it an exemplary case and an important reference for studying the interaction between tumor metabolism reprogramming and the systemic metabolic environment. The OA-HOXB9-ODC1-polyamine pathway was significantly enhanced in ECWMS patients. Whether this phenomenon is specific to EC warrants further investigation, and its applicability to other metabolism-related tumors also

remains to be explored. If this molecular mechanism is not significant in other types of tumors, it highlights that our findings provide a genuine reason as to why EC has a higher risk with increasing BMI compared to other tumor types.

Numerous studies have reported that OA promotes cancer malignancy[63–65]. In melanoma, OA upregulates the proportion of unsaturated fatty acids in the membrane through ACSL3, which inhibits oxidative stress and ferroptosis, thereby enhancing tumor cell survival and metastasis to lymph nodes[66]. However, there is a contrary report stating that OA (200 μM) inhibited EC cell proliferation via the PTEN/AKT/mTOR pathway[67]. To reconcile these findings, we offer the following explanations. First, we used 30 μM OA, a concentration closer to physiological levels[68,69]. Fatty acids in the tumor microenvironment are typically derived from lipoprotein catabolism and adipose tissue breakdown. Higher doses of fatty acids can be lipotoxic[70,71]. Second, FBS used in cell culture contains essential fatty acids necessary for cell growth. To observe the effects of fatty acids rather than lipotoxicity, it is preferable to pretreat cells with delipidated FBS[72] or serum-free conditions[73] to remove the effects of pre-existing fatty acids. In this study, we revealed the mechanism by which OA promotes EC cell proliferation and motility through the ODC1-polyamine metabolism axis.

This study reveals the OA-HOXB9-ODC1 axis by promoting protein stability through a series of molecular interactions. Without HOXB9, OA could not stabilize ODC1 (Fig. 4), making HOXB9 a crucial mediator for OA to stabilize ODC1. We further demonstrated that OA directly regulates and interacts with HOXB9 (Fig. 5). Moreover, OA facilitated the interaction between endogenous HOXB9 and ODC1 (Fig. 3). This finding implies that OA might form a phase-separated complex with HOXB9 and ODC1, where these components are mutually stabilizing and essential to one another. Specifically, OA promotes the interaction between endogenous HOXB9 and ODC1, thereby stabilizing this interaction. Subsequently, OA further upregulates HOXB9, and the upregulated HOXB9 in turn acts to stabilize ODC1. As a result, despite differences in their dynamics and phases, HOXB9 and ODC1 interact and mutually stabilize each other.

HOXB9 is not only a pro-cancer transcription factor but also a central player in lipid and polyamine metabolism, highlighting the non-transcriptional regulatory function of HOXB9. In comparison with our previous finding that HOXB9 senses glucose metabolism in the cytoplasm[37], this finding suggests that HOXB9 also acts as a receptor of fatty acid metabolism in tumors. Both glucose and lipid metabolism regulate HOXB9 at the level of degradation; however, the mechanisms differ. During glucose metabolism, glucose deprivation promotes the phosphorylation of HOXB9 leading to its degradation by the ubiquitination pathway, while high glucose levels inhibit the degradation of HOXB9, increasing its transcriptional activity. Conversely, HOXB9 stabilized by OA exhibits non-transcriptional effects by binding to ODC1 and inhibiting its degradation.

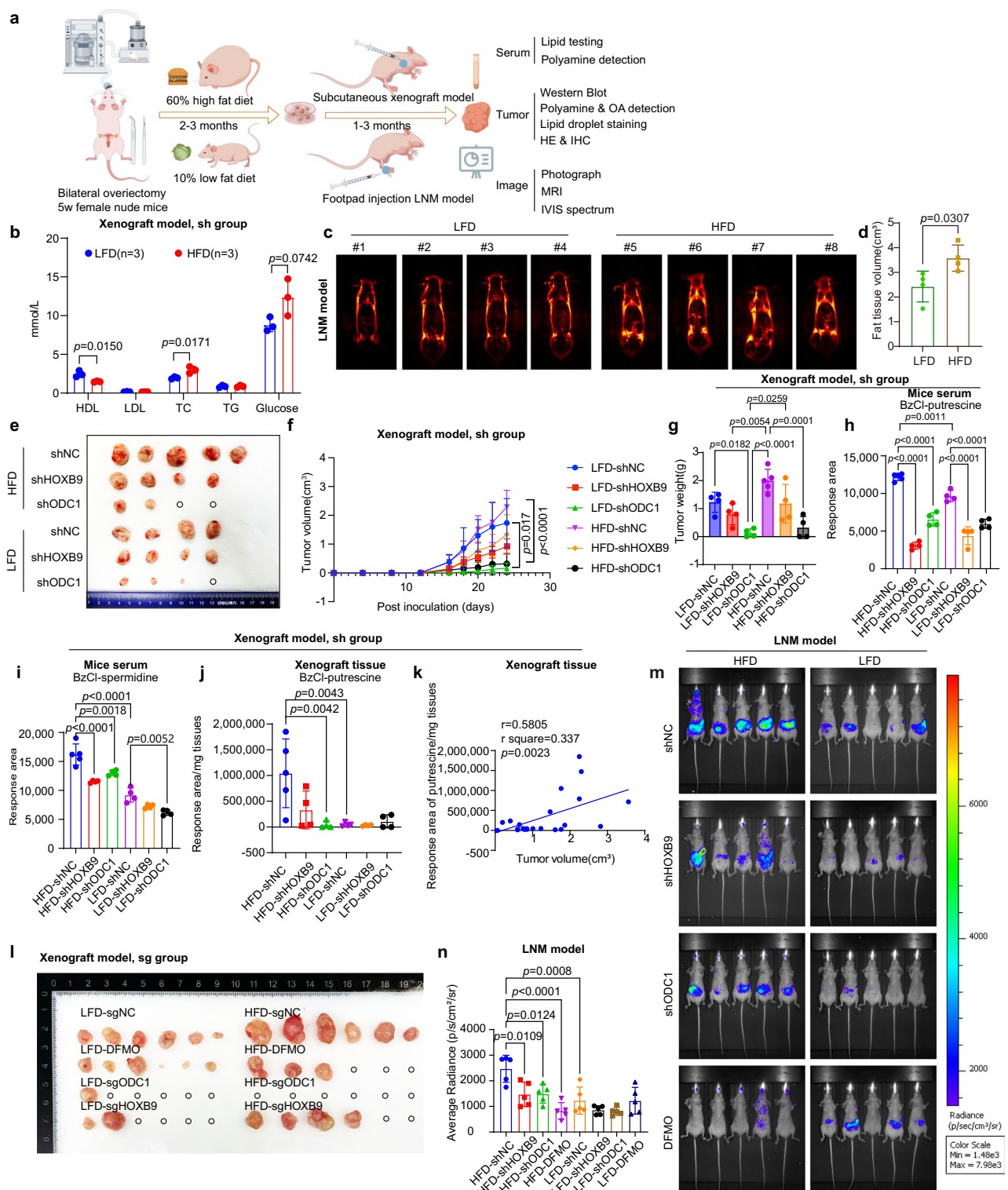

The polyamine metabolite putrescine further inhibited HOXB9's binding to Praja2 (Fig. 5), suggesting a positive feedback regulation of polyamines, acting not only as metabolites but also as signaling molecules. This long-term positive feedback in a continuous daily cycle has a more profound impact on HOXB9, ODC1, polyamine accumulation, and tumor progression than mere upregulation. In addition to our study, Monelli et al. found that elevated polyamines in vascular endothelial cells promote fatty acid catabolism in adipocytes via paracrine secretion, leading to elevated free fatty acids in the microenvironment[74]. Choi, etc. reported that ODC1 upregulates KLF2 to inhibit PPAR and suppress fatty acid catabolism in hepatocarcinoma cells[75]. These results indicated that accumulated polyamines may promote adipocyte lipolysis and free fatty acid accumulation in tumor microenvironment; the upregulation of ODC1 could further inhibit fatty acid catabolism and promote lipid droplet formation in tumor cells, creating a probable positive feedback loop.

In this study, we administered the ODC1 inhibitor, DFMO, to mice and PTCs and achieved promising results (Figs. 6 and 7, Supplementary Fig. 10). Future efforts could explore combining DFMO, or other small molecules that inhibit ODC1, with chemotherapeutic agents,

**Fig. 6 | Knocking down HOXB9 and ODC1 inhibits tumor formation and LNM in HFD-induced EC mice models. a** Schematic workflow of the animal experiment (created with Figdraw; https://www.figdraw.com/). **b** Serum levels of total trigly-cerides (TG), total cholesterol (TC), glucose, HDL, and LDL in subcutaneous tumor model mice (sh group) after 2-month diet induction [high-fat-diet (HFD): n = 3, low-fat-diet (LFD): n = 3)]. Data: mean ± SD; two-tailed student's t-test. **c** Representative MRI images of renal capsular fat tissue from lymph node metastasis (LNM) model mice (HFD: n = 4, LFD: n = 4) after 2-month diet induction. **d** Quantitative statistics of whole-body adipose tissue volume in MRI-scanned mice from (**c**). Data: mean ± SD (HFD: n = 4, LFD: n = 4); two-tailed student's t-test. **e** $3 \times 10^6$ Ishikawa cells (stable expressing shNC/shHOXB9/shODC1) were inoculated into the axilla of ovar-iectomized nude mice. Xenograft tumors were dissected and photographed 24 days post-inoculation. **f** Tumor growth curves (subcutaneous model, sh group: HFD-NC n = 5, others n = 4 per group). Data: mean ± SD; two-tailed one-way ANOVA. **g** Xenograft tumor weights [same group sizes as (**f**)]. Data: mean ± SD; two-tailed one-way ANOVA. **h** Serum BzCl-derivatized putrescine in subcutaneous model [same group sizes as (**f**)], measured by LC-MS. Data: mean ± SD; two-tailed one-way

ANOVA. **i** Serum BzCl-derivatized spermidine in subcutaneous model [same group sizes as (**f**)], measured by LC-MS. Data: mean ± SD; two-tailed one-way ANOVA. **j** BzCl-derivatized putrescine in xenograft tissues [same group sizes as (**f**)], mea-sured by LC-MS (0 for no tumor). Data: mean ± SD; two-tailed one-way ANOVA. **k** Correlation between tissue putrescine response area and tumor volume; two-tailed Pearson correlation. **l** $3 \times 10^6$ Ishikawa cells (stable expressing sgNC/sgHOXB9-exon1/sgODC1-exon4) were inoculated into ovariectomized nude mice. sgNC + DFMO group: 1% (w/v) DFMO in drinking water when tumors were palpable. Tumors dissected/photographed 30 days post-inoculation: n = 6 per group. **m** $5 \times 10^6$ luciferase-tagged AN3CA cells (stable expressing shNC/shHOXB9/shODC1) were inoculated into footpad of ovariectomized mice. shNC+DFMO group: 1% (w/v) DFMO in drinking water 2 months post-inoculation. Luciferase signals measured by IVIS Spectrum 3 months post-inoculation; n = 5 per group. **n** Quantification of lymph node metastasis signals [Average Radiance (photons per second per square centimeter per steradian (p/s/cm²/sr)] from (**m**). Data: mean ± SD; two-tailed one-way ANOVA; n = 5 per group. Multiple comparisons were corrected. Source data are provided.

immunotherapeutic agents, anti-angiogenic agents, and mTOR inhi-bitors for treating EC patients in advanced stages with recurrence and metastasis.

## Limitation

In animal experiments, conditions such as hyperglycemia, hyperlipi-demia, and obesity cannot be completely isolated for individual stu-dies, which limits the controllability of single-condition variables in vivo experiments. In our study, we differentiated the various con-ditions of hyperlipidemia through cellular experiments, which proved effective in identifying key oncogenic factors associated with MS.

## Methods

This study complied with all relevant ethical regulations and was approved by the Ethics Committee of Peking University People's Hospital (Ethics approval number: 2022PHB372-001), the Animal Eth-ics Committee of Peking University People's Hospital (Ethics approval number: 2020PHE094), and the IACUC of the Animal Experimental Center of Peking University Health Science Center (Ethics approval numbers: BCJF0170).

### Clinical samples and study approval

All human participants were postmenopausal females with EC (sex self-reported, as EC is female-specific; sex was integrated into the study design given the disease's sex restriction). All patients provided written informed consent; no participant remuneration was applicable. No clinical trials were conducted.

A total of 62 patients eligible for postmenopausal, combined with MS (30) or without MS (32) between 2012 and 2018 were included, and their sera were sent for untargeted metabolomics. The diagnostic criteria for MS referred to the CDS 2013 version set by the Diabetes Branch of the Chinese Medical Association: ①Abdominal obesity: waist circumference >85 cm in women; ② Hyperglycemia: Fasting blood glucose ≥6.1 mmol/L or 2-h postprandial blood glucose >7.8 mmol/L and/or those who have been diagnosed with diabetes mellitus; ③Hypertension: Blood pressure ≥130/85 mmHg and/or those who have been diagnosed with hypertension; ④Hyperlipidemia: Fasting TG ≥ 1.70 mmol/L; or Fasting HDL-C < 1.04 mmol/L; Meeting more than 3 items. In our study, patients were enrolled if they met all of the four standards. Sera from 156 post-menopausal patients with endometrioid EC, diagnosed between 2021 and 2022, were selected for targeted polyamine metabolomics analysis. Additionally, tissue sections from 17 EC patients with pathological diagnoses of LVSI or LNM between 2014 and 2020 were subjected to IHC. The clinicopathological details of these patients are presented in Supplementary Tables 1, 3–5. TMA samples were procured from Zhuohao Biotech Co., Ltd. (Endometrial Cancer EMC1351), and the corresponding clinical and pathological information is provided in Supplementary Table 6. Moreover, 6 frozen

tissue sections were obtained from patients who underwent inpatient surgery between January 2024 and December 2024, selected ran-domly, and subjected to Raman spectroscopy. The relevant clinical information is presented in Supplementary Table 7. Additionally, information on 5 patients from whom PTCs were obtained for drug sensitivity tests—selected randomly from those who underwent inpa-tient surgery between June 2024 and June 2025—is shown in Supple-mentary Table 8.

### Cell culture and organoid culture

The human EC cell lines Ishikawa (European Collection of Authen-ticated Cell Cultures, ECACC; Cat. No. 99040201), AN3CA (Amer-ican Type Culture Collection, ATCC; Cat. No. HTB-111™) and HEC-50B (Japanese Collection of Research Bioresources, JCRB; Cat. No. JCRB1145), and the human embryonic kidney cell line 293T (ATCC; Cat. No. CRL-3216™) were purchased from ECACC, ATCC and JCRB, respectively. Ishikawa was maintained in DMEM/F12 (Cat# CM10092, Macgene) supplemented with 10% FBS, while AN3CA was maintained in MEM (Cat# CM50011, Macgene) supplemented with 10% FBS. HEC-50B was maintained in MEM (Cat# CM50011, Mac-gene) supplemented with 15% FBS. 293T was maintained in high glucose DMEM (Cat# CM50011, Macgene) supplemented with 10% FBS. 2 mM glutamine, 100 units/mL penicillin, and 0.1 mg/mL streptomycin were added to the culture medium. Cells were incu-bated at 37 °C in a humidified chamber containing 5% CO² and passaged with 0.25% trypsin/0.02% EDTA to dissociate at 80% confluence.

Before undergoing fatty acid treatment, cells were cultured in a medium containing 10% delipidated FBS (Cat# AB-FBS-DL-0500, ABW) for 48 h to eliminate the influence of fatty acids in normal FBS[72]. Then, various BSA-linked fatty acids made according to Caroline CT Trem-blay's protocol[76] were added to the culture medium with 10% delipi-dated FBS and cultured for 12–72 h. All control groups were treated with BSA-sodium salt solvents and marked as delipidated (DL) in the figures.

The organoids were established from EC patients' tissue from Peking University People's Hospital. The organoids were maintained in Endometrial organoid amplification medium (Cat# M204, Chuangxin Keyan Ltd) mixed with Matrigel® Matrix (Cat# 356234, Corning) (1:1.3, v/v). They were incubated at 37 °C in a humidified chamber containing 5% CO² and passaged with TrypLE™ Express Enzyme (Cat# 12604013, Gibco).

### PTCs culture, immunofluorescence staining, drug test

PTCs were established and used for drug sensitization experiments by the group of XI Jianzhong from the School of Future Technology, Peking University[53]. Fresh tissues treated with ice-cold PBS (10 mM HEPES, 100 U/mL penicillin-streptomycin), minced, digested (1 mM

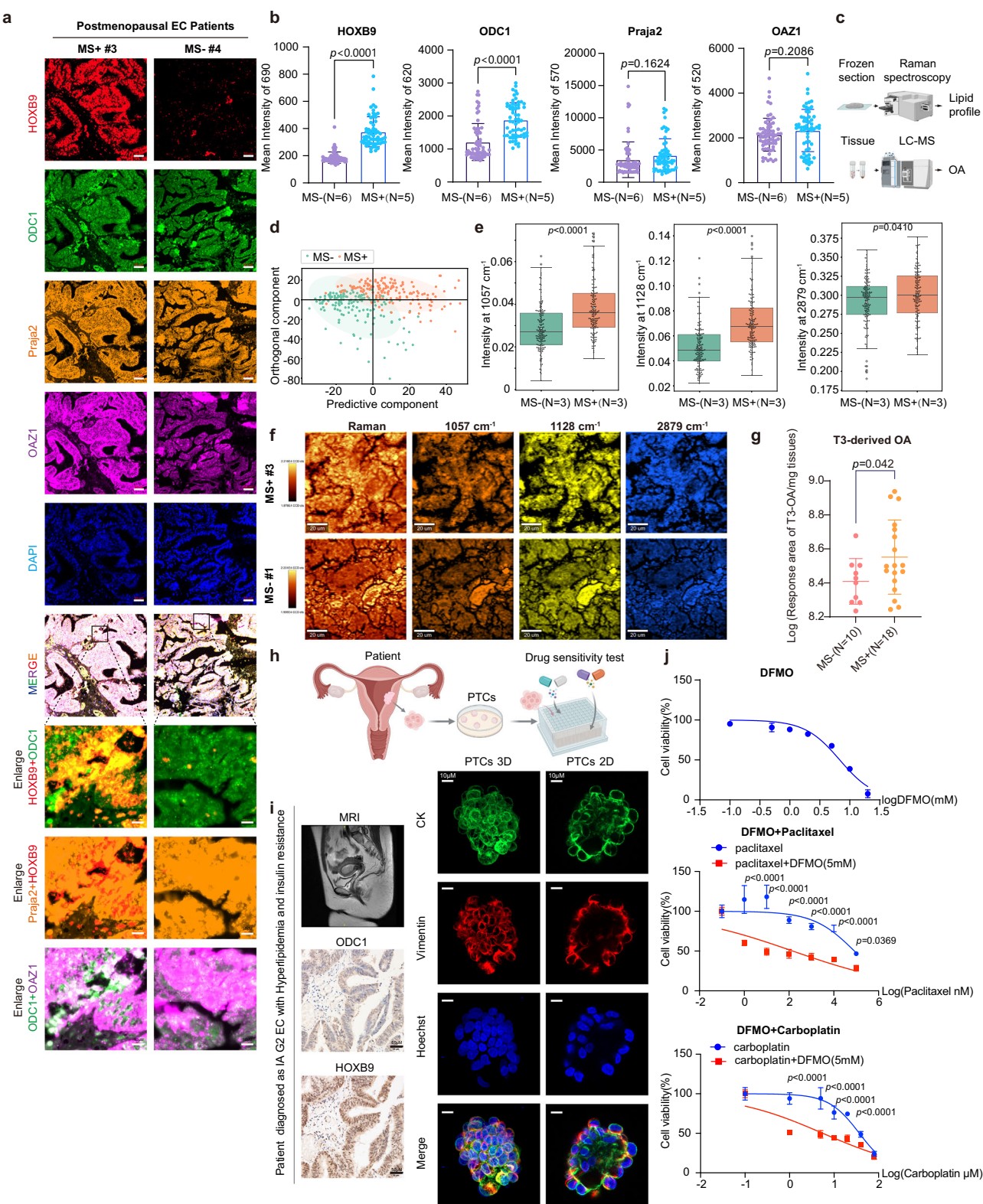

**Nature Communications** | (2026)17:388

PBS/EDTA + 200 U/mL collagenase I; 1 h), centrifuged (300 × *g*, 4 °C, 10 min), resuspended in Advanced DMEM (40 ng/mL EGF, Peprotech, AF-100-15; 20 ng/mL bFGF, Peprotech, 100-AF-18B; 10 μM Y-27632-07, GeneX Health, GX-C-07; 30 ng/mL HGF, PeproTech, 100-39; 10 nM β-estradiol, Sigma-Aldrich, E2578), seeded ($10^5$ cells/cm²) (37 °C, 5% $CO_2$; medium refreshed 2–3-day).

Immunofluorescence staining: Fixed (4% Paraformaldehyde, 20 min), washed (M2 medium, Sigma-Aldrich, M7167; 0.5% BSA),

permeabilized (1% Triton X-100), blocked (10% goat serum), stained (primary antibodies; Alexa488/Alexa549-conjugated secondary antibodies or fluorescent-conjugated primary antibodies; Hoechst 33342), imaged (Zeiss LSM980).

Drug tests: PTCs (>40 μm) collected, centrifuged, seeded (Teflon-modified chips, GeneX Health, GX-01; 30-50 PTCs), 50 μL drug-containing medium added, incubated, imaged (Nikon Ti-U micro-scope system). Viability via well and PTC areas[53].

**Fig. 7 | OA-HOXB9/Praja2-ODC1/OAZ1 axis exists in ECWMS. a** Representative multiplex immunofluorescence staining of postmenopausal EC samples (with/without MS): HOXB9 (red), Praja2 (yellow), ODC1 (green), OAZ1 (purple), DAPI (blue). Scale bars: 50 μm (overview); 10 μm (enlarged). **b** Statistical analysis of mean intensity for ODC1, OAZ1, HOXB9, Praja2 (MS + : N = 5 paraffin sections, 66 fields; MS-: N = 6 sections, 61 fields). Data: mean ± SEM; two-tailed student's t-test. **c** Schematic workflow: Raman spectroscopy (frozen sections) and LC-MS (tissue homogenates) for lipid profiles/OA content (Created in BioRender. Zhai, L. (2025) https://BioRender.com/o0nnrq8). **d** PCA of Raman spectroscopy data from EC samples (MS+: N = 3 patients; MS-: N = 3 patients). **e** Quantification of lipid peak shifts (Raman spectroscopy) in EC samples (MS+: N = 3 patients, 150 points; MS-: N = 3 patients, 150 points); two-tailed Mann-Whitney U test. Box plot: Center line: median (50th percentile); box limits: 25th to 75th percentiles; whiskers extend to 1.5 × interquartile range (IQR) from the box; mean: white circle marker; individual data points are displayed as black swarm points. **f** Raman spectroscopy imaging of distinct lipid peaks in EC samples (with/without MS). Scale bars: 20 μm. **g** Levels of log transformed T3 (2,4-bis(diethylamino)−6-hydrazino-1,3,5-triazine)-derivatized OA in EC tumor tissues (MS+: N = 18; MS-: N = 10, identification cohort). Data: mean ± SD; two-tailed Mann-Whitney test. **h** Schematic of Patient -derived tumor cells (PTCs) culture. Created in BioRender. Zhai, L. (2025) https://BioRender.com/h96j420. **i** PTCs from a reproductive-aged EC patient (stage IA Grade 2, hyperlipidemia, insulin resistance, recurrent from endocrine therapy). Left: Patient's sagittal pelvic MRI; representative IHC (ODC1, HOXB9) of the patient's tissues (Scale bars: 50 μm). Right: Representative 3D/2D confocal images of the patient's PTCs (CK [green], vimentin [red], Hoechst [blue]; Scale bars: 10 μm). **j** PTCs viability after 7-day treatment (different concentrations of DFMO, paclitaxel, carboplatin, or DFMO 5 mM co-administration). Data: mean ± SD (n = 3 replicate wells); two-tailed two-way ANOVA. 5 cases total; PTC results of the remaining 4 patients are in Supplementary figures. Multiple comparisons were corrected. Source data are provided as a Source Data file.

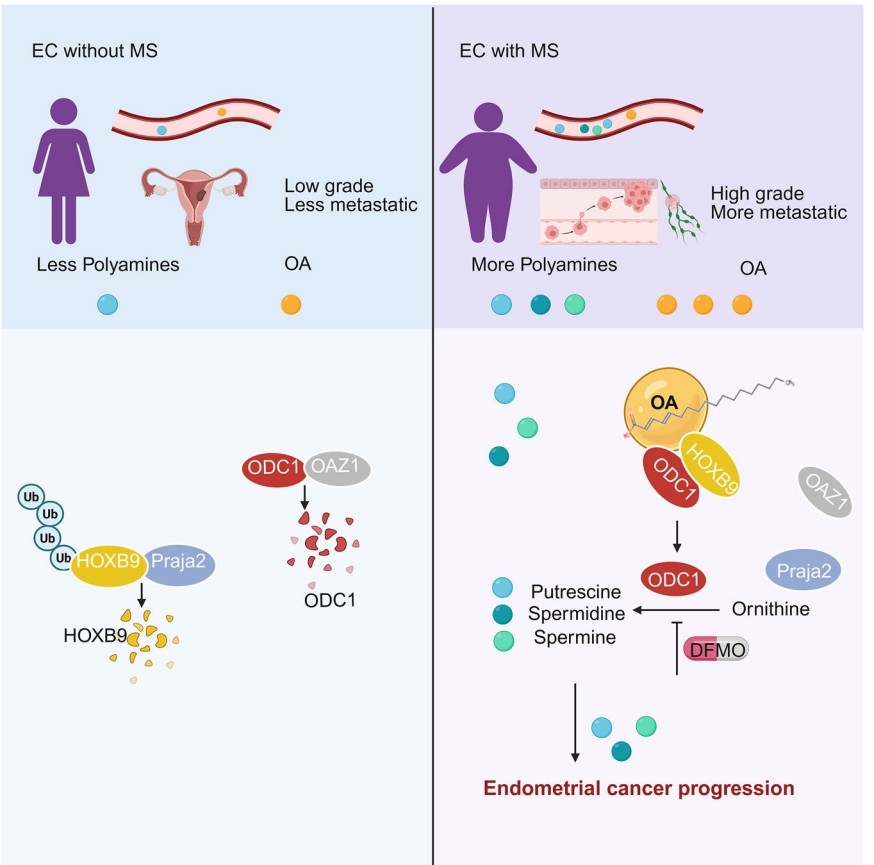

**Fig. 8 | Mechanism diagram: OA stabilizes HOXB9 to upregulate ODC1 and promote polyamine accumulation, thereby facilitating ECWMS.** In patients with ECWMS, the elevation of serum polyamines is attributed to the local increase of OA. OA interacts with HOXB9, which inhibits the binding of HOXB9 to the E3 ubiquitin ligase Praja2, thereby stabilizing HOXB9 and preventing its degradation. HOXB9 further interacts with ODC1, inhibiting the binding of ODC1 to OAZ1 and thus suppressing the degradation of ODC1. The increased ODC1 generates putrescine, spermidine, and spermine, which promote tumor progression and manifest as elevated serum polyamines. Created in BioRender. Zhai, L. (2025) https://BioRender.com/o82i414.

## siRNA transfection

All RNAi oligonucleotides were purchased from Beijing Tsingke Biotechnology Co., Ltd. Fusion cells grown to 70% −80% within 24 h of passaging were transfected with siRNA using lipofectamine 3000 (Cat# L3000015, Invitrogen) according to the manufacturer's instructions. Samples were collected 48 to 72 h after transfection for subsequent analysis. siRNA sequences used in this study were si*ODC1*: sense CCUCCAGAGAGGAUUAUCUAU; si*HOXB9*: #1 sense TGTCGAC TCGATCATAAGT; #2 sense AGAGGCCGGATCAAACCAA.

## Plasmid

GST-tagged prokaryotically expressed *ODC1* full-length and segmented plasmids, Flag-tagged *ODC1* plasmid, Flag-tagged *ODC1*-luciferase plasmid (Flag-*ODC1*-luc), GST-tagged eukaryotically expressed *ODC1* plasmid, and Flag-tagged *OAZ1* plasmid were purchased from Beijing Hesheng Biotechnology Co., Ltd. GST-tagged prokaryotically expressed *HOXB9* full-length and segmented plasmids, Flag-tagged *HOXB9* plasmid, and Flag-tagged *Praja2* plasmid were constructed by our research group[37,42]. Fusion cells grown to 70−80% within 24 h of

passaging were transfected with plasmids using polyethyleneimine (PEI, Polysciences) according to the manufacturer's instructions.

## Lentivirus package and stably transfected cell line establishment

Lentivirus packaged sh*ODC1* (CCUCCAGAGAGGAUUAUCUAU; PLV-hu6-shRNA-hef1a-hluc-P2A-puro) and sh*HOXB9*(TGTCGACTCGATCA-TAAGT; PLV-hu6-shRNA-hef1a-hluc-P2A-puro) were purchased from Beijing Hesheng Biotechnology Co., Ltd. Lentivirus packaged sg*ODC1*-exon4 (CACCGACCCTTGCTGCTACCGGGAC; LentiCRISPR v2-Puro) and sg*HOXB9*-exon1(CACCGGCTGCCGCGAGCTCGCGTAC; Lenti-CRISPR v2-Puro) were purchased from Beijing Tsingke Biotechnology Co., Ltd. The procedure was performed as previously described[77]. Briefly, for cells with 40-60% confluence in a six-well plate, lentivirus was lysed on ice and the amount of virus added was calculated based on the viral titer and MOI (Ishikawa: MOI = 4; AN3CA: MOI = 10). Viral volume = viral number/viral titer, viral number = cell number × MOI, and 8 µg/ml of polybrene were added to assist viral infection. Virus and polybrene were added to 1 ml of medium and replenished 1 ml after 4 h to reach 2 ml. The medium was replaced with fresh medium after 12-16 h and puromycin was added after 48-72 h for screening (1–4 µg/ml).

## Cell growth curve assay

Cells were seeded into 96-well plates at a density of 2000–4000 cells per well, with each group consisting of 6–10 replicate wells. At the time point when the cells had adhered to the well surface (designated as 0 h), the plates were imaged and scanned every 24 h using the Operetta CLS High-content system (PerkinElmer, USA). In each well, cell counts were performed at various time points under phase-contrast mode. Subsequently, cell growth curves were constructed using GraphPad Prism 9.0 software. All experiments were independently replicated three times, and the data were presented as the mean ± SEM of these three independent experiments.

## Transwell and Wound Healing Assay

Transwell assay: 1–5 × 10⁴ Ishikawa cells or 10–40 × 10⁴ AN3CA cells were seeded into a transwell insert (Cat#353097, BD, USA) with 8 µm wells, containing 0.2 mL medium with 0.1% FBS. Then the chamber was placed into a 24-well plate containing 0.6 mL medium containing 10–20% FBS. The cells were incubated for 24–72 h. At the endpoint, the chamber was fixed with 4% paraformaldehyde and stained with 0.1% crystal violet. The chamber was then rinsed and the interior side was gently wiped with a cotton swab. The bottom side of the chamber was photographed for at least five fields of view under a microscope (20×) and the number of migrated cells would be counted using Image J software. All experiments were independently repeated 3 times, and the data was represented as the mean ± SEM of the three independent experiments.

Wound Healing Assay: 50–80 × 10⁴ cells were seeded per well into 6-well plates 1 day in advance. The experiment would be performed when the density was about 80–100% and the cells were monolayers. The 200 µL tips were used for scratching. The detached cells were washed off with PBS and incubated with a medium containing 1% FBS in the following days. The same position was photographed with a microscope every 24 h after scratching, and the migration area was calculated by Image J software. All experiments were repeated 3 times independently, and the data was represented as the mean ± SEM of the three independent experiments.

## Immunofluorescence staining

10 × 10⁴ Ishikawa/AN3CA cells were seeded on cover glass slides. At the end of the experiment, the slides were washed with PBS and fixed with 4% paraformaldehyde. The slides were treated with 0.1% TritonX-100 to break the membrane and blocked with 5% BSA subsequently. Then it would be incubated with primary antibody overnight at 4 °C. The next day, the slides were incubated with the corresponding fluorescent secondary antibody (Alexa Fluor 488, 568, or 633 (Invitrogen)) at room temperature. Finally, the slides were stained with Hoechst (Cat. # 34412, Thermo Scientific). Images were captured by confocal microscope (Zeiss LSM 780 with Airyscan, or Nikon) and analyzed by Zen Blue or NIS Elements Viewer 5.2.

## Lipid droplet staining

**Living cells.** 2,000-4,000 cells per well were seeded into a special glass-based, black, opaque CellCarrier-96 microplate (Cat # 6005550, PerkinElmer). At the endpoint, the working solution of the lipid droplet staining solution would be prepared. The Nile Red stock solution (1 mM) or BODIPY 493/503 stock solution (10 mM) were diluted with PBS to make the working solution with a concentration of 200-1000 nM for Nile Red and 1–10 µM for BODIPY 493/503. Hoechst (Cat. # 34412, Thermo Scientific) was added to the lipid droplet staining solution at the same time. Each well was washed with PBS once gently and added with lipid droplet staining working solution and incubated for 10 min at room temperature. Then the wells were washed with PBS once gently and scanned with PerkinElmer Operetta CLS Harmony Imaging Analyzer System. The quantity, area, and fluorescence intensity of lipid droplets were analyzed by PerkinElmer Operetta CLS software.

**Fixed cells.** The previous steps were the same as immunofluorescence, after incubation with a secondary antibody, the slides were incubated with lipid droplet staining working solution and Hoechst (Cat # 34412, Thermo Scientific) for 10 min at room temperature. Images were captured by HIS-SIM super-resolution microscopy. The quantity, area, and fluorescence intensity of lipid droplets were analyzed by ImageJ software.

**Frozen tissues.** The tissues were embedded in Tissue-Tek O.C.T. Compound (Cat#, 4583, SAKURA) and frozen in a −80 °C freezer. The tissue block was cut into 15 µm thick sections by a frozen slicer (Leica). The cryosection would be equilibrated at room temperature for 10 min before use. Cryosections were then fixed with pre-chilled 4% paraformaldehyde for 1 h. After washing in PBS, the slides were incubated with lipid droplet staining working solution and Hoechst for 30 min at room temperature. Then the slides were photographed with a confocal microscope (Zeiss LSM 780 with Airyscan). The quantity, area, and fluorescence intensity of lipid droplets were analyzed by ImageJ software.

## RNA extraction and real-time qPCR

Total RNA was extracted from cells using TaKaRa MiniBEST RNA Extraction Kit (Cat# 9767, Takara). The reverse transcription reaction was performed by cDNA Synthesis SuperMix for the qPCR kit (Cat# 11141es60, Yeasen) according to the manufacturer's instructions. Real-time qPCR was performed by using Hieff qPCR SYBR Green Master Mix (Cat# 11201ES08, Yeasen) according to the manufacturer's instructions, and the reaction cycle was completed and analyzed by Light-Cycler® 96 Instrument (Roche, USA). Each group had 4 replicate wells. Each experiment was repeated 3 times independently. Data were processed using the $2^{-\Delta\Delta CT}$ method. The primers used for real-time qPCR are

*HOXB9*: F: 5'-ACAAAGAGAGGCCGGATCAA-3',
R: 5'-TCATTTTCATCCGCCGGTTC-3';
*ODC1*: F: 5'-ATGTTGCATCAGCTTTCACG-3',
R: 5'-ACTCTCCCAGGCACAAGACA-3';
*GAPDH*: F: 5'-GGAGCGAGATCCCTCCAAAAT-3',
R: 5'-GGCTGTTGTCATACTTCTCATGG-3';

## Protein extraction and WB

Tissues or cells were lysed using a strong RIPA buffer (1× PBS, pH 7.4, 0.5% sodium deoxycholate, 1% NP40, 0.1% SDS) containing cocktail inhibitor (Roche Basel, Switzerland) and PhosSTOP (Roche Basel,

Switzerland). Protein lysates were quantified using the BCA Protein Assay Kit (Thermo Fisher). Samples supplemented with 5 × loading buffer were heated at 100 °C for 8 min and then subjected to SDS-PAGE. PVDF membranes were immunoblotted with primary antibodies overnight at 4 °C. After three washes with TBST buffer, the membrane was incubated with HRP-conjugated secondary antibody for 1 h at room temperature. Enhanced chemiluminescence (Amersham Biosciences, Sunnyvale, CA, USA) was used to detect immobilized antibodies, and ImageJ software was used to quantify and measure the gray density. The list of antibodies used was shown in Supplementary Table 9.

## Subcellular fractionation
The nucleus and cytoplasmic protein extraction kit (Cat# P0027, Beyotime) was used to isolate cytoplasmic and nucleus proteins. Briefly, cell precipitates were suspended in Buffer A containing cocktail inhibitor for 10 min, then Buffer B was added for 1 min. The supernatant, cytoplasmic proteins, was extracted immediately after centrifugation at 4 °C and 13,800 × g for 5 min. The precipitate was then resuspended in Buffer C and shaken for 30 min at high speed for 15–30 sec at 1–2 min intervals. The mixture was centrifuged at 4 °C and 13,800 × g for 10 min. The supernatant was the nucleoprotein. The nuclear and cytoplasmic proteins were then quantified by the BCA Protein Assay Kit.

## Protein half-life experiment
Cycloheximide (CHX) (Cat # HY-12320, MCE) was added at the designed time point before cell collection, and the CHX-containing culture medium was changed every 6 h to prevent drug failure. WB was performed on proteins collected from cells after CHX treatment at different times. Bands were analyzed for gray density using Image J Software and degradation curves were plotted using GraphPad Prism 9.0.

## Luciferase stability assay
Luciferase activity was quantified using the Luciferase Reporter Gene Assay Kit (Yeasen, catalog: 11401ES76) following the manufacturer's instructions. Briefly, transfected cells in 6-well plates were lysed with 500 µL ice-cold lysis buffer per well for 5 min on ice. Twenty microliters of cell lysate were mixed with 100 µL of Firefly Luciferase Assay Reagent in a luminometer tube, and luminescence was immediately measured using a Promega GloMax 20/20 Luminometer. For protein quantification, parallel cell samples were processed using the BCA Protein Assay Kit (Beyotime, catalog: P0012). Luciferase signals were normalized to total protein concentration (reported as relative light units per milligram protein, RLU/mg/mL protein) to account for variations in transfection efficiency and cell number.

## Thermo shift assay
For cell lysates, the cells were rinsed with PBS and then lysed with strong RIPA buffer. The lysates were divided into several 50 µL aliquots and 30 µM of OA was added to each, then heated at different temperatures for 5 min and cooled on ice. The lysates were centrifuged at 20,000 × g for 20 min at 4 °C, and the supernatants were collected and added with 5x loading buffer for WB.

For the purified bacterial proteins, GST-HOXB9 and GST-ODC1, the samples were divided into several 50 µL aliquots and 30 µM of OA was added to each sample, which was then heated at different temperatures for 5 min and cooled on ice. The lysates were centrifuged at 20,000 × g for 20 min at 4 °C, and the supernatants were collected and added with 5x loading buffer for WB.

## Co-IP
The method of Co-IP has been previously described[37]. Briefly, cells were lysed on ice for 30 min using a weak RIPA buffer (1 × PBS, pH 7.4, 1% NP40) containing cocktail inhibitor and PhosSTOP. The cells were

then centrifuged at 13,800 × g for 15 min at 4 °C. Supernatants were pre-cleared and incubated overnight at 4 °C with the indicated antibodies and control IgG. The next day, lysates were incubated with 35–50 µL of protein A or protein G agarose (Cat# sc-2001, sc-2002, Santa Cruz Biotechnology) at 4 °C for 4–6 h. In the case of cellular samples transfected with Flag-tagged plasmids, lysates were incubated with anti-Flag affinity gel (Cat# A2220, Sigma-Aldrich) at 4 °C overnight. After incubation, the beads were washed five times with pre-cooled weak RIPA buffer and then detected by WB analysis. When the molecular weight of the target protein was similar to that of a heavy or light chain, the M2 beads would be eluted with a 3 × Flag peptide (Cat# P9801, Beyotime) according to the manufacturer's instructions. Alternatively, a special HRP-conjugated secondary antibody against the light or heavy chains would be used.

## GST pull-down
Plasmids expressing GST-HOXB9 and GST-ODC1 full-length and segmented proteins were transformed with Rosetta receptor bacteria. Protein expression was induced with IPTG (Cat# I6758-5G, Sigma Aldrich) at a low temperature. The collected bacterial fluid was centrifuged and the precipitate was resuspended with PBS, then the bacteria were lysed by ultrasound. After centrifugation at 13,800 × g for 30 min at 4 °C, the supernatant was the extracted bacterial protein. Bacterial proteins were incubated with Glutathione-Sepharose 4B beads (Cat# GE17075601, GE HealthCare) overnight, and the beads were washed with PBS the next day. Lysates of 293T cells transfected with Flag-tagged plasmids were pre-cleared with GST bacterial protein and 4B beads. The pre-cleared cell lysates were incubated overnight with different 4B beads bound to corresponding GST fusion proteins. The beads were finally washed with PBS and resuspended with 2 × loading buffer and heated at 100 °C for 8 min. WB and Coomassie brilliant blue staining were performed.

## SPR assay
The GST fusion protein was expressed and purified using glutathione Sepharose 4B beads followed by washing with elution buffer (20 mM GSH). Bacterially purified GST-HOXB9, GST-ODC1, and GST were covalently immobilized on the sensor CM5 chip using a Biocore T200 instrument (GE Healthcare) at 25 °C for SPR binding assays. OA was serially diluted and injected into the sensor chip at a flow rate of 30 mL/min for 120 s (contact phase), followed by a buffer flow for 120 s (dissociation phase). The running buffer for this instrument was HBS-EP (10 mM HEPES (pH 7.4), 150 mM NaCl, 3 mM EDTA, 0.5% (v/v) surfactant P20, 5% DMSO). KD values were obtained by a standard amine coupling procedure in 10 mM sodium acetate (pH 5.5) using Biocore T200 evaluation software version 1.0 (GE Healthcare) and steady-state analysis of equilibrium data.

## IHC and multiplex immune fluorescence staining
The IHC procedure was previously described[37]. Briefly, human and mouse tissues were fixed in formalin, embedded in paraffin, and then sectioned. Sections were deparaffinized with xylene and rehydrated with ethanol. Endogenous peroxidase was then eliminated with 3% hydrogen peroxide. The antigen was repaired with sodium citrate buffer at pH = 6.0 or EDTA buffer at pH = 9.0 for 30 min at 100 °C in a water bath. The primary antibody was then incubated overnight at 4 °C. HRP conjugated secondary antibody (Cat# PV9000, Zhongshanjinqiao, Beijing) was added and incubated for 30 min-2 h at room temperature. Diaminobenzidine (DAB) (Cat# ZLI-9017, Zhongshanjinqiao, Beijing) was used for staining, followed by hematoxylin staining of nuclei. Observed and photographed using a microscope (Olympus). The color was scored according to the intensity of the brown color on a scale of 0–4.

Multiplex immunofluorescent staining was carried out in accordance with the instructions of the Multiplex Immunofluorescence

Staining Kit (Cat# TGFP550, Tissue Gnostics). Image scanning was performed using the ZEISS Axioscan 7 Microscope Slide Scanner, and the analysis was completed with the ZEN software and Graphpad Prism 9.0.

## Raman spectroscopy

The patient's tissues were cryosectioned and mounted on Raman-specific slides (D-Band, Shanghai, China). Raman analysis was performed by D-Band Co., Ltd (Shanghai, China). The instrument was WITEC α 300. The single spectrum acquisition parameters were: power 13 hw; time 5 s; objective lens: 100X; 532 nm laser; grating: 600 g/mm; spectral range −190 to 3946 cm$^{-1}$; imaging range: 100 μm × 100 μm; power: 18 hw. Fifty Raman spectra were collected from each section and the lipid peaks of the glandular area were imaged and analyzed.

## Untargeted metabolomics

The untargeted metabolomics study was conducted by the Metabolomics Platform of Tsinghua University, including 62 sera from EC patients (30 MS-positive, 32 MS-negative). QC samples were prepared by mixing equal volumes of all samples (undergoing the same pretreatment and LC-MS analysis) and inserted every 15–20 samples to monitor system stability and reproducibility. For serum pretreatment: 100 μL serum was mixed with 400 μL pre-cooled methanol (1:4, v/v), vortexed for 5 min, stood at 4 °C for 30 min, centrifuged at 18,800 × g (4 °C) for 15 min; the supernatant was vacuum-freeze-dried and reconstituted in 50 μL methanol for LC-MS.

Metabolomics analysis used an Ultimate 3000 UHPLC coupled with an Orbitrap mass spectrometer (Thermo Fisher Scientific, USA). For positive ion mode: BEH Amide column (2.1×100 mm, Waters, USA) at 35℃; mobile phase A (0.63 g ammonium formate in 50 mL HPLC water + 950 mL acetonitrile + 1 μL formic acid), phase B (0.63 g ammonium formate in 500 mL HPLC water + 500 mL acetonitrile + 1 μL formic acid); gradient in Supplementary Table 10. For negative ion mode: BEH C18 column (2.1 × 100 mm, Waters, USA); mobile phase A (1L HPLC water with 5 mM ammonium bicarbonate), phase B (100% ACN); gradient in Supplementary Table 11.

MS analysis adopted data-dependent acquisition (DDA) mode: Positive ion mode (QExactive Orbitrap, Thermo): m/z 70–1050, resolution 70,000 (full scan)/17,500 (MS/MS); source parameters (spray voltage 3200 V, capillary temp 320 °C, heater temp 300 °C, sheath gas 35, auxiliary gas 10). Negative ion mode (QE HF, Thermo): m/z 80–1200, resolution 60,000 (full scan)/30,000 (MS/MS); source parameters (spray voltage 2800 V, other temps/gas flows same as positive mode).

Metabolite identification: DDA data analyzed via Tracefinder 3.2 (Thermo) with an in-house MS/MS library (>1000 metabolites); two levels (tentative via accurate ion masses, confirmed via library MS/MS). Mass tolerance: 10 ppm (precursor)/15 ppm (fragment); allowed retention time shift: 0.25 min. Peak areas (normalized by total chromatographic areas) for relative quantitation.

Data analysis: Normalized data subjected to PCA, PLS-DA, OPLS-DA (with permutation test) via Simca-P 14.1 (Umetrics); univariate analysis (fold change, t-test) with volcano plots. Differential metabolites screened by VIP > 1, fold change > 2/< 0.5, P < 0.05. Metabolic pathway analysis via MetaboAnalyst 4.0.

## Polyamine detection

Polyamines, including ornithine, putrescine, spermidine, and spermine, detection were analyzed by the Metabolomics Center at Key Laboratory of Natural and Biomimetic Drugs Large Instrument Technology Platform, Peking University Health Science Center. The standards and samples were derivatized as Wong's article described[78]. Derivatization improves the chromatographic sensitivity, specificity, and retention of polyamines. For human or mouse serum, 20 μL of serum was used and mixed with 80 μL of acetonitrile to precipitate proteins. Then the

mixture was centrifuged at 4 °C, 20,200 × g for 30 min. 20 μL of the supernatant underwent a derivatization reaction: 10 μL of 100 mM sodium carbonate, 10 μL of benzoyl chloride (BzCl) (2% (v/v) in acetonitrile), and 60 μL of water were added sequentially. The mixture was then centrifuged at 20,200 × g for 30 min, and the supernatant was sent to LC-MS analysis. For cell and tissue samples, the cells were scraped off using chilled 80% acetonitrile, tissues were homogenized in 80% acetonitrile. Parallel cell samples were prepared to count the cell numbers or protein concentration. Tissues were weighed before homogenization. After centrifugation at 4 °C, 20,200 × g for 30 min, the supernatant was freeze-dried under vacuum. The precipitate was dissolved in 50 μL 90% acetonitrile, and 20 μL was taken for derivatization reaction: 10 μL of 100 mM sodium carbonate, 10 μL of BzCl (2% (v/v) in acetonitrile), and 60 μL of water were added sequentially. The mixture was then centrifuged at 20,200 × g for 30 min, and the supernatant was sent to LC-MS analysis. Liquid chromatography was performed using a Waters ACQUITY UPLC i-Class Plus. Mass spectrometry was performed using a Waters Xevo TQ-S. Ion pair information was shown in the Supplementary Fig. 11 and Supplementary Table 12. Data acquisition software: Waters Masslynx V4.2. Data analysis software: Waters TargetLynx XS V.2. The content of derivatized polyamines was shown as the response area divided by the corresponding cell number, protein concentration (mg/mL) or tissue weight (mg).

## OA detection

OA was analyzed by the Metabolomics Center at Key Laboratory of Natural and Biomimetic Drugs Large Instrument Technology Platform, Peking University Health Science Center. Lipids in the samples were extracted by the method of Reis et al.[79]. Standards and samples were derived by the method of Hu et al.[80]. For human or mouse serum, 50 μL of serum was used and mixed with 800 μL of ice-cold hexane contained BHT (50 μg/ml):2-propanol (3:2, v/v), incubate for 20 min on ice with occasional vortex mixing. Then the mixture was reextracted with 200 μL hexane:MeOH (3:2, v/v) for 15 min as above. The organic phase was collected and freeze-dried under vacuum. Then the precipitate undergone derivatization reaction: 13 μL acetonitrile-N,N-dimethyl-formamide (4:1, v/v), 13 μL EDC (130 mg/mL in purified water), 13 μL T3 (2,4-bis(diethylamino)−6-hydrazino-1,3,5-triazine, 10 mg/mL in acetonitrile) and 13 μL HOBT (2 mg/mL in acetonitrile). The reaction were carried on 4 °C for 30 min. The mixture was then centrifuged at 4 °C 20,200 × g for 30 min, and the supernatant was sent to LC-MS analysis. For tissue samples, tissues were weighed before homogenization, tissues were homogenized in 2-propanol: MeOH: water (2:2:1, v/v/v). The mixture was then centrifuged at 4 °C 20,200 × g for 30 min. The supernatant was freeze-dried under vacuum. The precipitate was then undergone derivatization reaction as above described. Liquid chromatography was performed using a Waters ACQUITY UPLC i-Class Plus. Mass spectrometry was performed using a Waters Xevo TQ-S. Ion pair information was shown in the Supplementary Fig. 11. Data acquisition software: Waters Masslynx V4.2. Data analysis software: Waters TargetLynx XS V.2. The content of derivatized OA was shown as the response area divided by the corresponding tissue weight (mg).

## Mice

**Mice models of postmenopausal MS.** All experiments were approved by the Animal Ethics Committee of Peking University People's Hospital (No. 2020PHE094) and the IACUC of Peking University Health Science Center Animal Experimental Center (No. BCJF0170). Given postmenopausal EC predominates in females, only female Mus musculus (Mouse) BALB/c Nude mice (Strain Code: 401; Official Name: CAnN.Cg-Foxn1$^{nu}$ /Crl; an inbred strain with Foxn1$^{nu}$ mutation on a BALB/c genetic background; 4−5 weeks old, 5−8 per group) were used, with sex integrated into the study design.

Bilateral ovariectomy was performed on 4−5-week-old female nude mice. Mice were anesthetized with isoflurane via a respiratory

anesthesia machine, immobilized in the prone position, and incisions were made bilaterally at the lower 1/3 of the dorsal midline. Skin and dorsal muscles were incised to expose creamy-white, shiny fat pads with embedded pink punctate ovaries. Fat pads were grasped with fine forceps, exteriorized, then ovary-containing fat pads were isolated, blood vessels ligated, ovaries excised, and residual tissue returned to the abdominal cavity. Incisions were sutured and sterilized, with the contralateral ovary removed similarly. Postoperatively, analgesic jelly was applied, and a single intramuscular antibiotic injection was administered. Complete ovariectomy was confirmed by the absence of keratinocytes in vaginal smears for 3 consecutive days.

Dietary feeding was continued for 2–3 months. Three days after surgery, mice were randomly assigned to high-fat diet (HFD) and low-fat diet (LFD) groups. The HFD group received a 60% kcal fat diet (Cat# H10060, Beijing Huafukang Bioscience Co., Ltd., with 26% protein, 26% carbohydrates, 35% fat, energy density 5.24 kcal/g, fat from lard and soybean oil), which was previously used to establish a model of non-alcoholic fatty liver disease[81] and a model of EC with obesity[82]. The LFD group received a 10% kcal fat diet (Cat# H10010, Beijing Huafukang Bioscience Co., Ltd., with 19.2% protein, 67.3% carbohydrates, 4.3% fat, energy density 3.85 kcal/g).

Mice were housed in a specific pathogen-free (SPF) facility ($22 \pm 2\,°C$, 40–60% humidity, 12 h light/dark cycle). Both diets were fed in small amounts multiple times daily (HFD was completely replaced weekly), with sterile deionized water provided. At the end of dietary induction, several mice from each group were randomly selected for lipid testing and animal MRI to analyze their lipid status. Mouse whole blood was left overnight at $4\,°C$ and centrifuged at $800 \times g$ at $4\,°C$ for 20 min. The supernatant was serum. The blood lipid test was performed by a biochemical analyzer (BS-450, Myriad) in the Department of Animal Care and Laboratory, Peking University Health Sciences Center or biochemical analyzer (BS-240VET, Mindray) Beijing Jinglai Technology Co., Ltd.

## Animal MRI
Animal MRI testing was performed using a 3.0 T research MRI unit (Siemens Magnetom Trio TimSystem) at the Peking University Health Sciences Center. Magnetic field strength: 3.0 T. Gradient field strength: 40 mT/m. Gradient switching rate: 200 T/m/s. Magnet aperture: 60 cm (inner diameter). The mice were anesthetized with tribromoethanol and placed on the mouse coil in the supine position. Then the adipose tissue program was selected, including T1WI, T2WI, DWI, MRS, etc., to analyze different levels of fat, bone, muscle, water, etc. in the animal body.

## Subcutaneous tumor xenograft model
BALB/c nude mice (female, 4–5 weeks old, 18–20 g) were purchased from Beijing Vital River Company and cultured in the specific pathogen-free animal laboratory of Peking University People's Hospital. After ovariectomized mice were fed with a HFD or an LFD for 2–3 months, they were randomly divided into the shNC group, the shODC1 group, and the shHOXB9 group (the first batch), as well as the sgNC group, the sgODC1 group, the sgHOXB9 group, and the sgNC + DFMO group (the second batch). Ishikawa cells stably expressing shNC, shODC1, or shHOXB9 (in the first batch) or stably expressing sgNC, sgODC1, sgHOXB9 were subcutaneously injected into the axilla of the mice to establish xenograft tumor models, with $3 \times 10^6$ cells per mouse. sgNC + DFMO group would be giving DFMO (1% w/v) in the drink water 1 month after inoculation.

Tumor length (L) and width (W) were measured every 2 days (volume: $V = 0.5 \times L \times W^2$). The humane endpoint was a maximum tumor diameter of 2.0 cm (per ethics approval); mice were also monitored for >20% weight loss or distress. No animal had tumors exceeding the maximum allowable burden. All procedures were fully compliant with the ethical approval requirements. Serum and xenograft tissues were collected for polyamine/OA test, WB, HE/IHC, and lipid droplet staining.

## Foot pad injection LNM model
BALB/c nude mice (female, 4–5 weeks old, 18–20 g) were purchased from the Animal Experimental Center of Peking University Health Science Center and cultured in the specific pathogen-free animal laboratory of Peking University Health Science Center. Ovariectomized mice fed with a HFD or LFD for 2–3 months would be randomly divided into shNC, shODC1, shNC + DFMO, and shHOXB9 groups. AN3CA cells stably expressing luciferase-labeled shNC, shHOXB9, and shODC1 were injected into the foot pads of mice at $5 \times 10^6$ cells per mouse. shNC+DFMO group would be giving DFMO (1% w/v) in the drink water 2 months after inoculation. At 3 months post-injection, D-luciferin potassium salt (Solarbio, Beijing) was injected into the peritoneal cavity of mice, and the luciferase signal of LNM was detected using IVIS Spectrum (PerkinElmer). Experiments terminated for extensive metastasis or severe morbidity (>20% weight loss). Serum and lymph nodes were collected for polyamine analysis and HE/IHC. The lymph node area was calculated using Image J based on the images, with the unit standardized according to the image scale.

## Bioinformatics analysis
Molecular docking simulations were performed by the CB-DOCK online platform (https://cadd.labshare.cn/cb-dock2) on OA (SMILE: SCCCCCCC=CCCCCCCCCC(=O)O) and HOXB9 (https://alphafold.ebi.ac.uk/entry/, P17482). Predictive analysis of transcription factors and binding sites was performed by the UCSC Genome Browser Home (http://genome.ucsc.edu/) and JASPAR by inputting the promoter region of ODC1.

## Statistics and reproducibility
Clinical samples were collected from all patients enrolled in the specified year who met the inclusion criteria and did not meet the exclusion criteria. The sample size was predetermined via power analysis ($\alpha = 0.05$, power = 80%) using G*Power software (3.1.9.4), and the actual number of enrolled patients met the estimated requirement. No data were excluded. Given the observational nature of this study, it did not involve random group assignment. Instead, samples were included based on pre-defined eligibility criteria to reflect the real clinical population. To minimize observer bias, investigators were blinded to key sample information during experiments and outcome assessment. Results were presented as mean ± SD (clinical continuous variables) or counts/percentages (N, %, categorical variables); mean ± SEM for other experiments. Normally distributed continuous data used two-tailed Student's t-test, one-way/two-way ANOVA; non-normal/ranked data used two-tailed Mann-Whitney U or Kruskal-Wallis test. Multiple comparisons were corrected. Correlation used Pearson or Spearman test; survival analysis used log-rank or Gehan-Breslow-Wilcoxon test. Significance was $p < 0.05$ (*). Analyses: IBM SPSS 25.0; graphs: Graph-Pad Prism 9.0. All in vitro experiments were independently repeated ≥3 times ($n \geq 3$ independent experiments) with consistent trends. Source data are provided for verification.

## Reporting summary
Further information on research design is available in the Nature Portfolio Reporting Summary linked to this article.

# Data availability
All data supporting the findings of this study are available within the paper and its Supplementary Information. Source data are provided with this paper. The metabolomics data have been deposited to MetaboLights repository with the study identifier MTBLS13106. Relevant plasmids and oligonucleotides can be purchased from

commercial companies or obtained by contacting the corresponding author. Source data are provided with this paper.

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

## Acknowledgements

The study was supported by Key project of the National Natural Science Foundation of China 82230050 (J.L.W.), National Key Technology Research and Developmental Program of China 2022YFC2704400, 2022YFC2704402 (J.L.W.), National Natural Science Foundation of China 82372632, 82172972 (J.Z.), National Natural Science Foundation of China 82203568, 82173119 (Y.C.), Peking University Clinical Medicine + X Youth Project PKU2024LCXQ025 (Y.C.), Beijing Natural Science Foundation BJNSFC7252077 (J.Z.), National Key Research and Development Program of China 2022YFA1104003 (H.Q.Z.), National Natural Science Foundation of China 82230094 (HQZ), National Natural Science Foundation of China 82072861 (J.L.W.), Peking University People's Hospital Research and Development Funds RDEB2024-05 (L.R.Z.), Peking University Medicine Sailing Program for Young Scholars' Scientific & Technological Innovation and the Fundamental Research Funds for the Central Universities BMU2025YFJHPY020 (L.R.Z.), Peking University Medicine Fund of Fostering Young Scholars' Scientific & Technological Innovation BMU2022PYB028 (X.Y.), Peking University People's Hospital Research and Development Funds RDJP2022-09 (X.Y.). We thank Prof. Di Zhang of Peking University, Prof. Qunying Lei of Fudan University, and Prof. Binghui Li of Capital Medical University for their guidance on this research. We thank Dr. Yuan Wang and Dr. Pushu Wang from Peking University Health Science Center for metabolomics testing. We thank Dr.

Luzheng Xu from Peking University Health Science Center for mice MRI imaging and analysis. We thank Dr. Yin from Peking University for PTCs culture and drug testing. We thank Xiaojing Liu from Tissue Genomics Co., Ltd for mIHC analysis. We thank staffs from Animal Laboratory of Peking University of People's Hospital and Peking University Health Science Center.

## Author contributions

Conceptualization: J.L.W., J.Z., Y.C. Methodology: L.R.Z., Y.C., M.X.W., T.Z.W., M.C.Y., X.Y., C.C.L., B.W.S., M.H., Y.S., Y.Q.Z., Y.Q.X., B.L., M.Y., L.Z., Y.Y.L., Y.J.H. Investigation: L.R.Z., Y.C., M.X.W., T.Z.W., M.C.Y., B.W.S., M.H., C.C.L. Visualization: L.R.Z. Supervision: J.L.W., J.Z., Y.C., H.Q.Z. Writing—original draft: L.R.Z., J.Z. Writing—review and editing: L.R.Z., J.L.W., J.Z., H.Q.Z., Y.C.

## Competing interests

The authors declare no competing interests.
