## [Transparent Peer Review file · Nature Communications]

Metabolic Syndrome Promotes Endometrial Cancer by Oleic Acid-Mediated Polyamine Accumulation

Corresponding Author: Professor Jianliu Wang

Version 0:

Reviewer comments:

Reviewer #1

(Remarks to the Author)

The paper by Zhai et al. explores the effects of the oleic acid (OA)-HOXB9-ODC1-Polyamine pathway on endometrial carcinoma progression. The authors have found an increase in polyamine metabolites in the serum of endometrial carcinoma patients with metabolic syndrome (ECWMS). Authors report that OA stabilizes HOXB9 that in turn stabilizes ODC1, thereby catalyzing production of polyamine metabolites. It is concluded that targeting the OA-HOXB9-ODC1-polyamine pathway may represent a therapeutic approach. The manuscript is logically assembled and offers mechanistic insights into interactions of individual pathway components. Results are well summarized and clearly outline where this work fits in with already known or published mechanisms. The role of OA in cancer progression remains debatable and this study may contribute to addressing this important topic. However, there are several significant concerns about experimental design of this study and its conclusions.

Major concerns

1. Is endometrial carcinoma the only source of elevated concentration of polyamines ECWMS patients?
2. The majority of experimental findings are based on two 2D cell lines. The study would be significantly strengthened by including endometrial carcinoma organoids.
3. It is unclear if reported findings are applicable to all types of endometrial carcinoma or only endometrioid type. In Table S1 endometrial carcinoma type is not described.
4. Knock downs by siRNA or shRNA are known for side effects. Using gRNA-based gene inactivation would strengthen key conclusions.
5. It seems different cell lines were used for the different models (ISK=subcutaneous, AN3CA= foot pad). Is there a reason DMFO was not tested on subcutaneous tumor models?
6. All tetrazolium-based assays for cell viability, such as Kit-8 (CCK-8) assay, may be impacted by different metabolic state of assayed cells. Direct cell count is required to exclude this possibility.
7. Fig 3I: Differences in expression seem to be very minimal. Given low resolution of images it is impossible to exclude staining artifacts, different thickness of sections. Also, number of patients tested is very limited for this type of semi-quantitative studies.
8. Supp. Figure 2 Please show quantifications of your western blots to support your claim that "only OA significantly upregulated ODC...(Supp. Fig. 2A)". By eye, the WB does not convincingly show this. The same is true for a few of the WB in this figure and Figure 2.
9. Supp Figure 6,c,e images are of insufficiently high resolution to interpret histology, especially in case of lymph node metastasis. Some immune cells are known to express keratins. Additional staining, such as PAX2 would be more

convincing. Also antibodies recognizing only human epithelial cells could be used..

10. More thorough description of statistical assays is required. Nonparametric testing may be more applicable for many of the presented results.

11. The study has several overstatements or erroneous conclusions: Some examples below:

- a. The running title "Metabolic syndrome induces EC": The manuscript provides no experimental evidence that endometrial carcinoma is induced by metabolic syndrome.
- b. "This study demonstrated that increased polyamine metabolism is associated with induced EC progression" What is induced EC progression?
- c. "Hyperlipidemia in MS is the main driver of EC" How about known genetic drivers, such as PTEN mutations?
- d. "The OA-HOXB9-ODC1-polyamine axis fully explains the key mechanism by which MS leads to the progression of EC". Fully? No other possibilities? For example, LOA also significantly increases cell motility according to your data.

Minor concerns

1. Is there a direct correlation between oleic acid concentration and polyamine accumulation in serum of ECWMS patients?
2. Are there any known mutations in HOXB9-ODC1 pathway in endometrial carcinoma?
3. Needs some editing to improve readability. For example:
 - a. Lane 78: paragraph starting with "Tumor metabolic reprogramming is a hallmark for tumors", includes a very long, run on sentence that can be confusing. Try to aim for more concise language.
 - b. Lane 89: Clarify sentence: "They are important regulatory metabolites, not like glucose, lipids, and 90 amino acids are main "fuel" for tumor, which play crucial regulatory roles in multiple 91 physiological and pathological processes of the body, supporting for cancer metabolism and signal transduction^{25,26}"
4. Please make sure you are consistent using "untargeted" or "non-targeted" metabolomics.
5. Lane 211 and other places: "knocking out HOXB9 with siHOXB9 shortened it (Figure 3d)" You cannot knockout gene with siRNA or shRNA. Only knockdown can be accomplished this way.
6. Section "OA in hyperlipidemia triggers an increase in ODC1 and polyamines" Line 174 and below. Ratios following fatty acid names need to be explained. What do they signify?
7. The same section: Clarify why you chose to treat with high glucose. This seemed to come out of nowhere in a section focused on testing different fatty acids.
8. Figure 1c: The scale bar going from high to low is not clear that it correlates to the colors next to the chart indicating relative concentrations because no other colors on the gradient are present. Remove the bar and just state red is high and green is low.
9. Figure 1d: more explanation of what is shown is needed. What do the different node colors mean? What do the different node shapes mean?
10. Figure 1i: typo, Are you addressing, 1h not 1g?
11. Figure 1b: For improved readability, reorganize to have ODC1 and HOXB9 on the top row (since you mention them first in the text) with OAZ1 and Praja2 in the second row.
12. Figure 2b: make clear that "DL" is your control condition and be consistent. Figure 2F has the control labeled as control, so we don't know if there is an inherent difference between these.
13. Figure 2g: Are there any corresponding results for ISK cells? Since the two cell lines seem to show different responses to these conditions, such results are important.
14. Figure 4a/b: Please show results for both cell lines or explain why they may be behaving differently, and you think one's results may be more supportive than the others.
15. Supp Fig 4a: Add ISK label to the graph.
16. Figure 5f does not get cited.
17. Supp Fig 1a: Typo in Y axis label
18. Supp Table 4: What are 0 and 1?

Reviewer #2

(Remarks to the Author)

Reviewer #3

(Remarks to the Author)

The manuscript from Zhai et al focusses on the mechanisms linking hyperlipidemia to polyamine metabolism in cancer progression. By performing serum metabolomics of patients with endometrial cancer (EC) and with or without metabolic syndrome (MS), authors have identified high levels of polyamine metabolites as markers of EC/MS. Increased polyamines were associated with increased expression of ODC1 in tumors, a key enzyme of polyamine synthesis. They then unveil a new mechanism by which lipids, and especially oleic acid (OA), prevents HOXB9 proteasomal degradation, therefore enhancing its stability. HOXB9 in consequence interacts with ODC1, which precludes its degradation by OAZ1 and promotes polyamine synthesis. Polyamines finally promote cancer cell proliferation and migration. Authors link this mechanism to the worsened survival prognosis of EC and MS patients. They finally bring an in vivo proof-of-concept of the OA-HOXB9-ODC1-polyamine axis involvement in tumor growth. Overall authors have identified a new metabolic pathway by which MS and hyperlipidemia promote polyamine synthesis and support tumor growth.

The manuscript is well written and with a lot of detail. Conclusions are in general convincing and supported by a large number of experiments performed in cellular models, mouse models of EC and patients tissues, as well as by different experimental approaches. Nevertheless, I would still need some clarifications and would recommend additional measurements to be done in order to strengthen the demonstration:

1) Concerning patients' data:

- Please combine tables S1 and S2 and directly compare the two groups (including calculating significance of differences) to highlight any difference between them. In table S4, please also present and compare the two groups MS and no MS (validation cohort).
- Among the most enriched metabolites in EC + MS patients, metformin is more than 7 fold increased, suggesting that patients with MS may receive anti-diabetic treatments. In my view, this is a very important information, which needs to be included in the tables (S1, S2, S4). Metformin is a powerful drug and fasting mimetic, and can have serious metabolic consequences on its own. Thus, in case patients EC+MS would be under metformin treatment, a necessary control experiment is to treat EC cancer cells with metformin and measure polyamine levels in order to rule out any effect of anti-diabetic treatment(s) on the pathway of interest.
- The paragraph starting from line 147 speaks of "correlations" but no correlation analysis has been performed, so the results do not justify the conclusion. Either this needs to be re-phrased, or a different analysis and representation of the data is needed (i.e. correlation analyses) to justify the conclusion.
- Regarding Figure 1h-k: Metabolomics has been performed from patients' serum. Nevertheless, this is not representative of the tumor itself and may be the consequence of metabolic alteration in other tissues such as liver. I would therefore encourage the authors to confirm the increase in polyamine levels in patients' tumors.
- In Fig 7, I would rather recommend to perform lipidomic analysis to confirm the increase in OA levels rather than using an unspecific approach. In Fig 7f-l, I would also be interested in the effect of high and low expression in MS and non-MS patients and in patients with or without hyperlipidemia.
- Also in Fig. 7, there seems to be a significant effect on Praja2 levels in the quantification (7b, $p < 0.01$) but the bars are at exactly the same height, so this is unlikely. Also, the "representative" picture (7a) shows reduced Praja2 whereas the quantification shows increased – or unchanged(?) levels.
- Also in Fig. 7a-e, as I understand it, data from only 1 patient per group is shown (MS+#3 and MS-#1). While this may be fine for representative images as in 7a and e, data of at least 5 patients / group needs to be quantified here in order to make any meaningful conclusions.

2) Concerning in vitro mechanistic studies:

- Please specify what DL, which is often used as a control for OA treatment, means.
- In general, I regret that the authors mostly provide qualitative instead of quantitative data for many experiments. Some blots are N=1 per condition and the effect is not always clear by eye. In order to strengthen the claims, it will be important to provide a quantification and statistical analysis of the blots, especially for Fig 2e, 2j, 3j, 3k, 4a, 5b, S2a, S2b, S3a.
- Effect of OA on polyamines does not seem very stable (Fig 2d: unchanged at 48h in AN3CA cells) and transient (unchanged at 48h and decreased at 72h in ISK cells). It is therefore puzzling that the authors still see an effect on cancer cell migration in S2g despite no increase in polyamines. Could the authors comment on this? Same goes for Fig 4b where OA-siC has no effect on ODC1 stability compared to DL-siC. In Fig 4c and 4d, the decrease in cell proliferation and migration in OA + siODC1 is rescued by HOXB9 overexpression, however this is not mirrored by change in polyamine levels (Fig 4e and 4f), questioning the concept. Down the same line, it is not very logical that effects on ODC1 protein stability are observed at the 10 minutes timepoint (Fig. 2f, bottom) but an increase in polyamines only becomes apparent at 24h. Similarly, OA effect on HOXB9 stabilization is only observed after 24 hours, but it should stabilize ODC, which has a completely different kinetic. Authors should be much more careful in their conclusions, because the stability assays do not actually support the conclusions of this manuscript. Or, these kinetics need to be better explained.
- It is not logical that the authors conduct the experiment on ODC1 degradation (S3a) with the Flag overexpression construct, which obviously is unresponsive to OA (compare to Fig. 2g red vs orange bars) – this needs to be repeated using the endogenous protein that responds to OA. Same for Fig. 5f
- How did the authors start looking at HOXB9, 13? This is not clearly introduced in the paragraph starting on line 204. Why did the authors assess HOXB13 in Fig 3b ? Since HOXB13 also enhances ODC1 stability, what was the rationale to focus

exclusively on HOXB9 later?

- line 215: Exploring further, we observed an endogenous interaction between ODC1 and HOXB9 in EC cells treated with OA, which was notably weaker without OA treatment (Supplementary Figure 3d, Figure 3f) -> only data for +OA is shown in Figure 3f - if authors want to make a claim about "stronger" or "weaker" interaction (which is difficult with a semi-quantitative method such as co-IP anyway), they need to show +/- OA conditions on the same blots.
- Figure 5a: The western blot (left) does not show the same as the quantification (right). If normalized to time point 0 for both groups, HOXB9 degrades over time with OA but not without OA.

3) Concerning in vivo experiments in preclinical models:

- I am surprised that 2-3 months 60% HFD only led to a mild increase in body weight gain and lipid accumulation. In general, I am missing the validation of MS development in the HFD-fed animals. In addition to TAG levels, I would recommend to also measure free fatty acids in the plasma. I would also recommend to do a lipidomic analysis of mouse plasma and/or tumor to prove the enrichment in OA in HFD-fed mice. Also, could the authors provide the total fat and lean mass from Fig 6e ?
- In the second model, I am not really convinced about the difference in body weight shown in Fig S6a. Can the authors also provide quantitative data on body composition (fat and lean mass) and plasma free fatty acids and TAG levels ?
- In Fig S6b, I am not really convinced about the effect on shRNA on ODC1 and HOXB9 levels in the HFD group. Could the authors quantify the different protein levels to perform statistical analysis? It is unclear which band should be HOXB9, as no band disappears in the knockdown group specifically. As a minimum, the correct band must be indicated vs. unspecific bands, but actually this blot shows that the knockdown of HOXB9 was not very successful. Similar, but slightly more convincing, for ODC1
- In general, I find the effect of shODC1 on tumor growth very convincing but I am not fully convinced about the link between hyperlipidemia, polyamine metabolism and tumor growth. Indeed, despite increased polyamine levels in HFD versus LFD, there is no difference in tumor volume and tumor weight (6g). Also, shHOXB9 led to a stronger inhibition of polyamine synthesis than shODC1 but had no significant effect on tumor growth. So the association between polyamine levels and tumor growth in vivo is not really clear, could the authors comment on that?

Minor comments:

- N of each experiment needs to be reported in figure legends
- Statistical test used for each graph needs to be specified. (for each subfigure, not simply for the whole figure).
- Please, define each abbreviation in the figure legends (for instance abbreviations of the different lipid species)
- Some proteins are presented in the WBs but not mentioned in the text, please provide the rationale of assessing them. For instance: EZH2 and Twist (Fig 2j, S6b), ZEB1 (Fig 4a).
- Please note that the plural of serum is "sera", not "serums"
- Please change the color code from red-green (e.g. Fig. 1b, 1c) to red-blue or similar so that people who are color blind can also read the differences.
- Line 159, specify which "tissues"
- Scale bars missing for Fig. 3i, right
- A general recommendation: For some figures, the principle "less is more" applies. For example Figures 4c-f: It is almost impossible to read all the different conditions and situations here, and for clarity, a focus on the key message would strengthen, rather than weaken, the message.
- Lines 288, 289: here, authors obviously refer to figures S5e and 5f, not S3e and 3f
- Line 320: "Moreover, knockdown of HOXB9 and ODC1 could inhibit the increase of polyamine metabolites in serum induced by HFD (Figure 6i)." The conclusion is not fully supported by the data. Also, how can you do statistics on n=1 (spermine, black bar)
- When first introducing DFMO, it should be specified that it is an ODC1 inhibitor
- I was unable to find the ISK cell line under this name in the ATCC database. Did they have a different source, or are they sold under a different name at ATCC?

Reviewer #4

(Remarks to the Author)

Version 1:

Reviewer comments:

Reviewer #1

(Remarks to the Author)

The authors sufficiently addressed the majority of previous reviews. However, the manuscript could be improved further by addressing comments below.

Major Comments:

1. The text still has a few statements that oversell the findings of this paper.

- a. Line 125: "The OA-HOXB9-ODC1-polyamine axis ... leads to the progression of EC". The authors' data shows that it "supports" progression, but leading is a very strong conclusion.
 - b. Line 502: "The enhancement of ... is an EC-dependent mechanism of MS and is highly significant in EC". Could the authors state this without also analyzing healthy post-menopausal women with MS to see if they also have an upregulation of this pathway?
2. The authors' validation of lymph node metastasis with the anti-PAX2 IHC stain in SF 8j is not very convincing. Perhaps they could try to use some of human-specific markers, such as human-specific GAPDH, Ku80, hMito and/or Alu?
 3. It is unclear why the authors have decided to use different timepoints to analyze the two different cell lines in Figure 2. Perhaps the inherent differences between the cell lines – based on their derivation – could be useful to mention and explain why they may be treated/reacting differently in these experiments. The corresponding Results for Figure 3D is not shown for Ishikawa cells.
 4. Figure 3I: The authors' findings state that there is a correlation of expression of ODC1 and HOXB9 in patients with LVSI or LNM. Comparing between the two groups would be more convincing if the authors are tying this axis to EC progression in MS+ patients.
 5. It is interesting that the results in Figure 6c and Supplementary figure 7c are so different. Shouldn't the two models be consistent between the two models if they both had an LFD and HFD group? Also, in the caption for Supplemental Figure 7c, the authors do not mention for which subgroup of mice these measurements were done.
 6. Line 363: "we found that the tumor size was closely related with the putrescine content in the tissues". There is no analysis for this, and based on comparing Fig. 6f and 6j, the conclusion is not convincing.
 7. Supplementary Figure 9: based on the differences BMI groupings between the authors' ODC1 (BMI>40) and HOXB9 (30>BMI>34.9) analysis, it is not convincing that there is overlap in patients with high ODC1 and HOXB9. This weakens the authors' conclusion in line 426, "High levels of HOXB9 and ODC1 are closely related to patient weight and predict poor outcomes". What happens when the authors use the same patient groups?
 8. Regarding Figure 7j – these results are only from PTCs derived from a single patient sample? Considering individual patient variability, it is unclear how strong conclusions can be drawn from these results.
 9. What is "conservative treatment" mentioned on line 431?

Minor Comments:

1. The graphical abstract: the arrows in the MS half of the schematic are not logically placed, and it makes it difficult to clearly summarize the mechanism being described.
2. Please reword for clarity:
 - a. Line 54: "MS reprogramming polyamine and promoting EC progression". In its current state, it is unclear what key mechanism is referring to.
 - b. Line 89: "Rearrangement of glucose, ... preventing redox stress and ferroptosis."
3. Thank you for clarifying the authors' focus on endometrioid EC for the authors' analysis. Perhaps authors could add that high incidence and its link to metabolic disorders are the reason for their focus on EC.
4. Figure 1j: Please state that red=high (or upregulated) and blue=low (or downregulated) in the figure caption.
5. Supplementary Figure 2a: Is there a clearer way to indicate that the authors are looking at HL patients only? Based on the authors' labels from the previous figure, it looks like these samples are HL negative.
6. Figure 2: Please add the FFA acronym description to the caption.
7. Figure 3I caption: The levels of expression were not measured using IHC but detected. Additionally, instead of "Up" and "Down", the figures were "left" and "right".
8. Line 206: Cited Supplementary Figures 2c and 2d when referring to Supplementary Figure 3. Additionally, supplementary figure 3 citations are out of order. (ie: SF3c and SF3d are cited before SF3a or SF3b).
9. Line 307: "Our previous work ... proteasome pathway". Please state this earlier in the text when the authors first introduce Praja in Supplementary Figure 4.
10. Figure 4: Please state what the boxes are indicating in each subfigure (where applicable). Applicable for supplementary figure 5, as well.
11. Figure 7a: Please clearly indicate which frames are the enlarged frames being called out in the figure caption.

12. Figure 7f: The blue frames do not have a correlating scale. Is it just a nuclear stain?

13. Figures 7i-j: Please do not group the captions together. Each subfigure should have their own caption.

a. The IHC is not of the patient, but of a tissue. Please indicate what tissue.

b. Please check the authors' units. Does "uM" mean "µm"?

c. Please check the spelling of "Carboplatin" both in the authors' figure and the caption.

14. Supplementary figure 7c: please indicate the number of samples tested.

15. Supplementary figure 8b: It is unclear why the authors have shown these images when they were already added to Figure 6d. It would be expected that the result of the HFD and LFD should have been consistent between the LMN and subcutaneous models. If not, this needs to be explained.

16. Figure 7i – the authors show no staining for immune markers in this panel, though the authors claim these PTCs have immune cells present.

17. Please be consistent with untargeted and non-targeted between figures and the main text. This is still not accomplished in the revised version.

Reviewer #2

(Remarks to the Author)

Reviewer #3

(Remarks to the Author)

I thank the authors for their efforts in assessing some of our important comments. I think the new sets of data from patients, including measurements of polyamines and OA levels in tumors and assessing confounding factors such as metformin on their pathway of interest, really strengthen the overall concept of the study. The new in vivo experiments also strengthen the manuscript, although I would still have comments listed below. However, I am still not convinced about the in vitro data, and I regret that the authors did not assess important comments from our side, which were crucial in my view (please see comments below). Overall, I find the manuscript valuable and of interest for the scientific community, but I still think that it needs major revisions to represent data according to good science practices and to improve the trust of the readership.

Major comments:

- I really regret that authors did not follow the advice raised by us and the other reviewers concerning quantification of their western blots, especially for panels 2a, 2g, 2l, 3j, 3k, 4a, 5b, S3b, S4a, S7d, S8a, S8h for which we had specifically asked for. It is not sufficient to adjust only pictures' contrast. In vitro data are still mostly qualitative and seems to be only N=1 which does not support good scientific practice. I would prefer to have "less" crowded figures, but crucial experiments to be repeated several times with proper statistical analysis. What was the reason for ignoring these comments ?
- Also, co-IP data are really qualitative and often not sufficient on their own. Co-IP data are usually supported by luciferase assays for instance, to get quantitative activity data and to support the overall concept via another approach. For instance authors could use cells overexpressing ODC1-luciferase and activity could be recorded with different doses of HOXB9 overexpression plasmid in the presence or not of OA. Such data would strongly support the stability assays, which are not very convincing and mostly non-significant between DL and OA, except for one early time point in ISK cells.
- This is also the case for in vivo experiments, where I am still not convinced about knock-down efficiency for HOXB9. It is mandatory to assess mRNA levels and to quantify WBs to prove the significant reduction induced by the different silencing approaches, as quality controls. E.g. figures S8a, S8h.
- We had asked authors to include the N of each experiment in the figure legends, particularly for those western blots, which seemed to be only N=1. However, this has not been done, lacking transparency about data.
- Concerning statistical analysis, all reviewers asked for more appropriate statistical analysis – checking for normality and applying non-parametric and parametric tests whenever necessary. However, authors have decided by themselves whether data "seemed" normal or not based on QQ plots if their data did not pass normality tests, and therefore applied only parametric tests throughout the whole manuscript. Plus, for some graphs with more than 2 groups, authors have decided to use both ANOVA and t test (Figure 4c-e for instance), probably to get more significance, which is completely wrong. This refers to my previous comment where, in my view, it is better to have less experiments but a higher N for more reliable data, proper statistical analysis and conclusions.
- I am still not convinced about the specific effect of OA... First figure 2a has to be repeated and quantified to ensure that only OA can exhibit such an effect on ODC1. Second, for a few crucial experiments, such as lipid droplet formation and effect on HOXB9 and ODC1 stability, I would use another fatty acid as negative control, like PA which is strongly enriched in metabolic syndrome and a more toxic lipid species than OA. PA will also induce lipid droplet formation; I would then be curious to know the impact on the pathway of interest. Overall, PA would be a more suitable negative control than DL for the overall concept.
- Also, HFD is mostly made of "lard and soy oil" as mentioned in the methods section. So HFD is more enriched in PA and n-6 fatty acid than OA, which supports even more to make a few experiments in vitro to rule out the role of other fatty acids on

their pathway of interest.

Minor comments:

- Concerning in vivo experiments, I would include the new dataset with sgRNA in the main figure, as the effect is more convincing than the old dataset.
- Is HOXB9 known to bind lipids such as the PPAR nuclear receptor family ? Or is it something authors assume it works this way ? Do authors think HOXB9 only binds OA over other fatty acids ?
- I am surprised that authors could not measure total free fatty acid levels in serum for “technical reasons”. What was the problem ? FFA colorimetric assay, such as NEFA-HR R1 and R2 sets from FUJIFILM Wako Chemicals, are very easy to perform.
- Figure 6h, S8g: is there a good reason for not showing individual dots here ? Also, please represent the different groups always in the same order and same color codes (4h vs 4i, j, k).
- Once again, authors often mention correlations although they did not perform correlation analysis. Better to use “association” in those cases.
- Page 10, line 206: shouldn't be figures S3C and S3D instead of S2C and S2D ? I think figures S3A and S3B are not cited in the main text.
- Page 16, lines 364-367: “whereas knock-down of HOXB9 [...] reduced polyamines and OA (Figure 6j and k, S7g)”. Not true, NS for HOXB9 silencing compared to HFD controls for all these panels.
- Discussion, page 20, line 443-444: “We identified OA, the most prevalent and enriched monounsaturated fatty acids”. How can authors make such a claim without performing lipidomics in patients, or at least their mouse models under HFD ? If this is a fact of general knowledge, please include a proper reference.
- Discussion, page 21 line 463-466: “we approached the problem from the angle of metabolic reprogramming in tumor cells and their microenvironment”. This statement is not really accurate, as authors did not look at the microenvironment per se, otherwise explain better which part of the manuscript support such a claim.
- Page 21, line 467: “We established OA as a crucial upstream regulator among various fatty acids”. I disagree as this statement is only supported by figure 2a, with n=1 and no quantification.

Reviewer #4

(Remarks to the Author)

Version 2:

Reviewer comments:

Reviewer #1

(Remarks to the Author)

The manuscript has been much improved and better supports main findings.

However, it is still unclear why two different BMI cutoffs for each gene were chosen instead of showing the effect on survival for both cutoffs for each gene independently (Supp Fig 9d-g). It seems to be important information to provide.

Line 389-399: “... sub-iliac lymph nodes were significantly enlarged”. Using “significantly” in this statement is too strong without quantification and statistical analysis.

Reviewer #2

(Remarks to the Author)

Reviewer #3

(Remarks to the Author)

We thank the authors for the careful 2nd review of their manuscript. The authors have addressed all main points and added experiments or replicates to support their claims. The provision of the source data including statistical analyses immensely strengthens the manuscript.

Minor remaining points:

- Axis labels should be unified. For example, Fig., 6j, k, l all show the same (response area) but have different label including capital letters, unit etc. This is true for multiple other figures also
- Fig 6c, axis label should be mmol
- Regarding question 4 by reviewer 1 (“Figure 3l: The authors’ findings state that there is a correlation of expression of ODC1

and HOXB9 in patients with LVSI or LNM. Comparing between the two groups would be more convincing if the authors are tying this axis to EC progression in MS+ patients"): The authors claim, also in the figure legend, that 17 patients were used for the analysis, yet only 8 data points are shown. This holds true for both the data shown in the rebuttal letter and the manuscript, where the figure legend says 17 and the figure shows data from 11 patients. I assume some "dots" represent more than 1 patient but this then needs to be stated.

- Regarding same question, authors show the correlation between ODC1 score and grade in the rebuttal letter but not the manuscript - would that not be a valuable addition?

- For Fig. 5f, 5h: no repetitions are shown in the source data but the legend indicates 3 independent experiments; this data is not quantified

Reviewer #4

(Remarks to the Author)

Version 3:

Reviewer comments:

Reviewer #1

(Remarks to the Author)

The authors have satisfactorily addressed my comments from the last round of revisions. I recommend this manuscript for publication.

Reviewer #2

(Remarks to the Author)

Reviewers' comments:

Reviewer #1 (Remarks to the Author); expert in endometrial cancer

The paper by Zhai et al. explores the effects of the oleic acid (OA)-HOXB9-ODC1-Polyamine pathway on endometrial carcinoma progression. The authors have found an increase in polyamine metabolites in the serum of endometrial carcinoma patients with metabolic syndrome (ECWMS). Authors report that OA stabilizes HOXB9 that in turn stabilizes ODC1, thereby catalyzing production of polyamine metabolites. It is concluded that targeting the OA-HOXB9-ODC1-polyamine pathway may represent a therapeutic approach. The manuscript is logically assembled and offers mechanistic insights into interactions of individual pathway components. Results are well summarized and clearly outline where this work fits in with already known or published mechanisms. The role of OA in cancer progression remains debatable and this study may contribute to addressing this important topic. However, there are several significant concerns about experimental design of this study and its conclusions.

Major concerns

1. Is endometrial carcinoma the only source of elevated concentration of polyamines ECWMS patients?

Under physiological conditions, polyamines generally accumulate in the liver, small intestine, and thymus^[1]. The basal levels in the serum of obese female patients are low (putrescine: 20 ± 6 ng/ml, spermidine 14 ± 10 ng/ml, spermine 12 ± 20 ng/ml)^[2]. However, under the pathological conditions associated with EC, we believe that the much higher level is derived from the tumor tissue.

2. The majority of experimental findings are based on two 2D cell lines. The study would be significantly strengthened by including endometrial carcinoma organoids.

Thank you for your very good suggestion. We performed an EC organoid experiment in accordance with the reviewer's suggestion. The results showed that OA promoted organoid growth and proliferation (Figure 2c), strongly supporting our hypothesis. Furthermore, we constructed EC patient-derived tumor cells (PTCs)^[3], which represent EC tissue more accurately than organoids and have components such as epithelial cells, fibroblasts, and immune cells. We verified the effects of targeting ODC1 in PTCs of EC patients with metabolic disorders (Figure 7h-j).

Figure2c

Figure 7h-j

3. It is unclear if reported findings are applicable to all types of endometrial carcinoma or only endometrioid type. In Table S1 endometrial carcinoma type is not described.

Endometrioid EC (EEC) is the most common pathological type of EC, accounting for 80% all cases^[4]. Clinical observations have shown that EEC is often associated with metabolic disorders, while other types, such as USC and CCC, show no close relations to such disorders. Therefore, we focused on EEC, which is most closely related to metabolic disorders. We have added a description of the

histological type as EEC in the headings of Table S1, Table S3, and Table S4. The molecular mechanism outlined in this paper is currently applicable to EEC.

4. Knock downs by siRNA or shRNA are known for side effects. Using gRNA-based gene inactivation would strengthen key conclusions.

shRNA and siRNA are commonly used in tumor biology research. However, your suggestion is very reasonable, as sgRNA has a better effect for knocking out gene expression. Based on your suggestion, we designed and constructed sgRNA lentiviral vectors for HOXB9 and ODC1. Mice were subcutaneously transplanted with Ishikawa cells stably transfected with sgHOXB9-exon1 and sgODC1-exon4, and the results confirmed that knockdown of HOXB9 and ODC1 inhibited tumor growth. This has been added to Figure S8 in the revised manuscript.

Figure S8

5. It seems different cell lines were used for the different models (ISK=subcutaneous, AN3CA= foot pad). Is there a reason DMFO was not tested on subcutaneous tumor models?

Ishikawa is a highly differentiated EC adenocarcinoma cell line used in a mouse subcutaneous transplant tumor model. AN3CA is a lymph node metastasis-derived EC cell line, used in a footpad

injection lymph node metastasis mouse model. Based on your suggestion, we added a batch of mouse subcutaneous transplant tumor experiments, and found that DFMO inhibited tumor growth in vivo (Figure S8). Further, we also applied DFMO in patient-derived tumor cells (PTCs) (Figure 7h–j). Both experiments showed that DFMO inhibited the growth and progression of EC by inhibiting ODC1.

6. All tetrazolium-based assays for cell viability, such as Kit-8 (CCK-8) assay, may be impacted by different metabolic state of assayed cells. Direct cell count is required to exclude this possibility.

Thank you for your advice. We had not previously considered the impact of metabolic status on CCK8. We repeated this experiment by photographing and counting cells at different time points using an Operetta CLS High-Content Analysis System (PerkinElmer). This result has been added in Figure 2b in the revised manuscript. It was consistent with the previous results, indicating that OA indeed promotes EC cell proliferation. The metabolic changes caused by OA may have caused measurement error of CCK8. Overall, this does not affect our conclusion.

Figure 2b

7. Fig 3l: Differences in expression seem to be very minimal. Given low resolution of images it is impossible to exclude staining artifacts, different thickness of sections. Also, number of patients tested is very limited for this type of semi-quantitative studies.

Thank you for your suggestion. We re-selected representative photographs to reflect the positive correlation between the two molecules (Figure 3l). We also verified the correlation between HOXB9 and ODC1 by multiplex fluorescence immunohistochemical staining in EC tissue microarrays (TMAs)

(135 cores: 118 cancers + 17 peritumoral tissues) (Figure 3m). The previous IHC results were obtained in 17 cases, plus the 118 cases of TMA, both suggesting that HOXB9 and ODC1 expression were positively correlated.

Figure 3l and m

8. Supp. Figure 2 Please show quantifications of your western blots to support your claim that “only OA significantly upregulated ODC... (Supp. Fig. 2A)”. By eye, the WB does not convincingly show this. The same is true for a few of the WB in this figure and Figure 2.

We uniformly adjusted the contrast of Figure 2a (previous Figure S2a) WB band. We found that OA significantly upregulated ODC1. We also repeated the experiment in Figure 2g to make the WB band in which ODC1 was upregulated by OA more obvious.

Figure 2a

g

Figure 2g

9. Supp Figure 6,c,e images are of insufficiently high resolution to interpret histology, especially in case of lymph node metastasis. Some immune cells are known to express keratins. Additional staining, such as PAX2 would be more convincing. Also antibodies recognizing only human epithelial cells could be used.

We have replaced the images with higher resolution photographs. As the cell line used in this metastasis model was AN3CA, derived from lymph node metastasis, it has weak epithelial characteristics, a low degree of differentiation, and does not show significant expression of the epithelial markers keratin and PAX-2 (Figure S6h). However, histological analysis showed the tumor cells to have large nuclei, dark staining, and nuclear division (Figure S6h). We concluded that the tumor had metastasized to the lymph nodes. In addition, the inoculated cells were labeled with luciferase, and the fluorescent signal also represented the degree of metastasis (Figure 6l and o).

Figure S6h

Figure 6l and o

10. More thorough description of statistical assays is required. Nonparametric testing may be more applicable for many of the presented results.

The statistics expert in our hospital, Dr. Ren Wenhui, thoroughly reviewed the statistical methods used in this study and corrected the previous inaccuracies.

11. The study has several overstatements or erroneous conclusions: Some examples below:

a. The running title "Metabolic syndrome induces EC": The manuscript provides no experimental evidence that endometrial carcinoma is induced by metabolic syndrome.

a. We have changed the running title to "OA-HOXB9-ODC1 promotes EC progression" in the revised manuscript.

b. "This study demonstrated that increased polyamine metabolism is associated with induced EC progression" What is induced EC progression?

b. We have changed this description to “This study demonstrated that increased polyamine metabolism is associated with MS-promoted EC progression.”

c. "Hyperlipidemia in MS is the main driver of EC" How about known genetic drivers, such as PTEN mutations?

c. We have revised this sentence to “Hyperlipidemia in MS is one of the drivers of EC.”

d. "The OA-HOXB9-ODC1-polyamine axis fully explains the key mechanism by which MS leads to the progression of EC". Fully? No other possibilities? For example, LOA also significantly increases cell motility according to your data.

d. We have toned down our conclusion and removed the word “fully” in the revised manuscript. This article showed that the molecular mechanism of action of OA is involved in promotion of cancer by MS. LOA is also an unsaturated fatty acid similar to OA in its unsaturated characteristic and has similar effects. However, in subsequent experiments, LOA did not have the same effect on polyamine metabolism as OA, so we did not continue to focus on LOA in our study.

Minor concerns

1. Is there a direct correlation between oleic acid concentration and polyamine accumulation in serum of ECWMS patients?

We established a direct relationship between OA and polyamines at the cellular and molecular levels. Based on your suggestion, we tested OA and polyamines in the serum and tumor tissues of patients with EC. ECWMS increased serum polyamines, tumor tissue polyamines, and tumor tissue OA (Figure 1d, 1f, Figure 7g). Serum polyamine was associated with deep myometrial invasion (MI) (Figure 1e), and serum OA was elevated in the deep MI group, although the difference was not statistically significant due to the small number of samples (Figure 1e, right). Tissue polyamine was associated with deep MI, positive LVSI, hyperlipidemia, high grade, and recurrence, and positively correlated with tumor size (Figure S2b and c).

2. Are there any known mutations in HOXB9-ODC1 pathway in endometrial carcinoma?

We checked the distribution of ODC1 and HOXB9 mutations in different cancers in the COSMIC (Catalogue of Somatic Mutations in Cancer) database (<https://cancer.sanger.ac.uk/cosmic/>), as shown in the figure below. HOXB9 point mutation is the third most common in EC, while ODC1 point mutation has the highest incidence in EC.

3. Needs some editing to improve readability. For example:

- a. Lane 78: paragraph starting with “Tumor metabolic reprogramming is a hallmark for tumors”, includes a very long, run on sentence that can be confusing. Try to aim for more concise language.

We contacted a native speaker to polish our manuscript and improve the readability of the whole paper.

b. Lane 89: Clarify sentence: "They are important regulatory metabolites, not like glucose, lipids, and 90 amino acids are main "fuel" for tumor, which play crucial regulatory roles in multiple 91 physiological and pathological processes of the body, supporting for cancer metabolism and signal transduction^{25,26}"

We have simplified and clarified the text in accordance with your comments.

4. Please make sure you are consistent using "untargeted" or "non-targeted" metabolomics.

We have checked the full text to ensure consistency.

5. Lane 211 and other places: "knocking out HOXB9 with siHOXB9 shortened it (Figure 3d)" You cannot knockout gene with siRNA or shRNA. Only knockdown can be accomplished this way.

We have made changes to the relevant descriptions in the revised manuscript.

6. Section "OA in hyperlipidemia triggers an increase in ODC1 and polyamines" Line 174 and below. Ratios following fatty acid names need to be explained. What do they signify?

We have revised the description in the corresponding position in the original text. The ratios after fatty acids indicate the lengths of the carbon chains and the numbers of unsaturated bonds in the different fatty acids.

7. The same section: Clarify why you chose to treat with high glucose. This seemed to come out of nowhere in a section focused on testing different fatty acids.

The scientific question we raised concerned the molecular mechanism by which MS promotes EC. MS includes diabetes, hyperlipidemia, and hypertension. When screening for the importance of these three factors, hyperlipidemia was found to be the key regulatory factor. To exclude the regulation of polyamines by high glucose, we conducted WB experiments (Figure S2a) on the effects of high glucose on ODC1.

8. Figure 1c: The scale bar going from high to low is not clear that it correlates to the colors next to the chart indicating relative concentrations because no other colors on the gradient are present. Remove the bar and just state red is high and green is low.

We have removed the scale bar in the revised manuscript.

9. Figure 1d: more explanation of what is shown is needed. What do the different node colors mean?

What do the different node shapes mean?

We have added labels explaining the significance of the different node shapes and colors in Figure S1a in the revised manuscript.

Figure S1a

10. Figure 1i: typo, Are you addressing, 1h not 1g?

We have checked and revised the text in accordance with your comments.

11. Figure 1b: For improved readability, reorganize to have ODC1 and HOXB9 on the top row (since you mention them first in the text) with OAZ1 and Praja2 in the second row.

We have changed the order in Figure 7b in the revised manuscript.

12. Figure 2b: make clear that “DL” is your control condition and be consistent. Figure 2F has the control labeled as control, so we don’t know if there is an inherent difference between these.

We have clarified this point in the main text and the figure legends and kept it consistent in the figure.

We checked the marks in the figure and fixed similar problems in the revised manuscript.

13. Figure 2g: Are there any corresponding results for ISK cells? Since the two cell lines seem to show different responses to these conditions, such results are important.

We have added Ishikawa's results in Figure S3g. The experimental results were also consistent with our expectations.

Figure S3g

14. Figure 4a/b: Please show results for both cell lines or explain why they may be behaving differently, and you think one's results may be more supportive than the others.

In accordance with your suggestion, we repeated the experiment and supplemented the results in Figure S4a and b in the revised manuscript.

Figure S4

a

Figure S4a

Figure S4b

15. Supp Fig 4a: Add ISK label to the graph.

We have added the Ishikawa (ISK) labels in the appropriate places in the revised manuscript.

16. Figure 5f does not get cited.

We have added the citation in the revised manuscript.

17. Supp Fig 1a: Typo in Y axis label.

We have revised the text in accordance with your suggestion.

18. Supp Table 4: What are 0 and 1?

We have made changes and clarifications in the table in the revised manuscript.

Reviewer #2 (Remarks to the Author); participant in the Early Career Researcher co-review program

Reviewer #3 (Remarks to the Author); expert in metabolism:

The manuscript from Zhai et al focusses on the mechanisms linking hyperlipidemia to polyamine metabolism in cancer progression. By performing serum metabolomics of patients with endometrial

cancer (EC) and with or without metabolic syndrome (MS), authors have identified high levels of polyamine metabolites as markers of EC/MS. Increased polyamines were associated with increased expression of ODC1 in tumors, a key enzyme of polyamine synthesis. They then unveil a new mechanism by which lipids, and especially oleic acid (OA), prevents HOXB9 proteasomal degradation, therefore enhancing its stability. HOXB9 in consequence interacts with ODC1, which precludes its degradation by OAZ1 and promotes polyamine synthesis. Polyamines finally promote cancer cell proliferation and migration. Authors link this mechanism to the worsened survival prognosis of EC and MS patients. They finally bring an in vivo proof-of-concept of the OA-HOXB9-ODC1-polyamine axis involvement in tumor growth. Overall authors have identified a new metabolic pathway by which MS and hyperlipidemia promote polyamine synthesis and support tumor growth.

The manuscript is well written and with a lot of detail. Conclusions are in general convincing and supported by a large number of experiments performed in cellular models, mouse models of EC and patients tissues, as well as by different experimental approaches. Nevertheless, I would still need some clarifications and would recommend additional measurements to be done in order to strengthen the demonstration:

1) Concerning patients' data:

- Please combine tables S1 and S2 and directly compare the two groups (including calculating significance of differences) to highlight any difference between them. In table S4, please also present and compare the two groups MS and no MS (validation cohort).

Thank you for your comments. We have merged the two tables (original Tables S1 and S2) in the identification queue and performed statistical analysis of the differences between the two groups (new Table S1). In the validation queue (Table S3), we have also included a separate table for patients with or without MS and performed statistical analysis of the differences (new Table S4).

Table S1. The clinical characteristics of postmenopausal endometrioid EC patients with or without MS in the identification cohort.

EC without MS	EC with MS
---------------	------------

Variables	N	Percent	Variables	N	Percent	χ^2/t test	P
Total number	32		Total number	30			
Age			Age			18.4	<0.0001
<50	2	6.25%	<60	7	23.3%		
≥50, <60	22	68.75%	≥60, <70	18	60.0%		
≥60, <70	6	18.75%	≥70	5	16.6%		
≥70	1	3.13%					
N/A	1	3.13%					
Menopause			Menopause				
No	0	0.00%	No	0	0.00%		
Yes	32	100.00%	Yes	30	100.00%		
BMI			BMI			49.2 [#]	<0.0001 [#]
<24	24	75.00%	≥24, <28	13	43.3%		
≥24	8	25.00%	≥28, <32	11	36.6%		
			≥32	6	20.0%		
Hypertension			Hypertension			58.06*	<0.0001
No	32	100.00%	No	0	0.00%		
Yes	0	0.00%	Yes	30	100.00%		
Diabetes			Diabetes			58.06*	<0.0001
No	32	100.00%	No	0	0.00%		
Yes	0	0.00%	Yes	30	100.00%		
			Including metformin	13	43.3%		
			Including insulin	8	26.7%		

				Including other types		11	36.7%
				Dietary restriction or no treatment		6	20%
TG	Mean+SD	1.32+0.57		TG	Mean+SD	2.31+1.48	-3.32 ^a 0.002
HDL	Mean+SD	1.21+0.51		HDL	Mean+SD	1.05+0.23	0.57 ^a 0.122
Grade				Grade			3.57 0.168
	G1	7 21.88%		G1	13 43.3%		
	G2	14 43.75%		G2	11 36.6%		
	G3	11 34.38%		G3	6 20.0%		
Stage				Stage			8.55 0.287
	IA	24 75.00%		IA	21 70.0%		
	IB	5 15.63%		IB	6 20.0%		
	IIB	1 3.13%		IIIA	1 3.33%		
	IIIC	1 3.13%		IIIC	2 6.66%		
	IVB	1 3.13%					
LVSI				LVSI			0.18 0.667
	Negative	23 71.88%		Negative	23 76.6%		
	Positive	9 28.13%		Positive	7 23.3%		
LNM				LNM			<0.0001* 1
	Negative	29 90.63%		Negative	28 93.3%		
	Positive	3 9.38%		Positive	2 6.67%		
MI				MI			0.555 0.456
	No	26 81.25%		No	22		
	Yes	6 18.75%		Yes	8		

Cervical Involvement				Cervical Involvement				1.27	0.26
	No	29	90.63%		No	30	100.00%		
	Yes	3	9.38%		Yes	0	0.00%		
Ovary Involvement				Ovary Involvement					
	No	30	93.75%		No	28	93.33%	<0.0001*	1
	Yes	2	6.25%		Yes	2	6.67%		
Tumor Size				Tumor Size				4.79	0.091
	<2	10	31.25%		<2	12	40.00%		
	>2	15	46.88%		>2	6	20.00%		
	>4	7	21.88%		>4	11	36.67%		
					N/A	1	3.33%		
Endocrine Therapy				Endocrine Therapy				0.0001*	1
	No	30	93.75%		No	26	86.67%		
	Yes	2	6.25%		Yes	1	3.33%		
					N/A	3	10.00%		
Radio Therapy				Radio Therapy				2.536	0.111
	No	21	65.6%		No	25	74.2%		
	Yes	11	34.3%		Yes	5	25.8%		
Chemotherapy				Chemotherapy				0.969	0.325
	No	22	68.8%		No	17	56.7%		

	Yes	10	31.3%		Yes	13	43.3%		
Recurrence				Recurrence				0.008*	0.928
	Yes	3 ^a	9.4%		Yes	4 ^b	13.3%		
	No	29	90.6%		No	26	86.7%		
Death				Death				0.341*	0.559
	No	31	96.9%		No	27	90%		
	Yes	1	3.1%		Yes	3	10%		
DFS (days)				DFS (days)				-	0.524
	Mean+SD	3197.12+622.85			Mean+SD	4271.19+8512.9		0.642 ^a	
OS (days)				OS (days)				-0.648 ^a	0.520
	Mean+SD	3334.42+407.36			Mean+SD	4410.77+8641.6			

EC, Endometrial cancer. MS, Metabolic syndrome. BMI, body mass index. TG, Total Triglyceride.

HDL, High Density Lipoprotein. LVSI, lymph vascular space invasion. LNM, lymph node metastases.

MI, Myometrial invasion. DFS, disease free survival. # Fisher Precise significance. *Continuity

correction. at-test. bDeveloped hyperlipidemia during follow-up.

Table S4. The clinical characteristics of postmenopausal endometrioid EC patients with or without MS in the validation cohort.

EC without MS				EC with MS			χ^2	P
Variables	N	Percent	Variables	N	Percent			
Total number	28		Total number	22				
Menopause			Menopause					
	No	0	0.00%	No	0	0.00%		
	Yes	28	100.00%	Yes	22	100.00%		
Hypertension			Hypertension				50	
	No	28	100.00%	No	0	0.00%	<0.0001	
	Yes	0	0.00%	Yes	22	100.00%		

Diabetes				Diabetes			0.00%	50	<0.0001
	No	28	100.00%		No	0	0.00%		
	Yes	0	0.00%		Yes	22	100.00%		
							%		
				Metformin +/- others		122			
				Insulin only		2			
				Other only		3			
				Non or diet control		8			
Hyperlipidemia				Hyperlipidemia				50	<0.0001
	No	28			No	0			
	Yes	0			Yes	22			
Grade				Grade				2.09	0.654 ^b
	G1	11	39.30%		G1	7	31.80%		
	G2	12	42.90%		G2	8	36.40%		
	G3	5	17.90%		G3	6	27.30%		
Stage				Stage				2.59	0.621 ^a
	I	25	89.30%		I	18	81.80%	1	
	II	1	3.60%		II	0	0.00%		
	III-IV	2	7.10%		III-IV	4	18.10%		
LVSI				LVSI				0.31	0.75 ^b
	Negative	20	71.43%		Negative	14	63.64%	4	
	Positive	7	25.00%		Positive	7	31.82%		
	NA	1	3.57%			1	4.55%		
LNM				LNM				3.92	0.084 ^a
	Negative	27	96.43%		Negative	19	86.36%	2	
	Positive	0	0.00%		Positive	3	13.64%		
	NA	1	3.57%						
MI				Myometrial Infiltration				2.97	0.127 ^b
	No	21	75.00%		No	12	54.55%	5	
	Yes	6	21.43%		Yes	10	45.45%		
	NA	1	3.57%						
Cervical Involvement				Cervical Involvement				0.00	1.00 [#]
	No	26	92.86%		No	20	90.91%	0*	

	Yes	1	3.57%		Yes	1	4.55%		
	NA	1	3.57%		NA	1	4.55%		
Ovary Involvement				Ovary Involvement			0.00%	1.25	0.449 ^a
	No	27	96.43%		No	21	95.45%	3	
	Yes	0			Yes	1	4.55%		
	NA	1	3.57%						
recurrence or metastasis				recurrence or metastasis				0.56	0.451 [#]
	No	26	92.86%		No	18	81.82%	8*	
	Yes	2	7.14%		Yes	4	18.18%		

EC, Endometrial cancer. MS, Metabolic syndrome. LVSI, lymph vascular space invasion.

LNM, lymph node metastases. MI, Myometrial invasion. * Continuity correction. # Asymptotic significance. ^a Fisher Precise significance. ^b Precise significance.

-Among the most enriched metabolites in EC + MS patients, metformin is more than 7 fold increased, suggesting that patients with MS may receive anti-diabetic treatments. In my view, this is a very important information, which needs to be included in the tables (S1, S2, S4). Metformin is a powerful drug and fasting mimetic, and can have serious metabolic consequences on its own. Thus, in case patients EC+MS would be under metformin treatment, a necessary control experiment is to treat EC cancer cells with metformin and measure polyamine levels in order to rule out any effect of anti-diabetic treatment(s) on the pathway of interest.

We treated cells with in vivo concentrations^[5], which did not upregulate polyamine metabolism after treating EC cells with small doses of metformin (Figure S3b). The effect of high glucose treatment on the polyamine metabolic enzyme ODC1 was not as obvious as lipids (Figure S3a). Therefore, the effect of metformin on polyamine metabolism could be neglected. Further, we also searched the literature, which showed that treatment with metformin inhibited polyamine metabolism in colorectal cancer cells^[6]. In accordance with your suggestion, the information regarding anti-diabetic treatment has been incorporated into Tables S1 and S4.

Figure S3b

Figure S3a

Specific anti-diabetic treatment information of ECWMS patients in the Identification cohort

Metabolome ID	Diabetes treatment
M-1	DM 8 years; Metformin + Xiaohe Pills
M-2	DM10 years; Insulin
M-3	DM 6 months; Insulin
M-4	DM 3 years; Metformin
M-5	DM 7 years; Metformin + Glipizide
M-6	DM 12 years; Acarbose

M-7	DM 6 years; Irregular medication, metformin
M-8	DM 12 years; Insulin 6 years
M-9	DM 1 year; Insulin
M-10	DM 10 years; Insulin + Glucobay + Metformin
M-11	DM 4 years; Metformin + repaglinide
M-12	DM 3 years; Untreated
M-13	DM 4 days; Untreated
M-14	DM 5 years; Acarbose
M-15	DM 5 years; Diet control
M-16	DM 6 months; Acarbose
M-17	Diagnosis of DM during hospitalization
M-18	Diagnosis of DM based on glycated hemoglobin before surgery
M-19	DM 5 years; Insulin
M-20	DM 23 years; Insulin
M-21	DM 2 years; Metformin
M-22	DM 6 years; Acarbose
M-23	DM 2 years; Metformin + Sitagliptin
M-24	DM 10 years; Gliquidone + Metformin
M-25	DM 15 years; Metformin
M-26	DM 12 years; Insulin + Metformin + Pioglitazone + Saxagliptin
M-27	DM 3 years; Metformin
M-28	DM 12 years; Metformin
M-29	DM 10 years; Xiaokeyiangtang Capsule + Gliclazide Sulfonylurea + Metformin
M-30	Diagnosis of Diabetes after admission

- The paragraph starting from line 147 speaks of “correlations” but no correlation analysis has been performed, so the results do not justify the conclusion. Either this needs to be re-phrased, or a different analysis and representation of the data is needed (i.e. correlation analyses) to justify the conclusion.

This section discusses association analysis of enzymes and metabolites, not a correlation, and we have

adjusted the wording accordingly in the revised manuscript. Given the limited space in the main figure for our new results, we have added them to Supplementary Figure S1b.

- Regarding Figure 1h-k: Metabolomics has been performed from patients' serum. Nevertheless, this is not representative of the tumor itself and may be the consequence of metabolic alteration in other tissues such as liver. I would therefore encourage the authors to confirm the increase in polyamine levels in patients' tumors.

Thank you for your very good suggestion. We checked the remaining samples frozen in liquid nitrogen and found 18 vs. 10 cases (MS+ vs. MS-). We sent them for LC-MS to detect polyamines and the results were as expected. Polyamines and putrescine were elevated in patients with ECWMS. The results are shown in Figure 1e.

Figure 1e

- In Fig 7, I would rather recommend to perform lipidomic analysis to confirm the increase in OA levels rather than using an unspecific approach.

We agree that it is crucial to confirm the increase of OA in tissues. We performed OA determination in the 18 MS+ vs. 10 MS- frozen tissues mentioned in the response to the previous question. OA is poorly soluble in polar solvents and difficult to separate by chromatography. We established the corresponding tissue lipid extraction method and derivatization chromatography detection method. The results were consistent with expectations. The results are shown in Figure 7g.

Figure 7g

-In Fig 7f-I, I would also be interested in the effect of high and low expression in MS and non-MS patients and in patients with or without hyperlipidemia.

As there is no comprehensive diagnosis of MS in the TCGA database, we used BMI to indicate MS, suggesting that both HOXB9 and ODC1 can predict poor prognosis. As the mortality rate of EC is low, a prognostic relationship could not be established in our patients and we expect to establish such a relationship when we have data from more than 1000 cases. We have shown that elevated HOXB9 and ODC1 predict poor prognosis. It will take a considerable amount of time to ascertain whether HOXB9 and ODC1 can indicate an even worse prognosis in patients with MS and hyperlipidemia.

- Also in Fig. 7, there seems to be a significant effect on Praja2 levels in the quantification (7b, $p < 0.01$) but the bars are at exactly the same height, so this is unlikely. Also, the “representative” picture (7a) shows reduced Praja2 whereas the quantification shows increased – or unchanged(?) levels.

We increased the number of patients undergoing multiplex fluorescence immunohistochemistry and performed statistical analysis. Representative images and statistics of mean fluorescence intensity have been updated in Figure 7a and b. Praja2 exhibited no statistically significant difference between the two groups.

REDACTED

Figure 7

- Also in Fig. 7a-e, as I understand it, data from only 1 patient per group is shown (MS+#3 and MS-#1). While this may be fine for representative images as in 7a and e, data of at least 5 patients / group needs to be quantified here in order to make any meaningful conclusions.

As mentioned in the response to the previous question, we have added multiplex immunofluorescence histochemical staining data from 11 patients' samples (6 MS+ cases and 5 MS- cases). The results of the new statistical analysis have been updated in Figure 7b.

We also added fresh frozen samples from newly-admitted patients (hospitalized in December 2024) for Raman detection. The latest statistical analysis has been added in Figure 7 d and e and Supplemental Figure 9 a–c. The results were in line with our expectations. Specifically, in patients with ECWMS, the peaks corresponding to lipids showed an increase.

Further, as stated in the response to the previous question, we measured the OA content in frozen tissues from 18 MS+ vs. 10 MS- in the identification queue (Figure 7g).

Figure 7b

Figure 7d and e

Figure S9 a-c

Figure 7g

2) Concerning in vitro mechanistic studies:

- Please specify what DL, which is often used as a control for OA treatment, means.

To exclude the influence of fatty acids in serum, we used defatted serum to remove lipids, which has been indicated as DL (delipidated) in the revised manuscript and figure legends.

- In general, I regret that the authors mostly provide qualitative instead of quantitative data for many experiments. Some blots are N=1 per condition and the effect is not always clear by eye. In order to strengthen the claims, it will be important to provide a quantification and statistical analysis of the blots, especially for Fig 2e, 2j, 3j, 3k, 4a, 5b, S2a, S2b, S3a.

Repeated experiments have been performed and figures have been replaced. The overall contrast of some figures has been adjusted. Quantitative and statistical analyses have also been performed.

Fig2e

Fig2g

Fig2j

Fig2l

Fig3j

Fig3j

Fig3k

Fig3k

uniformly adjust contrast

Fig4a

Fig4a

Highlighted the groups that need attention.

Fig5b**Fig5b**
uniformly adjust contrast**FigS2a****Fig2a****FigS2b****FigS3b****FigS3a****FigS4a**
- Effect of OA on polyamines does not seem very stable (Fig 2d: unchanged at 48h in AN3CA cells) and transient (unchanged at 48h and decreased at 72h in ISK cells). It is therefore puzzling that the authors still see an effect on cancer cell migration in S2g despite no increase in polyamines. Could the authors comment on this?

The knockdown experiments discussed in this article all used cells harvested after 72 h, because the knockdown effect of siRNA was more obvious at this time point. However, this was not the same timing as that of the addition of drugs. OA was added to Ishikawa cells 24 h before harvesting, and to AN3CA cells 48 h before harvesting. This has been clarified in the figure legends. To avoid misunderstanding, we deleted the figures showing the polyamine content in Ishikawa cells at 48 hours

and 72 hours after the addition of OA (Figure S3e). It is not clear why it decreased at 48 and 72 h. We also repeated the experiment to measure the levels of polyamines following addition of OA to the two cell lines at different time points. The results showed that the levels were also high at 48 h in AN3CA.

Figure S3e

New experiment

Same goes for Fig 4b where OA-siC has no effect on ODC1 stability compared to DL-siC. In Fig 4c and 4d, the decrease in cell proliferation and migration in OA + siODC1 is rescued by HOXB9 overexpression, however this is not mirrored by change in polyamine levels (Fig 4e and 4f), questioning the concept.

OA can be seen to stabilize ODC1 very obviously at 120 min (Figure 4b, left), indicating that OA can stabilize the half-life of ODC1, causing its gradual accumulation. We repeated the siODC1+Flag-HOXB9 polyamine metabolite rescue experiment you mentioned. The results showed that siODC1+Flag-HOXB9 rescued polyamine metabolites (Figure 4e, Supplementary Figure S5d).

Figure 4b

Figure 4e

Figure S5d

Down the same line, it is not very logical that effects on ODC1 protein stability are observed at the 10 minutes timepoint (Fig. 2f, bottom) but an increase in polyamines only becomes apparent at 24h. Similarly, OA effect on HOXB9 stabilization is only observed after 24 hours, but it should stabilize

ODC, which has a completely different kinetic. Authors should be much more careful in their conclusions, because the stability assays do not actually support the conclusions of this manuscript. Or, these kinetics need to be better explained.

We found that OA can promote the interaction between HOXB9 and ODC1 (Figure 3f and Figure 5e), so it can stabilize ODC1 before the level of HOXB9 increased.

These observations suggest that OA helps form a complex that stabilizes HOXB9-ODC1, resulting in the co-stabilization of the two molecules. None of the three are indispensable. Without OA, HOXB9 does not interact with ODC1 (Figure 3f); without HOXB9, OA cannot stabilize ODC1 (Figure 4b); without ODC1, HOXB9 will not be upregulated by OA (Figure 4a). The three interact and stabilize each other. OA may provide a phase separation environment, preventing the binding of the corresponding E3 ligase or degradation enzyme to HOXB9 and ODC1, thus playing a physical protective role. The interaction between molecules is faster and more sensitive than the amount. Previously, we focused on the OA-HOXB9-ODC1 axis, because ODC1 is the main reason for regulating the production of polyamines and ultimately performs functions, so we listed HOXB9 as an auxiliary member of the regulatory axis. In fact, however, OA forms a complex that interacts and stabilizes HOXB9-ODC1, and there is positive feedback. ODC1 and polyamines can both show positive feedback and stabilize HOXB9 (Figure S4 f and g, Figure 5b), ultimately leading to changes in quantity and effects. This has also been explained further in the Discussion in the revised manuscript.

Figure 3f

Figure 5e

Figure 4b

Figure 4a

Figure S4 f and g

Figure 5b

- It is not logical that the authors conduct the experiment on ODC1 degradation (S3a) with the Flag overexpression construct, which obviously is unresponsive to OA (compare to Fig. 2g red vs orange bars) – this needs to be repeated using the endogenous protein that responds to OA. Same for Fig. 5f. This is because the exogenous overexpression OAZ1 is dominant. The transfection of OAZ1 can degrade ODC1, so the latter two columns show lower values or no change compared to the first. OA has a stabilizing effect on ODC1, but this effect is masked by OAZ1. The strong degradation of a large amount of OAZ1 is dominant, and OA cannot prevent the binding and degradation of OAZ1 and ODC1. Figure 2g is more obvious without the effect of exogenous OAZ1. For Figure 5f, we finally selected Ishikawa cells because they exhibited a more pronounced effect and were more representative in demonstrating the up-regulation of ODC1 by OA (as indicated by the input

Flag band).

Figure S4a

Figure 2g

Figure 5f

- How did the authors start looking at HOXB9, 13? This is not clearly introduced in the paragraph starting on line 204. Why did the authors assess HOXB13 in Fig 3b? Since HOXB13 also enhances ODC1 stability, what was the rationale to focus exclusively on HOXB9 later?

We discovered the HOXB family by transcription factor prediction of ODC1. HOXB9 and HOXB13 are two representative members of the HOXB family. However, HOXB13-mediated upregulation of

ODC1 is different from that of HOXB9, which requires further study. However, HOXB13 was not within the scope of this article, so we deleted the discussion from the manuscript.

Figure 3a and b

- line 215: Exploring further, we observed an endogenous interaction between ODC1 and HOXB9 in EC cells treated with OA, which was notably weaker without OA treatment (Supplementary Figure 3d, Figure 3f) -> only data for +OA is shown in Figure 3f - if authors want to make a claim about “stronger” or “weaker” interaction (which is difficult with a semi-quantitative method such as co-IP anyway), they need to show +/- OA conditions on the same blots.

We have changed the treatment conditions and WB upsampling method so that the different conditions are shown in the same blot. The latest results have been replaced in Figure 3f.

Figure 3f

- Figure 5a: The western blot (left) does not show the same as the quantification (right). If normalized to time point 0 for both groups, HOXB9 degrades over time with OA but not without OA.

We repeated the experiment. The latest blot is shown in Figure 5a, which makes the result more obvious. HOXB9 in the OA group had not yet decayed at 15 h.

Figure 5a

3) Concerning in vivo experiments in preclinical models:

- I am surprised that 2-3 months 60% HFD only led to a mild increase in body weight gain and lipid accumulation. In general, I am missing the validation of MS development in the HFD-fed animals. In addition to TAG levels, I would recommend to also measure free fatty acids in the plasma. I would also recommend to do a lipidomic analysis of mouse plasma and/or tumor to prove the enrichment in OA in HFD-fed mice. Also, could the authors provide the total fat and lean mass from Fig 6e ?

Nude mice are not as sensitive to obesity as C57BL/6 mice. The increases in body weight and blood lipids in nude mice after feeding a high-fat diet are not as obvious as in C57BL/6 mice, but are still significant. Given the lack of mouse EC cell lines, nude mice were used as obesity models and tumor-bearing experiments.

Previously, we detected polyamines and oleic acid (OA) in the serum and tumor tissue samples of patients. We found that tissue OA, tissue polyamines, and serum polyamines were associated with MS (Figure 1e, Figure 7g), whereas serum OA showed a weaker correlation. Therefore, here we mainly supplemented the OA detection in the tissues of mouse xenografts by LC-MS (Figure 6k). The results showed that OA was increased in tumor tissues of the HFD group. OA was reduced in transplanted tumors transfected with shHOXB9 or shODC1.

In accordance with your suggestion, we performed quantitative and statistical analyses of the adipose

tissue volume on MRIs and found that HFD caused fat gain in mice (Figure 6e). However, MRI is not sensitive to the discrimination of lean muscle, so statistical analysis was not performed. We repeated the nude mouse subcutaneous tumor transplantation experiment and performed MRI and body composition analysis on the new batch of mice. Based on the results of body composition analysis, we calculated the lean muscle mass and total fat mass. The results were consistent with the previous experiment, with increased body fat in the HFD group (Figure S8 b and c).

Figure 1e

Figure 7g

Figure 6k

Figure 6d and e

Fig. S8

Figure S8

- In the second model, I am not really convinced about the difference in body weight shown in Fig S6a.
 Can the authors also provide quantitative data on body composition (fat and lean mass) and plasma free

fatty acids and TAG levels?

As mentioned in the response to the previous question, nude mice are not very sensitive to obesity, and the difference in weight gain did not reach 20%. However, the difference between the two groups was statistically significant. In accordance with your suggestion, we measured lipids in the serum of lymph node metastasis model mice, and the results showed that the increases in TC, LDL-C, and glucose were statistically significant (Figure S7c). Due to technical limitations, we did not measure plasma free fatty acids. Figure 6d shows an MRI of lymph node metastasis model mice, and the results of statistical analysis of adipose tissue content have been added (Figure 6e). Figure S8 shows MRI and fat content analysis of a new batch of subcutaneous tumor mice. The results all showed that the body fat content increased in the HFD group.

Figure S7c

- In Fig S6b, I am not really convinced about the effect on shRNA on ODC1 and HOXB9 levels in the HFD group. Could the authors quantify the different protein levels to perform statistical analysis? It is unclear which band should be HOXB9, as no band disappears in the knockdown group specifically. As a minimum, the correct band must be indicated vs. unspecific bands, but actually this blot shows that the knockdown of HOXB9 was not very successful. Similar, but slightly more convincing, for ODC1. HOXB9 has been marked in the band of Figure S7d (previous Figure S6b). As high fat increased the stability and upregulation of HOXB9 and ODC1, the knockdown effect was eventually weakened in WB, which was consistent with the mechanism discussed in the paper. We further repeated the mouse subcutaneous tumor transplantation experiment using cell lines stably expressing sgHOXB9 and sgODC1, and the results also showed that knocking out HOXB9 or ODC1 significantly inhibited the formation and growth of subcutaneous tumors (Figure S8d and e).

Figure S7d

- In general, I find the effect of shODC1 on tumor growth very convincing but I am not fully convinced about the link between hyperlipidemia, polyamine metabolism and tumor growth. Indeed, despite increased polyamine levels in HFD versus LFD, there is no difference in tumor volume and tumor weight (6g).

High fat promotes tumor progression, which was the basic model of our experiment. The gross figure showed that the tumors were smaller in the LFD-NC group than the HFD-NC group (Figure 7f). We reviewed the data and found that we had accidentally misreported the tumor mass in the LFD-NC group, resulting in no significant difference. We corrected the data and repeated the statistical analysis. The results showed that the tumor masses were significantly different between the HFD-shNC and LFD-shNC groups (Figure 6h). To better illustrate this, we also repeated a batch of subcutaneous tumor model experiments. Cell lines stably transfected with sgRNA were used. The tumor growth curve and gross figure also showed that the tumors were significantly larger in the HFD group than the LFD group (Figure S8d and e).

Figure 7

-Also, shHOXB9 led to a stronger inhibition of polyamine synthesis than shODC1 but had no significant effect on tumor growth. So the association between polyamine levels and tumor growth in vivo is not really clear, could the authors comment on that?

We analyzed the polyamine components in mouse serum and tissues at the same time. Our findings revealed that tissue polyamines exhibited a closer correlation with tumor size (Figure 6j, Figure S7f). This observation implies that the relationship between serum and tumor growth is not as strong as that between tissues and tumor growth. This is also consistent with the patients' results. The correlation analysis between polyamines in patients' serum and tissue samples and their clinical pathological features further demonstrated that serum polyamines were associated with tumor invasion, such as myometrial invasion (MI) (Figure 1f, Figure S2a). In contrast, polyamines in tumor tissues were not only correlated with invasiveness indicators like MI and lymphovascular invasion (LVSI) but also strongly associated with tumor size (Figure S2b).

Figure 6j

Figure S7

Figure 1f

Fig. S2

Figure S2

Minor comments:

- N of each experiment needs to be reported in figure legends

Thank you for your suggestion, which has been addressed in the revised manuscript.

- Statistical test used for each graph needs to be specified. (for each subfigure, not simply for the whole figure).

Thank you for your suggestions, which have been addressed in the revised manuscript.

- Please, define each abbreviation in the figure legends (for instance abbreviations of the different lipid species)

Thank you for your suggestions, which have been addressed in the revised manuscript.

- Some proteins are presented in the WBs but not mentioned in the text, please provide the rationale of assessing them. For instance: EZH2 and Twist (Fig 2j, S6b), ZEB1 (Fig 4a).

Thank you for your suggestion, which has been addressed in the main text. EZH2 is a target of HOXB9, and was used to validate the effects of HOXB9. Twist and ZEB are EMT-associated transcription factors used to assess the effects of the OA-HOXB9-ODC1 axis on migration.

- Please note that the plural of serum is “sera”, not “serums”

Thank you for your suggestion, which has been addressed in the revised manuscript.

- Please change the color code from red-green (e.g. Fig. 1b, 1c) to red-blue or similar so that people who are color blind can also read the differences.

Thank you for your suggestions. The colors in Figures 1b and 1c have been changed to the red-blue (RGB) mode. Besides the fluorescence image, other red - green codings have also been modified.

Figure 1

Figure 1
Panel a) by figdraw.com

- Line 159, specify which "tissues"

Thank you for your suggestion. The text has been edited accordingly in the revised manuscript.

- Scale bars missing for Fig. 3i, right-

Thank you for your suggestion. We have added a scale bar in the enlarged photo and supplemented description in the figure legends in the revised manuscript.

- A general recommendation: For some figures, the principle “less is more” applies. For example Figures 4c-f: It is almost impossible to read all the different conditions and situations here, and for clarity, a focus on the key message would strengthen, rather than weaken, the message.

Thank you for the suggestion. The text has been edited accordingly in the revised manuscript. The groups requiring attention in Figure 4 are indicated by red symbols and black boxes.

Figure 4

Figure 4

- Lines 288, 289: here, authors obviously refer to figures S5e and 5f, not S3e and 3f

Thank you for your suggestion. The text has been edited accordingly in the revised manuscript.

- Line 320: "Moreover, knockdown of HOXB9 and ODC1 could 321 inhibit the increase of polyamine metabolites in serum induced by HFD (Figure 6i)." The conclusion is not fully supported by the data.

Also, how can you do statistics on n=1 (spermine, black bar)

Thank you for your question. We have corrected the data for n = 1 in Figure S7f (previous Figure 6i) in the revised manuscript.

Figure S7f

- When first introducing DFMO, it should be specified that it is an ODC1 inhibitor

Thank you for your suggestion. This point is explained in the article. We also applied DFMO in patient organoid models, showing that chemotherapy combined with DFMO can improve chemotherapy resistance (Figure 7h-j).

Figure 7 h-j

- I was unable to find the ISK cell line under this name in the ATCC database. Did they have a different source, or are they sold under a different name at ATCC?

Thank you for your suggestion. The full name Ishikawa has been used in the revised manuscript.

Reviewer #4 (Remarks to the Author); participant in the Early Career Researcher co-review program

- [1] Muñoz-Esparza NC, Latorre-Moratalla ML, Comas-Basté O, Toro-Funes N, Veciana-Nogués MT and Vidal-Carou MC. Polyamines in Food. *Front Nutr.* 2019, 6: 108.
- [2] Magnes C, Fauland A, Gander E, Narath S, Ratzer M, Eisenberg T, Madeo F, Pieber T and Sinner F. Polyamines in biological samples: Rapid and robust quantification by solid-phase extraction online-coupled to liquid chromatography - tandem mass spectrometry. *Journal of Chromatography A.* 2014, 1331: 44-51.
- [3] Yin S, Yu Y, Wu N, Zhuo M, Wang Y, Niu Y, Ni Y, Hu F, Ding C, Liu H, Cheng X, Peng J, Li J, He Y, Li J, Wang J, Zhang H, Zhai X, Liu B, Wang Y, Yan S, Chen M, Li W, Peng J, Peng F, Xi R, Ye B, Jiang L and Xi JJ. Patient-derived tumor-like cell clusters for personalized chemo- and immunotherapies in non-small cell lung cancer. *Cell Stem Cell.* 2024, 31(5): 717-733 e718.
- [4] Lu KH and Broaddus RR. Endometrial Cancer. *N Engl J Med.* 2020, 383(21): 2053-2064.
- [5] Ma T, Tian X, Zhang B, Li M, Wang Y, Yang C, Wu J, Wei X, Qu Q, Yu Y, Long S, Feng J-W, Li C, Zhang C, Xie C, Wu Y, Xu Z, Chen J, Yu Y, Huang X, He Y, Yao L, Zhang L, Zhu M, Wang W, Wang Z-C, Zhang M, Bao Y, Jia W, Lin S-Y, Ye Z, Piao H-L, Deng X, Zhang C-S and Lin S-C. Low-dose metformin targets the lysosomal AMPK pathway through PEN2. *Nature.* 2022, 603(7899): 159-165.
- [6] Zhang T, Hu L, Tang JF, Xu H, Tian K, Wu MN, Huang SY, Du YM, Zhou P,

Lu RJ, He S, Xu JM, Si JJ, Li J, Chen DL and Ran JH. Metformin Inhibits the Urea Cycle and Reduces Putrescine Generation in Colorectal Cancer Cell Lines. *Molecules*. 2021, 26(7).

Reviewer #1 (Remarks to the Author):

The authors sufficiently addressed the majority of previous reviews. However, the manuscript could be improved further by addressing comments below.

Major Comments:

1. The text still has a few statements that oversell the findings of this paper.

a. Line 125: “The OA-HOXB9-ODC1-polyamine axis ... leads to the progression of EC”.

The authors’ data shows that it “supports” progression, but leading is a very strong conclusion.

The expression has been revised.

121	The OA-HOXB9-ODC1-polyamine axis explains the key mechanism by which MS
122	promotes the progression of EC.↵

b. Line 502: “The enhancement of ... is an EC-dependent mechanism of MS and is highly significant in EC”. Could the authors state this without also analyzing healthy post-menopausal women with MS to see if they also have an upregulation of this pathway?

The expression has been revised. Our previous statement was not sufficiently rigorous.

515	metabolism reprogramming and the systemic metabolic environment. The OA-HOXB9-
516	ODC1-polyamine pathway was significantly enhanced in ECWMS patients. Whether this
517	phenomenon is specific to EC warrants further investigation, and its applicability to other
518	metabolism-related tumors also remains to be explored. If this molecular mechanism is not

2. The authors’ validation of lymph node metastasis with the anti-PAX2 IHC stain in SF 8j is not very convincing. Perhaps they could try to use some of human-specific markers, such as human-specific GAPDH, Ku80, hMito and/or Alu?

Thank you for the suggestion! We used a human-specific CK18 antibody (Santa Cruz sc-6259) for immunohistochemistry. This antibody specifically recognizes human cytokeratin 18 without cross-reactivity to rodent homologs, and is highly expressed in the endometrial cancer AN3CA cells, while being almost undetectable in lymph node stromal cells and normal mouse tissues. The results showed nest-like distribution of CK18-positive cells in metastatic lymph nodes, which have been supplemented in the **Supplementary Figure S8m**. Your question has been answered satisfactorily.

3. It is unclear why the authors have decided to use different timepoints to analyze the two different cell lines in Figure 2. Perhaps the inherent differences between the cell lines – based on their derivation – could be useful to mention and explain why they may be treated/reacting differently in these experiments. The corresponding Results for Figure 3D is not shown for Ishikawa cells.

Regarding the selection of different time points in Figure 2: It has been supplemented in the text that due to differences in biological characteristics and malignancy degrees, Ishikawa cells (derived from G3 adenocarcinoma) and AN3CA cells (derived from lymph node metastasis) exhibit distinct metabolic response times to OA. Polyamine metabolomics data show that after treatment at different time points, Ishikawa cells demonstrate more significant responses at 12-24h of OA treatment (Figure S3e), while AN3CA cells show more obvious responses at 36-72h (Figure 2f). Therefore, corresponding time points were selected for subsequent experiments based on these results.

212 We found that OA significantly promoted proliferation in EC cells and organoid (Figure
213 2b and c) and promoted the motility and migration of EC cells (Figure 2d and e). Further,
214 we found that OA upregulated polyamine metabolites and ODC1 in a time-dependent
215 manner (Figure 2f and g; Supplementary Figure 3e and f). However, due to differences in
216 biological characteristics and malignancy degrees, Ishikawa cells (derived from G3
217 adenocarcinoma) and AN3CA cells (derived from lymph node metastasis) exhibit distinct
218 metabolic response times to OA. Ishikawa cells demonstrate more significant responses at
219 12-24h of OA treatment (Supplementary Figure 3f), while AN3CA cells show more
220 obvious responses at 36-72h (Figure 2f). Therefore, corresponding time points were
221 selected for subsequent experiments based on these results. However, the mRNA level of

Regarding the absence of Ishikawa cell results in Figure 3D: The basal expression level of HOXB9 in Ishikawa cells is extremely low (with a qPCR CT value of approximately 33), making knockout difficult. To strengthen our conclusions, we supplemented experiments using HEC-50B cells, which are derived from human endometrioid adenocarcinoma Grade 3 (G3) and originate from ascitic metastatic lesions, with high HOXB9 expression. The results showed that knockdown of HOXB9 in HEC-50B cells shortened the half-life of ODC1. The relevant data is presented in Figure 3d, and the original bands along with statistical analyses are provided in the source data file. Your question has been satisfactorily addressed.

4. Figure 3l: The authors' findings state that there is a correlation of expression of ODC1 and HOXB9 in patients with LVSI or LNM. Comparing between the two groups would be more convincing if the authors are tying this axis to EC progression in MS+ patients.

Thank you for your insightful question, which we consider highly important. We performed Spearman correlation analyses between the IHC scores of ODC1 and HOXB9 and clinicopathological parameters including stage, grade, BMI, diabetes, hypertension, and LNM in the 17 cases of Figure 3l. The results showed that ODC1 was positively correlated with grade ($p=0.01$).

In the tissue microarray cohort of 118 patients in Figure 3m, there was only 1 patient with both metabolic syndrome (MS) and metastasis. To establish the correlation between MS and

HOXB9/ODC1 as well as disease progression, we will need to expand the sample size in future studies. In this figure, our focus was to reveal the positive correlation between HOXB9 and ODC1, which was confirmed in both the 17 cases (Figure 3l) and 118 cases (Figure 3m). In Figure 6, we demonstrated that the expression of ODC1 and HOXB9 is increased in MS patients, and in obese patients, high expression of ODC1 and HOXB9 was associated with poor prognosis (Figure S9).

5. It is interesting that the results in Figure 6c and Supplementary figure 7c are so different. Shouldn't the two models be consistent between the two models if they both had an LFD and HFD group?

Thank you for your question. The purpose of this experiment was to demonstrate that a high-fat diet (HFD) induces dyslipidemia in mice, which is supported by both Figure 6c and Supplementary Figure 7c, as both show that HFD led to varying degrees of dyslipidemia. The differences between the two figures arise from three factors: ① Different collection time points: Figure 6c presents serum data from the subcutaneous tumor model after 3 months of HFD; Supplementary Figure 7c shows serum results from the lymph node metastasis model at the endpoint (3 months after tumor implantation, following 3 months of prior HFD). ② Different testing institutions: The first batch of samples was analyzed at the Testing Center of Peking University Health Science Center (biochemical analyzer, BS-450, Myriad), while the second batch was tested by Beijing Jinglai Technology Co., Ltd (biochemical analyzer, BS-240VET, Mindray). ③ Different sample sizes: The first group had fewer samples.

Notwithstanding these differences, the core findings remain consistent: both figures clearly confirm that HFD induces dyslipidemia in mice, validating the experimental hypothesis.

Thank you.

Also, in the caption for Supplemental Figure 7c, the authors do not mention for which subgroup of mice these measurements were done.

Thank you for your careful reminder. We have supplemented the relevant description in the caption of Supplemental Figure 7c to specify the subgroup of mice for which these measurements were performed. We have also reviewed and supplemented the captions of other figures to ensure clarity and completeness.

(c) Serum levels of glucose, TG, TC, LDL-c, and HDL-c in LNM model mice after 2-month diet induction (LFD: n=5; HFD: n=5) measured by biochemistry analyzer. Data: mean \pm SD. *p < 0.05, **p < 0.01, ***p < 0.001, ****p < 0.0001 Student's t-test. [↵]
--

6. Line 363: “we found that the tumor size was closely related with the putrescine content in

the tissues”. There is no analysis for this, and based on comparing Fig. 6f and 6j, the conclusion is not convincing.

Thank you very much for your question, which has helped us identify this oversight. We have supplemented the correlation analysis between tissue putrescine content and tumor volume, and the results confirm a positive correlation between them. This analysis has been added to the Figure 6l to support the conclusion, and we appreciate your valuable input that has improved the rigor of our study.

7. Supplementary Figure 9: based on the differences BMI groupings between the authors' ODC1 (BMI>40) and HOXB9 (30>BMI>34.9) analysis, it is not convincing that there is overlap in patients with high ODC1 and HOXB9. This weakens the authors' conclusion in line 426, "High levels of HOXB9 and ODC1 are closely related to patient weight and predict poor outcomes". What happens when the authors use the same patient groups?

Thank you for your insightful question. In response to comments from another reviewer, we presented survival curves analyzing the prognostic impact of varying HOXB9 and ODC1 expression levels under conditions of metabolic syndrome (MS), specifically in obese or severely obese subgroups. Taking your suggestions into account, we have re-described this part of the results and revised the imprecise conclusion, for which we are very grateful.

This figure aims to emphasize that the three factors in our study—HOXB9, ODC1, and obesity—each predict poor prognosis. To clarify, the two survival curves (Supplementary Figure 9d and 9e) were intended to separately illustrate that in obese patient subgroups (with different BMI cut-offs), high expression of either HOXB9 or ODC1 is associated with poor prognosis. The subsequent survival analyses (Supplementary Figure 9f and 9g) were designed to demonstrate the association between obese status itself and poor prognosis in the context of high HOXB9 or ODC1 expression. Collectively, these analyses consistently validate the

prognostic significance of each factor, reinforcing the robustness of our findings.

429 We performed survival analysis using profiles of *ODC1*, *HOXB9*, and BMI, combined
430 with survival data of EC patients retrieved from The Cancer Genome Atlas (TCGA)
431 database. In patients with extreme obesity ($BMI \geq 40$), high *ODC1* expression was
432 associated with poorer OS compared to low *ODC1* expression ($HR = 2.32$; $p = 0.03$)
433 (Supplementary Figure 9d). In obese patients ($BMI \geq 30$ and < 34.9), high *HOXB9*
434 expression was linked to worse OS than low *HOXB9* expression ($HR = 3.23$; $p = 0.007$)
435 (Supplementary Figure 9e), indicating that high *HOXB9* and high *ODC1* expression in
436 patients with obesity are associated with poor prognosis.⁴

437 No statistically significant survival difference was observed between extremely obese
438 patients and normal-weight patients with high *ODC1* expression (Supplementary Figure
439 9f). However, among patients with high *HOXB9* expression, obese patients had a worse
440 prognosis than normal-weight patients ($HR = 1.87$; $p = 0.04$) (Supplementary Figure 9g),
441 suggesting that obesity is associated with poor prognosis.⁴

Supplementary Figure 9d-g: TCGA-UCEC patient data re-analyzed by our team via grouping

by HOXB9, ODC1 and BMI for survival curve analysis.

8. Regarding Figure 7j – these results are only from PTCs derived from a single patient sample? Considering individual patient variability, it is unclear how strong conclusions can be drawn from these results.

Thank you for your attention to the rigor of our experiments. It is true that the preliminary validation in the original study was based on PTCs derived from a single patient sample. To further enhance the reliability of our conclusions, we have supplemented drug sensitivity assays using PTC samples from 4 additional patients. All 4 samples exhibited varying degrees of improved chemosensitivity, consistently demonstrating that ODC1 inhibitors can effectively enhance chemotherapy sensitivity. The results indicate that DFMO significantly improves sensitivity to paclitaxel and carboplatin. These supplementary data have been added to Figure S10, which effectively mitigates biases arising from individual variability and supports the potential value of DFMO in improving drug sensitivity.

Fig. S10

Table S8. The clinical characteristics of PTCs received drug resistance experiments.

Patient characteristics	N	Percent
Number	5	100%

Age			
	Mean±SD	42±10	
MI			
	None	2	40.00%
	Superficial	2	40.00%
	Deep	1	20.00%
Grade			
	G1	4	80.00%
	G2	1	20.00%
LNM			
	Negative	5	100.00%
	Positive	0	0.00%
LVSI			
	Negative	5	100.00%
	Positive	0	0.00%
Stage			
	I A	3	60.00%
	I B	1	20.00%
	IIIB	1	20.00%
hyperlipdemia			
	Negative	3	60.00%
	Positive	2	40.00%
Diabetes			
	Negative	5	100.00%
	Positive	0	0.00%
Hypertension			

	Negative	4	80.00%
	Positive	1	20.00%
Cervical involvement			
	Negative	5	100.00%
	Positive	0	0.00%
Parametrial infiltration			
	Negative	4	80.00%
	Positive	1	20.00%

9. What is “conservative treatment” mentioned on line 431?

The "conservative treatment" mentioned in line 431 specifically refers to endocrine therapy, a therapeutic regimen primarily based on progestin drugs, such as medroxyprogesterone acetate and megestrol acetate. These drugs inhibit the proliferation of endometrial cancer cells by regulating hormone levels, and are commonly used in endometrial cancer patients who need to preserve fertility or cannot tolerate surgery.

Minor Comments:

1. The graphical abstract: the arrows in the MS half of the schematic are not logically placed, and it makes it difficult to clearly summarize the mechanism being described.

Thank you for your valuable feedback. We have revised the mechanism diagram to make it more concise and logical. In accordance with the requirements of the journal, the graphical abstract has been moved to Figure 8.

REDACTED

2. Please reword for clarity:

a. Line 54: “MS reprogramming polyamine and promoting EC progression”. In its current state, it is unclear what key mechanism is referring to.

Thank you for your attention. Since this teaser is not required by Nature Communications, we have removed this part. Thank you.

b. Line 89: “Rearrangement of glucose, ... preventing redox stress and ferroptosis.”

Thank you. The expression has been revised.

85 Tumor metabolic reprogramming is a hallmark for tumors. ~~Rearranged metabolism of~~
86 ~~glucose, lipids, and amino acids serves a dual role: it supplies energy for rapid tumor~~
87 ~~proliferation and migration, while also providing metabolic intermediates that protect~~
88 ~~against redox stress and ferroptosis~~¹⁴. Metabolites also act as oncometabolites to promote

3. Thank you for clarifying the authors' focus on endometrioid EC for the authors' analysis. Perhaps authors could add that high incidence and its link to metabolic disorders are the reason for their focus on EC.

Thank you for your valuable suggestion. We have supplemented in the first part of the Results (lines 127-129). This addition aims to emphasize the epidemiological basis and metabolic correlation of focusing on this subtype, so as to enhance the rationality of the research background.

125 To clarify the key metabolites involved in the malignant progression of EC caused by MS,
126 we screened and included a total of 62 postmenopausal endometrioid type EC patients
127 between 2012 and 2018 (mean age 60 ± 7 years). ~~Endometrioid type is the most common~~
128 ~~histological type of EC (accounting for approximately 80%), and its incidence is closely~~
129 ~~associated with metabolic disorders (such as obesity and insulin resistance).~~ There were 30

4. Figure 1j: Please state that red=high (or upregulated) and blue=low (or downregulated) in the figure caption.

Thank you for your suggestions, which have been supplemented. We suspect that you might be referring to Figure 1c, and we have added description in the caption of Figure 1c.

1263 (c), Left: Top 15 metabolites identified by PLS-DA (VIP>1); right: colored boxes indicate
1264 relative metabolite concentrations (red: high/upregulated; blue:
1265 low/downregulated). Right: Quantification of spermine (Mann-Whitney test) and
1266 N-acetylputrescine (Student's t-test). Data: mean \pm SD (MS+: N=30; MS-: N=32).
1267 * $p < 0.05$, ** $p < 0.01$, *** $p < 0.001$, **** $p < 0.0001$.[↵]

5. Supplementary Figure 2a: Is there a clearer way to indicate that the authors are looking at HL patients only? Based on the authors' labels from the previous figure, it looks like these samples are HL negative.

Thank you for your insightful suggestion. We have made the corresponding revisions.

Fig. S2

6. Figure 2: Please add the FFA acronym description to the caption.

Thank you for the suggestion. The clarification has been added.

[295 (a), ODC1 protein levels (WB) in Ishikawa and AN3CA cells treated with various free
 [296 fatty acids (FFAs) (30µM). Right: Grayscale quantification of ODC1 blots. Data:
 [297 mean ± SEM (n=3 independent experiments). *p<0.05, **p<0.01, ***p<0.001,
 [298 ****p<0.0001 One-way ANOVA. ←

[352 FFA: free fatty acid. DL: delipidated. OA: oleic acid. PA: palmitic acid. SA: Stearic Acid.
 [353 AA: Arachidonic acid. LA: Lauric acid. BA: Butyric acid. HA: Hexanoic acid. DEA:
 [354 Decanoic, acid. CA: Caprylic acid. MA: Myristic acid. LOA: Linoleic acid. EEC:
 [355 Endometrioid endometrial cancer. BzCl: Benzoyl chloride. Source data are provided as a
 [356 Source Data file. Exact p-values are provided in the Source Data file. ←

7. Figure 3l caption: The levels of expression were not measured using IHC but detected.

Additionally, instead of “Up” and “Down”, the figures were “left” and “right”.

Thank you for your meticulous review. We have revised and simplified the figure caption for Figure 3l. The indicator terms have been adjusted according to the latest layout. Additionally, we have checked the captions of other figures for similar issues and made corresponding revisions to ensure consistency and accuracy.

l406 (I), Upper: IHC staining of ODC1 and HOXB9 in 17 EC tissues with LVSI or LNM. Scale
 l407 bars: 50 μ m. Lower: Spearman correlation of HOXB9 and ODC1 IHC scores in
 l408 these tissues. \leftarrow

8. Line 206: Cited Supplementary Figures 2c and 2d when referring to Supplementary Figure 3. Additionally, supplementary figure 3 citations are out of order. (ie: SF3c and SF3d are cited before SF3a or SF3b).

Thank you for your careful review! We have corrected the citation errors and order issues for Supplementary Figure 3. We also conducted a full-text check to ensure all figure citations (main text and legends) match the correct numbering and subfigure order.

9. Line 307: “Our previous work ... proteasome pathway”. Please state this earlier in the text when the authors first introduce Praja in Supplementary Figure 4.

Thank you for highlighting this issue! We have added a clarification on Praja2 protein at line 291 where Supplementary Figure 4g is first cited, ensuring proper contextualization upon its introduction. We appreciate your attention to the manuscript's coherence, which helps improve the clarity of our findings.

286 them (Supplementary Figure 4i). Our previous work has shown that Praja2 is an E3
287 ubiquitin ligase for HOXB9, which binds to Praja2 and is degraded through a ubiquitin-
288 dependent proteasome pathway³⁷. Transfection with Flag-*ODC1* inhibits the interaction
289 between HOXB9 and Praja2 (Supplementary Figure 4j), suggesting a positive feedback
290 regulation between HOXB9 and ODC1.⁴

10. Figure 4: Please state what the boxes are indicating in each subfigure (where applicable).
Applicable for supplementary figure 5, as well.

Thank you for the reminder! At Reviewer 3's suggestion, we've labeled the key rescue groups (si*ODC1* + Flag-*HOXB9* and si*HOXB9* + Flag-*ODC1*) in boxes within Figure 4 and Supplementary Figure 5, as these groups require special attention amid multiple experimental groups. The figure legends now clearly indicate these boxed rescue groups in Figure 4 and Supplementary Figure 5.

Fig. 4. HOXB9 is required for OA-induced accumulation of ODC1 and EC progression

(a), Protein levels of ODC1, HOXB9, and ZEB1 (WB) in AN3CA cells treated with/without OA (30 μ M), transfected with siNC, siHOXB9, siODC1, Flag, Flag-HOXB9, or Flag-ODC1 for 72h (OA added 48h before harvest). **Right: Grayscale quantification of ODC1 blots. Data: mean \pm SEM (n=3 independent experiments). *p < 0.05, **p < 0.01, ***p < 0.001, ****p < 0.0001 One-way ANOVA. Boxed areas indicate key rescue groups: siODC1 + Flag-HOXB9 and siHOXB9 + Flag-ODC1.**

(b), ODC1 protein levels (WB) in Ishikawa cells treated with/without OA (30 μ M), transfected with siNC or siHOXB9 for 48h; CHX (100 μ g/mL) added at different time points before harvest. Right: ODC1 degradation curve (based on left blot

grayscale quantification). Data: mean \pm SEM (n=3 independent experiments). *p < 0.05, **p < 0.01, ***p < 0.001, ****p < 0.0001 **Two-way ANOVA.**

(c), Wound-healing assay for migration ability of Ishikawa and AN3CA cells treated with/without OA (30 μ M), transfected with siNC, siHOXB9, siODC1, Flag, Flag-HOXB9, or Flag-ODC1 for 72h (OA added 48h before harvest). Data: mean \pm SEM (n=3 independent experiments). *p < 0.05, **p < 0.01, ***p < 0.001, ****p < 0.0001 **One-way ANOVA. Boxed areas indicate key rescue groups: siODC1 + Flag-HOXB9 and siHOXB9 + Flag-ODC1.**

(d), Transwell assay for migration ability of Ishikawa and AN3CA cells under conditions in (c). Data: mean \pm SEM (n=3 independent experiments). *p < 0.05, **p < 0.01, ***p < 0.001, ****p < 0.0001 **One-way ANOVA. Boxed areas indicate key rescue groups: siODC1 + Flag-HOXB9 and siHOXB9 + Flag-ODC1.**

(e), LC-MS of BzCl-derivatized putrescine, spermidine, and spermine in AN3CA cells under conditions in (c). Data: mean \pm SD (n=3). *p < 0.05, **p < 0.01, ***p < 0.001, ****p < 0.0001 **One-way ANOVA. Boxed areas indicate key rescue groups: siODC1 + Flag-HOXB9 and siHOXB9 + Flag-ODC1.**

OA: Oleic acid. **BzCl:** Benzoyl chloride. Source data are provided as a Source Data file. Exact p-values are provided in the Source Data file.

Fig. S5. HOXB9-ODC1 is required for OA-induced polyamine accumulation (related to Figure 4)

(a) ODC1 and HOXB9 protein levels (WB) in Ishikawa cells: treated with/without OA (30 μ M), transfected with *siNC*, *siHOXB9*, *siODC1*, Flag, Flag-*HOXB9*, or Flag-*ODC1* for 72h (OA added 24h before harvest). **Boxed areas indicate key rescue groups: *siODC1* + Flag-*HOXB9* and *siHOXB9* + Flag-*ODC1*.**

(b) AN3CA cells were treated with or without OA (30 μ M) and transfected with *siNC*, *siHOXB9*, Flag, or Flag-*HOXB9* for 48 hours. CHX (100 μ g/mL) was added at different time points prior to protein harvesting. ODC1 protein levels were detected by WB. Right: ODC1 degradation curve

(based on left blot grayscale quantification). Data: mean \pm SEM (n=3 independent experiments). *p < 0.05, **p < 0.01, ***p < 0.001, ****p < 0.0001 One-way ANOVA.

(c) Ishikawa cells were treated with or without OA (30 μ M) and transfected with Flag or Flag-*HOXB9* for 48 hours. CHX (100 μ g/mL) was added at different time points prior to protein harvesting. ODC1 protein levels were detected by WB. Right: ODC1 degradation curve (based on left blot grayscale quantification). Data: mean \pm SEM (n=3 independent experiments). *p < 0.05, **p < 0.01, ***p < 0.001, ****p < 0.0001 One-way ANOVA.

(d) LC-MS of *BzCl*-derivatized putrescine, spermidine, and spermine in Ishikawa cells: treated with/without OA (30 μ M) and transfected with *siNC*, *siHOXB9*, *siODC1*, Flag, Flag-*HOXB9*, or Flag-*ODC1* for 72 hours (OA added 24h before harvest). Data: mean \pm SD (n=3). *p < 0.05, **p < 0.01, ***p < 0.001, ****p < 0.0001 One-way ANOVA. **Boxed areas indicate key rescue groups: *siODC1* + Flag-*HOXB9* and *siHOXB9* + Flag-*ODC1*.**

DL: delipidated. **OA:** Oleic acid. **BzCl:** Benzoyl chloride. **Source data are provided as a Source Data file. Exact p-values are provided in the Source Data file.**

11. Figure 7a: Please clearly indicate which frames are the enlarged frames being called out in the figure caption.

Thank you for the reminder! The relevant revisions have been completed: currently, in Figure 7a, the areas selected for enlargement have been clearly marked with dashed lines, and the label "enlarge" has been added in the figure to ensure the magnified regions are clearly indicated.

We greatly appreciate your meticulous review, which has helped enhance the rigor of our manuscript.

12. Figure 7f: The blue frames do not have a correlating scale. Is it just a nuclear stain?

Thank you for your question! The blue-framed images in Figure 7f do have scale bars in their lower left corners. The colors shown are pseudocolor images representing Raman scattering at corresponding wavelengths, not nuclear stains.

13. Figures 7i-j: Please do not group the captions together. Each subfigure should have their own caption.

- a. The IHC is not of the patient, but of a tissue. Please indicate what tissue.
- b. Please check the authors' units. Does "uM" mean "µm"?
- c. Please check the spelling of "Carboplatin" both in the authors' figure and the caption.

Thank you for the feedback! The figure legends for Figure 7i and 7j have been separated to clarify each subfigure. The legends now explicitly state:

(i), PTCs from a reproductive-aged EC patient (stage IA Grade 2, with hyperlipidemia, insulin resistance, and recurrence after endocrine therapy). Left: Sagittal pelvic MRI of the patient; IHC staining for ODC1 and HOXB9 in EC tissue sections. Right: 3D and 2D confocal images of PTCs stained with CK (green), vimentin (red), and DAPI (blue). Scale bars: 10 µm.↵

(j) Viability of PTCs after 7-day treatment with different concentrations of DFMO, paclitaxel, carboplatin, or DFMO (5mM) co-administration. Data: mean ± SD (n=3) *p < 0.05, **p < 0.01, ***p < 0.001, ****p < 0.0001 Two-way ANOVA.

This revision ensures each subfigure has independent labeling and specifies the tissue type as patient-derived EC.

b. Unit correction (uM to µM)

The notation "uM" was a typo and has been corrected to "µM" (micromolar, equivalent to µmol/L) throughout the figures and legends.

c. Carboplatin spelling correction

The spelling of "Carboplatin" has been thoroughly checked and corrected in all figures and legends to ensure consistency and accuracy.

We appreciate your meticulous review, which helps improve the precision and professionalism of our manuscript.

14. Supplementary figure 7c: please indicate the number of samples tested.

Thank you for the reminder! The number of samples tested in Supplementary Figure 7c has been clearly indicated in the figure.

15. Supplementary figure 8b: It is unclear why the authors have shown these images when they were already added to Figure 6d. It would be expected that the result of the HFD and LFD should have been consistent between the LMN and subcutaneous models. If not, this needs to be explained.

We apologize for the unclear labeling in the figure legend. To clarify:

Figure 6d shows MRI images of mice in the lymph node metastasis (LMN) model, all of which successfully underwent obesity modeling.

493 (d), Representative MRI images of fat tissue in the renal capsular area from two groups of
494 mice (LNM model; HFD: n=4, LFD: n=4) after 2-month diet induction.

Supplementary Figure 8b depicts MRI images of mice in the subcutaneous tumor model, and these mice also achieved successful obesity modeling.

(b) Representative MRI images of fat content in the renal capsule plane of mice (subcutaneous tumor model, sg group; HFD: n=3, LFD: n=3) after 2-month diet induction.

This supplementation was performed as requested by Reviewer 3 to provide quantitative analysis and compare results between different models (LMN vs. subcutaneous). The images illustrate distinct model systems, and we have ensured the legends now clearly differentiate between them while highlighting the successful obesity induction in both models.

Thank you for helping us improve the clarity of our data presentation.

16. Figure 7i – the authors show no staining for immune markers in this panel, though the authors claim these PTCs have immune cells present.

Thank you for your feedback! We note that the manuscript previously mentioned the presence of immune cells in PTCs based on another study of ours that is currently under publication, but this study does not provide direct evidence. To avoid misinterpretation, we have removed the relevant statement to ensure all claims align with the data presented in this study.

440 Excitingly, we applied DFMO to patient-derived tumor cells (PTCs)⁵³ of EC. **These PTCs**
441 **are similar to EC tissue, as are organoids, and contain components such as epithelial cells**
442 **and fibroblasts** (Figure 7h and i). A patient with stage IA grade 2 EC experienced

17. Please be consistent with untargeted and non-targeted between figures and the main text. This is still not accomplished in the revised version.

Thank you! We found it in the supplementary materials and have revised it accordingly.

(b) Association of spermine intensity (**untargeted metabolomics**) with pathological factors (grade, LNM, LVSI, MI, ovarian involvement, stage) in the identification cohort (N=62). Data: mean \pm SD. *p < 0.05, **p < 0.01, ***p < 0.001, ****p < 0.0001. **Mann-Whitney-U test (two groups); Kruskal-Wallis test (multiple groups)**. ↵

Reviewer #2 (Remarks to the Author):

Reviewer #3 (Remarks to the Author):

I thank the authors for their efforts in assessing some our important comments. I think the new sets of data from patients, including measurements of polyamines and OA levels in tumors and assessing confounding factors such as metformin on their pathway of interest, really strengthen the overall concept of the study. The new in vivo experiments also strengthen the manuscript, although I would still have comments listed below. However, I am still not convinced about the in vitro data, and I regret that the authors did not assess important comments from our side, which were crucial in my view (please see comments below). Overall, I find the manuscript valuable and of interest for the scientific community, but I still think that it needs major revisions to represent data according to good science practices and to improve the trust of the readership.

Major comments:

- I really regret that authors did not follow the advice raised by us and the other reviewers concerning quantification of their western blots, especially for panels 2a, 2g, 2l, 3j, 3k, 4a, 5b, S3b, S4a, S7d, S8a, S8h for which we had specifically asked for. It is not sufficient to adjust only pictures' contrast. In vitro data are still mostly qualitative and seems to be only N=1

which does not support good scientific practice. I would prefer to have “less” crowded figures, but crucial experiments to be repeated several times with proper statistical analysis. What was the reason for ignoring these comments ?

We sincerely apologize for not conducting the quantification of Western blots as advised. All experiments (Figures 2a, 2g, 2l, 3j-k, 4a, 5b, S3b, S4a, 7d, 8a, 8h) were independently repeated ≥ 3 times, and quantitative analyses (gray value statistics) along with statistical graphs have now been supplemented in the figures. The raw blots and statistical data are provided in the Source data file.

Thank you for emphasizing methodological rigor, which has improved our work.

Figure 2a

Figure 2g The Quantification is in Figure S3e

Figure 2l

Figure 2

Figure 3j has now been split into Figure 3j and Figure S4f.

Figure 3

Fig.S4

Figure 3k has now been split into Figure 3k and Figure S4g.

Figure 3

Fig.S4

Figure 4a

Figure 4

Figure S5a

Figure S5

Figure 5b

Figure 5 b

Figure S3b has now been revised to Figure S3c.

Fig.S3

Figure S4a

Fig.S4

Figure S7d has now been revised to Figure S7e.

Figure S8a has now been revised to Figure S7j.

Fig. S7

Figure S8h has now been revised to Figure S8g.

- Also, co-IP data are really qualitative and often not sufficient on their own. Co-IP data are usually supported by luciferase assays for instance, to get quantitative activity data and to support the overall concept via another approach. For instance authors could use cells overexpressing ODC1-luciferase and activity could be recorded with different doses of HOXB9 overexpression plasmid in the presence or not of OA. Such data would strongly support the stability assays, which are not very convincing and mostly non-significant between DL and OA, except for one early time point in ISK cells.

Thank you very much for your valuable suggestion. We highly appreciate your insight that Co-IP data are qualitative and need to be supported by quantitative assays like luciferase assays to strengthen the overall conclusion.

Following your recommendation, we constructed a Flag-*ODC1*-luc plasmid and designed experiments to observe the effects of GFP-*HOXB9* on the stability of Flag-*ODC1*-luc and the changes in luciferase signal under different doses of GFP-*HOXB9* transfection, with or without OA treatment.

After extensive experimental optimization, we successfully addressed challenges arising from co-transfection, including variability in transfection efficiency, difficulties in balancing protein expression levels, and limitations imposed by intracellular saturation effects. Through systematic refinement, we optimized the experimental conditions and established the following protocol: Ishikawa and AN3CA cells were transfected with 3 μ g of Flag-*ODC1*-luc plasmid in 10-cm dishes. On day 3

(D3), cells were trypsinized and seeded into 6-well plates at a density equivalent to 0.25 μg of Flag-*ODC1*-luc per well. On D4, cells were co-transfected with 0.1–0.6 μg of GFP-*HOXB9* plasmid, supplemented with empty GFP vector to a total of 1 μg DNA per well, with/without OA. On D6, luciferase activity was measured. Parallel samples were processed for protein quantification using a BCA Protein Assay Kit. Luciferase signals were normalized to total protein concentration to control for transfection efficiency and cell number variability. This two-step approach ensured each well contained only 0.25 μg of Flag-*ODC1*-luc, thereby avoiding saturation effects on its degradation or stabilization pathways. Under these conditions, exogenous GFP-*HOXB9* and OA treatment exerted maximal effects.

The results of this optimized experiment clearly showed that GFP-*HOXB9* could stabilize the luciferase signal in a dose-dependent manner, and OA could further enhance this fluorescence signal. These data perfectly support our conclusion and address your concern. This result has been included in the Supplementary Figure 4e, and the methodology has also been listed in the "Materials and Methods" section.

Thank you again for your constructive suggestion, which has significantly improved the robustness of our study.

- This is also the case for *in vivo* experiments, where I am still not convinced about knock-down efficiency for *HOXB9*. It is mandatory to assess mRNA levels and to quantify WBs to prove the significant reduction induced by the different silencing approaches, as quality

controls. E.g. figures S8a, S8h.

Thank you for your question. Following your suggestion, we have supplemented the corresponding experiments, and the relevant images have been added to the figures.

For the subcutaneous mouse model (sg group): Statistical analyses of WB bands have been supplemented in Figure S7j (previously Figure S8a) and Figure S8g (previously Figure S8h). The qPCR results of xenograft tissues have been added to Figure S8h. Additionally, the statistical results of WB and qPCR for Ishikawa cell lines stably infected with sgNC, sgHOXB9, and sgODC1, which were used in this model, have been supplemented in Figure S8a. These multi-angle data collectively verify the knockdown efficiency.

Fig. S7

Fig. S8

Fig. S8

For the subcutaneous mouse model (sh group): The statistical analysis of WB results for xenograft tissues has been supplemented in Figure S7e. The WB and qPCR results of Ishikawa cells stably infected with shNC, shHOXB9, and shODC1 (which were used in this

model) have been supplemented in Figure S7d, and these results demonstrate sufficient knockdown efficiency in the cells. In the xenograft tissues, the LFD group also shows favorable knockdown effects, whereas the knockdown efficiency of HOXB9 in the HFD group may be compromised by high lipids.

Fig. S7

For the fat pad injection lymph node metastasis (LNM) mouse model (sh group): The WB and qPCR results of AN3CA cells stably infected with shNC, shHOXB9, and shODC1 (which were used in this model) have been supplemented in Figure S8i, and these results demonstrate excellent knockdown efficiency.

Fig. S8

- We had asked authors to include the N of each experiment in the figure legends, particularly for those western blots, which seemed to be only N=1. However, this has not been done, lacking transparency about data.

We sincerely appreciate your attention to the annotation of sample sizes (N) in the figure legends. We have explicitly indicated the sample size (N) for each experiment in all figure legends, including the Western blot experiments.

All Western blot experiments were performed with appropriate independent repetitions, and the N values corresponding to these experiments have been clearly stated in the respective figure legends to ensure the transparency of our data.

We are grateful for your valuable feedback, which reminds us to double-check the clarity of such annotations, and we will continue to ensure the rigor and transparency of our work.

- Concerning statistical analysis, all reviewers asked for more appropriate statistical analysis – checking for normality and applying non-parametric and parametric tests whenever necessary. However, authors have decided by themselves whether data “seemed” normal or not based on QQ plots if their data did not pass normality tests, and therefore applied only parametric tests throughout the whole manuscript. Plus, for some graphs with more than 2 groups, authors have decided to use both ANOVA and t test (Figure 4c-e for instance), probably to get more significance, which is completely wrong. This refers to my previous comment where, in my view, it is better to have less experiments but a higher N for more reliable data, proper statistical analysis and conclusions.

We sincerely appreciate your rigorous comments on statistical analysis, which are crucial for enhancing the robustness of our study.

Regarding statistical testing: We have revised the statistical analyses comprehensively. All datasets were re-evaluated for normality, with parametric tests applied to normally distributed data and nonparametric tests to non-normally distributed data. Corresponding adjustments have been made to the relevant data, figures, and the Methods section. Raw data, including normality test results, and details of parametric/nonparametric analyses, are provided in the source data file.

Concerning the statistical methods in Figures 4c-e: Thank you for pointing out the ambiguity. The figures actually use one-way ANOVA (consistent with the Methods section stating that ANOVA is applied for comparisons involving more than two groups). The misleading description "Student's t-test or one-way ANOVA" in the figure legends, intended to indicate a choice of method based on group numbers, has been corrected. We have thoroughly reviewed all figure legends to clarify the specific statistical methods used and apologize for the confusion.

We are grateful for your valuable feedback, which helps strengthen the rigor of our work.

- I am still not convinced about the specific effect of OA... First figure 2a has to be repeated and quantified to ensure that only OA can exhibit such an effect on ODC1. Second, for a few crucial experiments, such as lipid droplet formation and effect on HOXB9 and ODC1 stability, I would use another fatty acid as negative control, like PA which is strongly enriched in metabolic syndrome and a more toxic lipid species than OA. PA will also induce lipid droplet formation; I would then be curious to know the impact on the pathway of interest. Overall, PA would be a more suitable negative control than DL for the overall concept.

Thank you for your insightful comments on the specific effects of OA. We have addressed your suggestions as follows:

Figure 2a has been independently replicated ≥ 3 times with supplementary statistical quantification, confirming that OA can stably upregulate the expression of ODC1 (see Figure a below).

Following your advice, we conducted experiments using palmitic acid (PA) as a control. The results showed that although PA, similar to OA, can induce the formation of lipid droplets, unlike OA, PA does not prolong the half-lives of HOXB9 or ODC1 (see Figures b and c below) nor enhance the interaction between them (see Figure d below). These results verify the specificity of OA's effects.

Figure a. OA stably upregulates ODC1.

Figure b. PA on ODC1 half-life: No significant enhancement was observed.

Figure c. PA on HOXB9 half-life: No significant enhancement was observed.

Figure d. PA on HOXB9 and ODC1 interaction: No significant enhancement was observed.

- Also, HFD is mostly made of “lard and soy oil” as mentioned in the methods section. So HFD is more enriched in PA and n-6 fatty acid than OA, which support even more to make a few experiments *in vitro* to rule out the role of other fatty acids on their pathway of interest. Thank you for your astute observation regarding the composition of the high-fat diet (HFD), which is rich in palmitic acid (PA) and n-6 fatty acids—with their content exceeding that of oleic acid (OA). This insight further underscores the importance of clarifying the specific role of OA in our proposed pathway, and we fully agree with the rationale behind your suggestion.

As addressed earlier, we have replicated the key experiment in Figure 2a with statistical quantification, confirming that OA stably upregulates ODC1. However, we wish to clarify two points:

Our findings do not imply that PA is less pro-carcinogenic than OA. Instead, we highlight a specific mechanism by which OA promotes carcinogenesis via HOXB9 and ODC1—whereas PA may exert its effects through other pathways unrelated to HOXB9/ODC1.

We acknowledge the biological significance of other fatty acids in the HFD; our focus on OA’s unique role in this particular pathway does not negate the importance of investigating other lipids in broader contexts.

These clarifications aim to better contextualize our conclusions, and we appreciate your guidance in refining the specificity of our claims.

Minor comments:

- Concerning in vivo experiments, I would include the new dataset with sgRNA in the main figure, as the effect is more convincing than the old dataset.

Thank you for your suggestion on in vivo experiments. We agree that the new sgRNA dataset, with stronger efficacy, enhances persuasiveness. Accordingly, we have included the tumor images in Figure 6m. We appreciate your guidance.

- Is HOXB9 known to bind lipids such as the PPAR nuclear receptor family ? Or is it something authors assume it works this way ? Do authors think HOXB9 only binds OA over other fatty acids ?

Thank you for your insightful questions regarding HOXB9's lipid-binding properties, which have helped clarify the novelty and scope of our findings.

Traditionally, HOXB9 is known to function primarily in the nucleus as a transcriptional regulator, and its ability to bind lipids (including OA) has not been previously reported. A key innovation of our study is the discovery that HOXB9 can localize to the cytoplasm, where it binds OA and subsequently interacts with ODC1—an unexpected role distinct from its canonical nuclear function. It should be emphasized that our work does not involve any assumptions, nor does it address whether nuclear HOXB9, similar to PPAR, can bind lipids in the nucleus. However, your perspective is highly interesting and merits attention in future studies.

As noted, our current work focuses on characterizing the interaction between HOXB9 and OA, and we have not yet validated or excluded its potential binding to other fatty acids. This remains an important avenue for future investigation.

We appreciate your attention to this mechanistic detail, which helps frame the specificity and boundaries of our current findings.

- I am surprised that authors could not measure total free fatty acid levels in serum for “technical reasons”. What was the problem ? FFA colorimetric assay, such as NEFA-HR R1 and R2 sets from FUJIFILM Wako Chemicals, are very easy to perform.

Thank you for your advice on serum FFA measurement. The initial "technical reasons" stemmed from immature mass spectrometry methods in our institution. Following your suggestion, we used the NEFA-HR kits from FUJIFILM Wako. The results showed that the serum FFA levels in the HFD groups of the three batches of mice were not significantly elevated, except that the serum FFA levels in the subcutaneous tumor model mice (sh group) were increased.

Consistent with previous findings, serum FFA levels in patients with MS+ were not significantly elevated, but there were differences in the levels of TC, TG, and polyamines. In

tumor tissues, high levels of OA were detected in both MS+ patients and HFD mice. This suggests that serum polyamines may serve as potential predictive markers.

These results support the local action of OA: increased uptake by tumor cells (likely from surrounding adipocyte breakdown) elevates local polyamines, promoting tumor progression.

We appreciate your guidance in clarifying the roles of systemic and local OA.

- Figure 6h, S8g: is there a good reason for not showing individual dots here ? Also, please represent the different groups always in the same order and same color codes (4h vs 4i, j, k).

Thank you for your valuable comments. Regarding your points: we have checked all figures, and individual data points are now shown in Figure 6h and Supplementary Figure 8g as in others. Additionally, the order and color coding of different groups have been standardized across all figures, including Figure 4h, 4i, 4j, and 4k, to ensure consistency.

Thanks again for your guidance.

- Once again, authors often mention correlations although they did not perform correlation analysis. Better to use "association" in those cases.

Thank you for your perceptive comment. We truly appreciate your meticulous attention to this detail. We have thoroughly checked the entire manuscript and revised the relevant expressions accordingly. Specifically, we have replaced inappropriate references to "correlations" with either "differences" (when describing group comparisons) or "association" (when indicating relationships without formal correlation analysis) as appropriate in each context. Thank you again for your valuable guidance.

- Page 10, line 206: shouldn't be figures S3C and S3D instead of S2C and S2D ? I think figures S3A and S3B are not cited in the main text.

Thank you for your careful attention to this detail. We apologize for the incorrect figure

numbering. We have corrected the reference from Supplementary Figures 2C and 2D to Supplementary Figures 3C and 3D as you pointed out. Additionally, we have adjusted the citation order of the figures, ensuring that Supplementary Figures 3A to 3D are cited in sequence throughout the manuscript.

- Page 16, lines 364-367: “whereas knock-down of HOXB9 [...] reduced polyamines and OA (Figure 6j and k, S7g)”. Not true, NS for HOXB9 silencing compared to HFD controls for all these panels.

Thank you for your careful observation. We have revised the relevant description to accurately reflect that there were no significant differences in polyamine and OA levels between the HOXB9-silenced group, as shown in the figures. We appreciate your valuable guidance in improving the accuracy of our manuscript.

371	sera induced by HFD (Figure 6i, j and Supplementary Figure 7g). We further measured
372	the levels of polyamines and OA in subcutaneous xenograft tumor tissues and found that
373	tumor size was closely correlated with putrescine content in the tissues (Figure 6k, l and
374	Supplementary Figure 7h, i). Polyamine levels were increased in xenograft tumors from
375	HFD-fed mice, whereas ODC1 knockdown led to a reduction in polyamines (Figure 6k
376	and Supplementary Figure 7h). ⁴

- Discussion, page 20, line 443-444: “We identified OA, the most prevalent and enriched monounsaturated fatty acids”. How can authors make such a claim without performing lipidomics in patients, or at least their mouse models under HFD ? If this is a fact of general knowledge, please include a proper reference.

Thank you for your insightful comment. We appreciate your professional perspective on this point.

As suggested, we have toned down the relevant statement in the discussion to avoid overreaching, ensuring the description of OA's characteristics is more cautious and aligned with the scope of our study. We have revised this sentence more precisely as follows:

457	polyamine metabolism emerged as a significant scientific question. We identified OA
458	(Figure 2), a relatively common and widely studied monounsaturated fatty acid, as a

- Discussion, page 21 line 463-466: “we approached the problem from the angle of metabolic

reprogramming in tumor cells and their microenvironment”. This statement is not really accurate, as authors did not look at the microenvironment per se, otherwise explain better which part of the manuscript support such a claim.

Thank you for your astute observation. We fully agree with your comment that our previous statement overreached in referencing the tumor microenvironment, as our study did not specifically investigate it.

We have revised the relevant sentence in the discussion (Page 21, Lines 463-466) to accurately reflect that our approach focused on metabolic reprogramming in tumor cells, removing the mention of the microenvironment to ensure consistency with the actual scope of our research. Thank you again for your valuable guidance in refining the precision of our manuscript.

478 and did not individually screen for significant factors in obesity. ~~In this study, we~~
 479 ~~approached the problem from the angle of metabolic reprogramming in tumor cells,~~
 480 ~~examining the interplay between metabolites and oncogenic signals.~~ We identified

- Page 21, line 467: “We established OA as a crucial upstream regulator among various fatty acids”. I disagree as this statement is only supported by figure 2a, with n=1 and no quantification.

Thank you for your critical comment. We fully acknowledge your concern regarding the previous statement about OA being a crucial upstream regulator among various fatty acids.

To address this, we have performed multiple independent replicate experiments and supplemented quantitative data with statistical analyses to support the role of OA, ensuring the reliability of our conclusion. The revised content now accurately reflects the strengthened experimental evidence.

Reviewer #4 (Remarks to the Author):

REVIEWER COMMENTS

Reviewer #1 (Remarks to the Author):

The manuscript has been much improved and better supports main findings.

However, it is still unclear why two different BMI cutoffs for each gene were chosen instead of showing the effect on survival for both cutoffs for each gene independently (Supp Fig 9d-g). It seems to be important information to provide.

Thank you for your valuable suggestion. Following your advice, we have unified the cut-off values and obtained the updated results, which have been incorporated into Supp Fig 9d and f. The relevant text and source data have been revised accordingly. We appreciate your careful review and insightful comments.

Supp Fig 9d-g:

Line 389-399: "... sub-iliac lymph nodes were significantly enlarged". Using "significantly" in this statement is too strong without quantification and statistical analysis.

Thank you for your comment. We fully agree that using "significantly" without quantification and statistical analysis is inappropriate. Following your suggestion, we have statistically analyzed the lymph node areas based on the gross images and scale bar in Supp Fig 8l, and added the corresponding statistical graph to Supp Fig 8l. The relevant text and raw data have been updated respectively. We appreciate your careful review which helps improve the manuscript.

Supp Fig 8l:

Reviewer #2 (Remarks to the Author):

Reviewer #3 (Remarks to the Author):

We thank the authors for the careful 2nd review of their manuscript. The authors have addressed all main points and added experiments or replicates to support their claims. The provision of the source data including statistical analyses immensely strengthens the manuscript.

Minor remaining points:

- Axis labels should be unified. For example, Fig., 6j, k, l all show the same (response area) but have different label including capital letters, unit etc. This is true for multiple other figures also

Thank you for your careful observation. We fully agree that unified axis labels are crucial for clarity, and we appreciate you pointing out this inconsistency. Following your suggestion, we have thoroughly checked all figures, standardized the axis labels (including consistent use of capitalization, units, and terminology), and ensured uniformity across the manuscript.

- Fig 6c, axis label should be mmol

Thank you for your careful note. We agree with your suggestion and have corrected the axis label of Fig. 6c to "mmol". Thank you again for helping us ensure the accuracy of the figures.

Figure 6

- Regarding question 4 by reviewer 1 (“Figure 3l: The authors’ findings state that there is a correlation of expression of ODC1 and HOXB9 in patients with LVSI or LNM. Comparing between the two groups would be more convincing if the authors are tying this axis to EC progression in MS+ patients”): The authors claim, also in the figure legend, that 17 patients were used for the analysis, yet only 8 data points are shown. This holds true for both the data shown in the rebuttal letter and the manuscript, where the figure legend says 17 and the figure shows data from 11 patients. I assume some “dots” represent more than 1 patient but this then needs to be stated.

- Regarding same question, authors show the correlation between ODC1 score and grade in the rebuttal letter but not in the manuscript - would that not be a valuable addition?

Fig 6 panel a) by figdraw.com

Thank you for your careful review and valuable comments. We fully agree with the issues you pointed out.

Regarding the number of data points in Figure 3l, we have adjusted the figure: for cases where multiple data points overlap, we have increased the size of the corresponding points to better reflect the sample size information.

In response to your suggestion about the correlation between ODC1 score and grade, we agree this is valuable information and have added this analysis to Figure 3l.

The relevant text in the manuscript and raw data have been updated accordingly. Thank you again for your insights, which have helped improve the accuracy and completeness of our work.

Figure 3l:

- For Fig. 5f, 5h: no repetitions are shown in the source data but the legend indicates 3 independent experiments; this data is not quantified

Thank you for your careful review. We apologize for the incomplete presentation of data in the source data file. All experiments were independently repeated at least three times. Following your feedback, we have added statistical charts next to the bands in Figure 5f and supplemented quantitative results above the bands in Figure 5h. The original blots from the three independent repetitions, as well as the raw data for statistical analysis, have now been added to the source data. We appreciate your valuable comments, which have helped improve the rigor and transparency of our manuscript.

Figure 5

Reviewer #4 (Remarks to the Author):

Point-to-Point Response to Reviewers' Comments

Reviewer #1 The authors have satisfactorily addressed my comments from the last round of revisions. I recommend this manuscript for publication.

We thank Reviewer #1 for their positive evaluation and recommendation. All comments from the last round have been fully addressed.

Reviewer #2 I co-reviewed this manuscript with one of the listed reviewers, as part of Nature Communications' initiative to train Early Career Researchers in peer review and recognize their contributions.

We thank Reviewer #2 for the co-review.